# RESTRICTED STRONG CONVEXITY OF DEEP LEARNING MODELS WITH SMOOTH ACTIVATIONS

**Arindam Banerjee**
Department of Computer Science
University of Illinois at Urbana-Champaign
arindamb@illinois.edu

**Pedro Cisneros-Velarde**
Department of Computer Science
University of Illinois at Urbana-Champaign
pacisne@gmail.com

**Libin Zhu**
Department of Computer Science
University of California, San Diego
l5zhu@ucsd.edu

**Mikhail Belkin**
Haliciouglu Data Science Institute
University of California, San Diego
mbelkin@ucsd.edu

## ABSTRACT

We consider the problem of optimization of deep learning models with smooth activation functions. While there exist influential results on the problem from the "near initialization" perspective, we shed considerable new light on the problem. In particular, we make two key technical contributions for such models with $L$ layers, $m$ width, and $\sigma_0^2$ initialization variance. First, for suitable $\sigma_0^2$, we establish a $O(\frac{\text{poly}(L)}{\sqrt{m}})$ upper bound on the spectral norm of the Hessian of such models, considerably sharpening prior results. Second, we introduce a new analysis of optimization based on Restricted Strong Convexity (RSC) which holds as long as the squared norm of the average gradient of predictors is $\Omega(\frac{\text{poly}(L)}{\sqrt{m}})$ for the square loss. We also present results for more general losses. The RSC based analysis does not need the "near initialization" perspective and guarantees geometric convergence for gradient descent (GD). To the best of our knowledge, ours is the first result on establishing geometric convergence of GD based on RSC for deep learning models, thus becoming an alternative sufficient condition for convergence that does not depend on the widely-used Neural Tangent Kernel (NTK). We share preliminary experimental results supporting our theoretical advances.

## 1 INTRODUCTION

Recent years have seen advances in understanding convergence of gradient descent (GD) and variants for deep learning models (Du et al., 2019; Allen-Zhu et al., 2019; Zou & Gu, 2019; Zou et al., 2020; Liu et al., 2022; Ji & Telgarsky, 2019; Oymak & Soltanolkotabi, 2020; Nguyen, 2021). Despite the fact that such optimization problems are non-convex, a series of recent results have shown that GD has geometric convergence and finds near global solution "near initialization" for wide networks. Such analysis is typically done based on the Neural Tangent Kernel (NTK) (Jacot et al., 2018), in particular by showing that the NTK is positive definite "near initialization," in turn implying the optimization problem satisfies a condition closely related to the Polyak-Łojasiewicz (PL) condition, which in turn implies geometric convergence to the global minima (Liu et al., 2022; Nguyen, 2021). Such results have been generalized to more flexible forms of "lazy learning" where similar guarantees hold (Chizat et al., 2019). However, there are concerns regarding whether such "near initialization" or "lazy learning" truly explains the optimization behavior in realistic deep learning models (Geiger et al., 2020; Yang & Hu, 2020; Fort et al., 2020; Chizat et al., 2019).

Our work focuses on optimization of deep models with smooth activation functions, which have become increasingly popular in recent years (Du et al., 2019; Liu et al., 2022; Huang & Yau, 2020). Much of the theoretical convergence analysis of GD has focused on ReLU networks (Allen-Zhu et al., 2019; Nguyen, 2021). Some progress has also been made for deep models with smooth activations, but existing results are based on a variant of the NTK analysis, and the requirements on the width of

such models are high (Du et al., 2019; Liu et al., 2022). Based on such background and context, the motivating question behind our work is: Are there other (meaningful) sufficient conditions beyond NTK which lead to (geometric) convergence of GD for deep learning optimization?

Based on such motivation, we make two technical contributions in this paper which shed light on optimization of deep learning models with smooth activations and with $L$ layers, $m$ width, and $\sigma_0^2$ initialization variance. First, for suitable $\sigma_0^2$, we establish a $O(\frac{\text{poly}(L)}{\sqrt{m}})$ upper bound on the spectral norm of the Hessian of such models (Section 4). The bound holds over a large *layerwise spectral norm* (instead of Frobenius norm) ball $B_{\rho,\rho_1}^{\text{Spec}}(\theta_0)$ around the random initialization $\theta_0$, where the radius $\rho < \sqrt{m}$, arguably much bigger than what real world deep models need. Our analysis builds on and sharpens recent prior work on the topic (Liu et al., 2020). While our analysis holds for Gaussian random initialization of weights with any variance $\sigma_0^2$, the poly$(L)$ dependence happens when $\sigma_0^2 \le \frac{1}{4+o(1)} \frac{1}{m}$ (we handle the $\frac{1}{m}$ scaling explicitly) .

Second, based on our Hessian spectral norm bound, we introduce a new approach to the analysis of optimization of deep models with smooth activations based on the concept of Restricted Strong Convexity (RSC) (Section 5) (Wainwright, 2019; Negahban et al., 2012; Negahban & Wainwright, 2012; Banerjee et al., 2014; Chen & Banerjee, 2015). While RSC has been a core theme in high-dimensional statistics especially for linear models and convex losses (Wainwright, 2019), to the best of our knowledge, RSC has not been considered in the context of non-convex optimization of overparameterized deep models. For a normalized total loss function $\mathcal{L}(\theta) = \frac{1}{n} \sum_{i=1}^{n} \ell(y_i, \hat{y}_i), \hat{y}_i = f(\theta; x_i)$ with predictor or neural network model $f$ parameterized by vector $\theta$ and data points $\{\mathbf{x}_i, y_i\}_{i=1}^{n}$, when $\ell$ corresponds to the square loss we show that the total loss function satisfies RSC on a suitable restricted set $Q_\kappa^t \subset \mathbb{R}^p$ (Definition 5.2 in Section 5) at step $t$ as long as $\left\| \frac{1}{n} \sum_{i=1}^{n} \nabla_\theta f(\theta_t; \mathbf{x}_i) \right\|_2^2 = \Omega(\frac{1}{\sqrt{m}})$. We also present similar results for general losses for which additional assumptions are needed. We show that the RSC property implies a Restricted Polyak-Łojasiewicz (RPL) condition on $Q_\kappa^t$, in turn implying a geometric one-step decrease of the loss towards the minimum in $Q_\kappa^t$, and subsequently implying geometric decrease of the loss towards the minimum in the large (layerwise spectral norm) ball $B_{\rho,\rho_1}^{\text{Spec}}(\theta_0)$. The geometric convergence due to RSC is a novel approach in the context of deep learning optimization which does not depend on properties of the NTK. Thus, the RSC condition provides an alternative sufficient condition for geometric convergence for deep learning optimization to the widely-used NTK condition.

The rest of the paper is organized as follows. We briefly present related work in Section 2 and discuss the problem setup in Section 3. We establish the Hessian spectral norm bound in Section 4 and introduce the RSC based optimization analysis in Section 5. We experimental results corresponding to the RSC condition in Section 6 and conclude in Section 7. All technical proofs are in the Appendix.

## 2 RELATED WORK

The literature on gradient descent and variants for deep learning is increasingly large, and we refer the readers to the following surveys for an overview of the field (Fan et al., 2021; Bartlett et al., 2021). Among the theoretical works, we consider (Du et al., 2019; Allen-Zhu et al., 2019; Zou & Gu, 2019; Zou et al., 2020; Liu et al., 2022) as the closest to our work in terms of their study of convergence on multi-layer neural networks. For a literature review on shallow and/or linear networks, we refer to the recent survey (Fang et al., 2021). Due to the rapidly growing related work, we only refer to the most related or recent work for most parts.

Du et al. (2019); Zou & Gu (2019); Allen-Zhu et al. (2019); Liu et al. (2022) considered optimization of square loss, which we also consider for our main results, and we also present extensions to more general class of loss functions. Zou & Gu (2019); Zou et al. (2020); Allen-Zhu et al. (2019); Nguyen & Mondelli (2020); Nguyen (2021); Nguyen et al. (2021) analyzed deep ReLU networks. Instead, we consider smooth activation functions, similar to (Du et al., 2019; Liu et al., 2022). The convergence analysis of the gradient descent in (Du et al., 2019; Allen-Zhu et al., 2019; Zou & Gu, 2019; Zou et al., 2020; Liu et al., 2022) relied on the near constancy of NTK for wide neural networks (Jacot et al., 2018; Lee et al., 2019; Arora et al., 2019; Liu et al., 2020), which yield certain desirable properties for their training using gradient descent based methods. One such property is related to the PL condition (Karimi et al., 2016; Nguyen, 2021), formulated as PL* condition in (Liu et al., 2022). Our work uses a different optimization analysis based on RSC (Wainwright, 2019; Negahban et al.,

2012; Negahban & Wainwright, 2012) related to a restricted version of the PL condition. Furthermore, Du et al. (2019); Allen-Zhu et al. (2019); Zou & Gu (2019); Zou et al. (2020) showed convergence in value to a global minimizer of the total loss, as we also do.

## 3 PROBLEM SETUP: DEEP LEARNING WITH SMOOTH ACTIVATIONS

Consider a training set $\mathcal{D} = \{\mathbf{x}_i, y_i\}_{i=1}^n, \mathbf{x}_i \in \mathcal{X} \subseteq \mathbb{R}^d, y_i \in \mathcal{Y} \subseteq \mathbb{R}$. We will denote by $X \in \mathbb{R}^{n \times d}$ the matrix whose $i$th row is $\mathbf{x}_i^\top$. For a suitable loss function $\ell$, the goal is to minimize the empirical loss: $\mathcal{L}(\theta) = \frac{1}{n} \sum_{i=1}^n \ell(y_i, \hat{y}_i) = \frac{1}{n} \sum_{i=1}^n \ell(y_i, f(\theta; \mathbf{x}_i))$, where the prediction $\hat{y}_i := f(\theta; \mathbf{x}_i)$ is from a deep model, and the parameter vector $\theta \in \mathbb{R}^p$. In our setting $f$ is a feed-forward multi-layer (fully-connected) neural network with depth $L$ and widths $m_l, l \in [L] := \{1, \ldots, L\}$ given by

$$
\begin{aligned}
\alpha^{(0)}(\mathbf{x}) &= \mathbf{x}, \\
\alpha^{(l)}(\mathbf{x}) &= \phi\left(\frac{1}{\sqrt{m_{l-1}}} W^{(l)} \alpha^{(l-1)}(\mathbf{x})\right), \quad l = 1, \ldots, L, \\
f(\theta; \mathbf{x}) = \alpha^{(L+1)}(\mathbf{x}) &= \frac{1}{\sqrt{m_L}} \mathbf{v}^\top \alpha^{(L)}(\mathbf{x}),
\end{aligned}
\tag{1}
$$

where $W^{(l)} \in \mathbb{R}^{m_l \times m_{l-1}}, l \in [L]$ are layer-wise weight matrices, $\mathbf{v} \in \mathbb{R}^{m_L}$ is the last layer vector, $\phi(\cdot)$ is the smooth (pointwise) activation function, and the total set of parameters

$$
\theta := (\text{vec}(W^{(1)})^\top, \ldots, \text{vec}(W^{(L)})^\top, \mathbf{v}^\top)^\top \in \mathbb{R}^{\sum_{k=1}^L m_k m_{k-1} + m_L}, \tag{2}
$$

with $m_0 = d$. For simplicity, we will assume that the width of all the layers is the same, i.e., $m_l = m$, $l \in [L]$, and so that $\theta \in \mathbb{R}^{Lm^2+m}$. For simplicity, we also consider deep models with only one output, i.e., $f(\theta; \mathbf{x}) \in \mathbb{R}$ as in (Du et al., 2019), but our results can be extended to multi-dimension outputs as in (Zou & Gu, 2019), using $\mathbf{V} \in \mathbb{R}^{m_L \times k}$ for $k$ outputs at the last layer; see Appendix C.

Define the pointwise loss $\ell_i := \ell(y_i, \cdot) : \mathbb{R} \to \mathbb{R}_+$ and denote its first- and second-derivative as $\ell_i' := \frac{d\ell(y_i, \hat{y}_i)}{d\hat{y}_i}$ and $\ell_i'' := \frac{d^2 \ell(y_i, \hat{y}_i)}{d\hat{y}_i^2}$. The particular case of square loss is $\ell(y_i, \hat{y}_i) = (y_i - \hat{y}_i)^2$. We denote the gradient and Hessian of $f(\cdot; \mathbf{x}_i) : \mathbb{R}^p \to \mathbb{R}$ as $\nabla_i f := \frac{\partial f(\theta; \mathbf{x}_i)}{\partial \theta}$, and $\nabla_i^2 f := \frac{\partial^2 f(\theta; \mathbf{x}_i)}{\partial \theta^2}$. The *neural tangent kernel* (NTK) $K_{\text{ntk}}(\cdot; \theta) \in \mathbb{R}^{n \times n}$ corresponding to parameter $\theta$ is defined as $K_{\text{ntk}}(\mathbf{x}_i, \mathbf{x}_j; \theta) = \langle \nabla_i f, \nabla_j f \rangle$. By chain rule, the gradient and Hessian of the empirical loss w.r.t. $\theta$ are given by $\frac{\partial \mathcal{L}(\theta)}{\partial \theta} = \frac{1}{n} \sum_{i=1}^n \ell_i' \nabla_i f$ and $\frac{\partial^2 \mathcal{L}(\theta)}{\partial \theta^2} = \frac{1}{n} \sum_{i=1}^n [\ell_i'' \nabla_i f \nabla_i f^\top + \ell_i' \nabla_i^2 f]$. Let $\|\cdot\|_2$ denote the spectral norm for matrices and $L_2$-norm for vectors

We make the following assumption regarding the activation function $\phi$:

**Assumption 1** (**Activation function**). *The activation $\phi$ is 1-Lipschitz, i.e., $|\phi'| \leq 1$, and $\beta_\phi$-smooth, i.e., $|\phi_l''| \leq \beta_\phi$.*

**Remark 3.1.** Our analysis holds for any $\varsigma_\phi$-Lipchitz smooth activations, with a dependence on $\varsigma_\phi$ on most key results. The main (qualitative) conclusions stay true if $\varsigma_\phi \leq 1 + o(1)$ or $\varsigma_\phi = \text{poly}(L)$, which is typically satisfied for commonly used smooth activations and moderate values of $L$.   □

We define two types of balls over parameters that will be used throughout our analysis.

**Definition 3.1** (**Norm balls**). *Given $\bar{\theta} \in \mathbb{R}^p$ of the form (2) with parameters $\overline{W}^{(l)}, l \in [L], \overline{\mathbf{v}}$, we define*

$$
B_{\rho, \rho_1}^{\text{Spec}}(\bar{\theta}) := \left\{ \theta \in \mathbb{R}^p \text{ as in (2)} \mid \|W^{(\ell)} - \overline{W}^{(\ell)}\|_2 \leq \rho, \ell \in [L], \|\mathbf{v} - \bar{\mathbf{v}}\|_2 \leq \rho_1 \right\}, \tag{3}
$$

$$
B_\rho^{\text{Euc}}(\bar{\theta}) := \left\{ \theta \in \mathbb{R}^p \text{ as in (2)} \mid \|\theta - \bar{\theta}\|_2 \leq \rho \right\}. \tag{4}
$$

**Remark 3.2.** The layerwise spectral norm ball $B_{\rho, \rho_1}^{\text{Spec}}$ plays a key role in our analysis. The last layer radius of $\rho_1$ gives more flexibility, and we will usually assume $\rho_1 \leq \rho$; e.g., we could choose the desirable operating regime of $\rho < \sqrt{m}$ and $\rho_1 = O(1)$. Our analysis in fact goes through for any choice of $\rho, \rho_1$ and the detailed results will indicate specific dependencies on both $\rho$ and $\rho_1$.   □

## 4 SPECTRAL NORM OF THE HESSIAN OF THE MODEL

We start with the following assumption regarding the random initialization of the weights.

**Assumption 2** (**Initialization weights and data normalization**). *The initialization weights $w_{0,ij}^{(l)} \sim \mathcal{N}(0, \sigma_0^2)$ for $l \in [L]$ where $\sigma_0 = \frac{\sigma_1}{2\left(1 + \frac{\sqrt{\log m}}{\sqrt{2m}}\right)}, \sigma_1 > 0$, and $\mathbf{v}_0$ is a random unit vector with $\|\mathbf{v}_0\|_2 = 1$. Further, we assume the input data satisfies: $\|\mathbf{x}_i\|_2 = \sqrt{d}, i \in [n]$.*

We focus on bounding the spectral norm of the Hessian $\|\nabla_\theta^2 f(\theta; \mathbf{x})\|_2$ for $\theta \in B_{\rho,\rho_1}^{\mathrm{Spec}}(\theta_0)$ and any input $\mathbf{x} \in \mathbb{R}^d$ with $\|\mathbf{x}\|_2 = \sqrt{d}$. The assumption $\|\mathbf{x}\|_2 = \sqrt{d}$ is for convenient scaling, such assumptions are common in the literature (Allen-Zhu et al., 2019; Oymak & Soltanolkotabi, 2020; Nguyen et al., 2021). Prior work (Liu et al., 2020) has considered a similar analysis for $\theta \in B_\rho^{\mathrm{Euc}}(\theta_0)$, effectively the layerwise Frobenius norm ball, which is much smaller than $B_{\rho,\rho_1}^{\mathrm{Spec}}(\theta_0)$, the layerwise spectral norm ball. We choose a unit value for the last layer's weight norm for convenience, since our results hold under appropriate scaling for any other constant in $O(1)$. All missing proofs are in Appendix A.

**Theorem 4.1** (**Hessian Spectral Norm Bound**). *Under Assumptions 1 and 2, for $\theta \in B_{\rho,\rho_1}^{\mathrm{Spec}}(\theta_0)$, with probability at least $(1 - \frac{2(L+1)}{m})$, for any $\mathbf{x}_i, i \in [n]$, we have*

$$\left\|\nabla_\theta^2 f(\theta; \mathbf{x}_i)\right\|_2 \leq \frac{c_H}{\sqrt{m}}, \tag{5}$$

*with $c_H = O(L^5(1 + \gamma^{6L})(1 + \rho_1))$ where $\gamma := \sigma_1 + \frac{\rho}{\sqrt{m}}$.*

**Remark 4.1** (**Desirable operating regimes**). The constant $\gamma$ needs careful scrutiny as $c_H$ depends on $\gamma^{6L}$. Let us choose $\rho_1 = O(\mathrm{poly}(L))$. For any choice of the spectral norm radius $\rho < \sqrt{m}$, we can choose $\sigma_1 \leq 1 - \frac{\rho}{\sqrt{m}}$ ensuring $\gamma \leq 1$ and hence $c_H = O(\mathrm{poly}(L))$. If $\rho = O(1)$, we can keep $\sigma_1 = 1$ so that $\gamma = 1 + \frac{O(1)}{\sqrt{m}}$, and $c_H = O(\mathrm{poly(L)})$ as long as $L < \sqrt{m}$, which is common. Both of these give good choices for $\sigma_1$ and desirable operating regime for the result. If we choose $\sigma_1 > 1$, an undesirable operating regime, then $c_H = O(c^{\Theta(L)})$, $c > 1$, and we will need $m = \Omega(c^{\Theta(L)})$ for the result to be of interest. $\qquad\square$

**Remark 4.2** (**Recent Related Work**). In recent work, Liu et al. (2020) analyzed the Hessian spectral norm bound and showed that $c_H = \tilde{O}(\rho^{3L})$ for $\theta \in B_\rho^{\mathrm{Euc}}(\theta_0)$ (logarithmic terms hidden in $\tilde{O}(\cdot)$). Our analysis builds on and sharpens the result in (Liu et al., 2020) in three respects: (a) we have $c_H = O(\mathrm{poly}(L)(1 + \gamma^{6L}))$ for $\rho_1 = O(\mathrm{poly}(L))$ where we can choose $\sigma_1$ to make $\gamma \leq 1$ and thus obtain $c_H = O(\mathrm{poly}(L))$, instead of the worse $c_H = \tilde{O}(\rho^{3L})$ in Liu et al. (2020)[1]; (b) even for the same $\rho$, our results hold for a much larger spectral norm ball $B_{\rho,\rho_1}^{\mathrm{Spec}}(\theta_0)$ compared to their Euclidean norm ball $B_\rho^{\mathrm{Euc}}(\theta_0)$ in (Liu et al., 2020); and (c) to avoid an exponential term, the bound in (Liu et al., 2020) needs $\rho \leq 1$ whereas our result can use radius $\rho < \sqrt{m}$ for all intermediate layer matrices and $\rho_1 = O(\mathrm{poly}(L))$ for the last layer vector. Moreover, as a consequence of (b) and (c), our results hold for a larger (spectral norm) ball whose radius can increase with $m$, unlike the results in Liu et al. (2020) which hold for a smaller (Euclidean) ball with constant radius, i.e., "near initialization." $\quad\square$

**Remark 4.3** (**Exact constant $c_H$**). For completeness, we show the exact expression of the constant $c_H$ in Theorem 4.1 so the dependencies on different factors is clear. Let $h(l) := \gamma^{l-1} + |\phi(0)| \sum_{i=1}^{l-1} \gamma^{i-1}$. Then,

$$c_H = 2L(L^2\gamma^{2L} + L\gamma^L + 1) \cdot (1 + \rho_1) \cdot \psi_H \cdot \max_{l \in [L]} \gamma^{L-l} + 2L\gamma^L \max_{l \in [L]} h(l), \tag{6}$$

where

$$\psi_H = \max_{1 \leq l_1 < l_2 \leq L} \left\{ \beta_\phi(h(l_1))^2, h(l_1)\left(\frac{\beta_\phi}{2}(\gamma^2 + (h(l_2))^2) + 1\right), \beta_\phi\gamma^2 h(l_1)h(l_2) \right\}. \tag{7}$$

The source of the terms will be discussed shortly. Note the dependence on $\rho_1$, the radius for the last layer in $B_{\rho,\rho_1}^{\mathrm{Spec}}(\theta_0)$, and why $\rho_1 = O(\mathrm{poly}(L))$ is a desirable operating regime. $\qquad\square$

---

[1] See the end of Appendix A for a quick note about the network architecture in our work and the one in (Liu et al., 2020).

Next, we give a high level outline of the proof of Theorem 4.1.

*Proof sketch.* Our analysis follows the structure developed in Liu et al. (2020), but is considerably sharper as discussed in Remark 4.2. We start by defining the following quantities: $\mathcal{Q}_\infty(f) := \max_{1 \le l \le L} \left\{ \left\| \frac{\partial f}{\partial \alpha^{(l)}} \right\|_\infty \right\}$, $\frac{\partial f}{\partial \alpha^{(l)}} \in \mathbb{R}^m$, $\mathcal{Q}_2(f) := \max_{1 \le l \le L} \left\{ \left\| \frac{\partial \alpha^{(l)}}{\partial \mathbf{w}^{(l)}} \right\|_2 \right\}$, $\mathbf{w}^{(l)} := \text{vec}(W^{(l)})$, $\frac{\partial \alpha^{(l)}}{\partial \mathbf{w}^{(l)}} \in \mathbb{R}^{m \times m^2}$, and $\mathcal{Q}_{2,2,1}(f)$ is the maximum over $1 \le l_1 < l_2 < l_3 \le L$ among the three quantities $\left\| \frac{\partial^2 \alpha^{(l_1)}}{\partial \mathbf{w}^{(l_1)2}} \right\|_{2,2,1}$, $\left\| \frac{\partial \alpha^{(l_1)}}{\partial \mathbf{w}^{(l_1)}} \right\|_2 \left\| \frac{\partial^2 \alpha^{(l_2)}}{\partial \alpha^{(l_2-1)} \partial \mathbf{w}^{(l_2)}} \right\|_{2,2,1}$, and $\left\| \frac{\partial \alpha^{(l_1)}}{\partial \mathbf{w}^{(l_1)}} \right\|_2 \left\| \frac{\partial \alpha^{(l_2)}}{\partial \mathbf{w}^{(l_2)}} \right\|_2 \left\| \frac{\partial^2 \alpha^{(l_3)}}{\partial \alpha^{(l_3-1)2}} \right\|_{2,2,1}$. where for an order-3 tensor $T \in \mathbb{R}^{d_1 \times d_2 \times d_3}$ we define the $(2,2,1)$−norm as $\|T\|_{2,2,1} := \sup_{\|\mathbf{x}\|_2 = \|\mathbf{z}\|_2 = 1} \sum_{k=1}^{d_3} \left| \sum_{i=1}^{d_1} \sum_{j=1}^{d_2} T_{ijk} x_i z_j \right|$, $\mathbf{x} \in \mathbb{R}^{d_1}, \mathbf{z} \in \mathbb{R}^{d_2}$. The following result in (Liu et al., 2020) provides an upper bound to the spectral norm of the Hessian.

**Theorem 4.2** (Liu et al. (2020), Theorem 3.1). *Under Assumptions 1, assuming there is $\delta$ such that* $\left\| \frac{\partial \alpha^{(l)}}{\partial \alpha^{(l-1)}} \right\|_2 \le \delta$, *with $C_1 \le L^2 \delta^{2L} + L \delta^L + L$ and $C_2 \le L \delta^L$, we have*

$$\left\| \nabla_\theta^2 f(\theta; \mathbf{x}) \right\|_2 \le 2 C_1 \mathcal{Q}_{2,2,1}(f) \mathcal{Q}_\infty(f) + \frac{2}{\sqrt{m}} C_2 \mathcal{Q}_2(f) , \tag{8}$$

In order to prove Theorem 4.1, we prove that Theorem 4.2 holds with high-probability where $\delta = \gamma$, $\mathcal{Q}_2(f) = O(L(1 + \gamma^L))$, $\mathcal{Q}_{2,2,1}(f) = O(L^3(1 + \gamma^{3L}))$, and $\mathcal{Q}_\infty(f) = O\left( \frac{(1 + \gamma^L)(1 + \rho_1)}{\sqrt{m}} \right)$. Thus we obtain that the upper bound (4.2) becomes $O\left( \frac{\text{poly}(L)(1 + \gamma^{6L})(1 + \rho_1)}{\sqrt{m}} \right)$, providing a benign polynomial dependence on $L$ when $\gamma \le 1$, rather than an exponential dependence on the radius $\rho$ as in (Liu et al., 2020). $\square$

The analysis for bounding the spectral norm of the Hessian can be used to establish additional bounds, which we believe are of independent interest, some of which will be used later in Section 5. First, we bound the norms of gradient of the predictor and the loss w.r.t. the weight vector $\theta$ and the input data $\mathbf{x}$.

**Lemma 4.1** (**Predictor gradient bounds**). *Under Assumptions 1 and 2, for $\theta \in B_{\rho,\rho_1}^{\text{Spec}}(\theta_0)$, with probability at least $\left(1 - \frac{2(L+1)}{m}\right)$, we have*

$$\|\nabla_\theta f(\theta; \mathbf{x})\|_2 \le \varrho \qquad and \qquad \|\nabla_\mathbf{x} f(\theta; \mathbf{x})\|_2 \le \frac{\gamma^L}{\sqrt{m}}(1 + \rho_1) , \tag{9}$$

*with $\varrho^2 = (h(L+1))^2 + \frac{1}{m}(1+\rho_1)^2 \sum_{l=1}^L (h(l))^2 \gamma^{2(L-l)}$, $\gamma = \sigma_1 + \frac{\rho}{\sqrt{m}}$, $h(l) = \gamma^{l-1} + |\phi(0)| \sum_{i=1}^{l-1} \gamma^{i-1}$.*

**Remark 4.4.** Our analysis in Lemma 4.1 provides a bound on the Lipschitz constant of the predictor, a quantity which has generated interest in recent work on robust training (Salman et al., 2019; Cohen et al., 2020; Bubeck & Sellke, 2021).

Under the assumption of square losses, further bounds can be obtained.

**Lemma 4.2** (**Loss bounds**). *Consider the square loss. Under Assumptions 1, and 2, for $\gamma = \sigma_1 + \frac{\rho}{\sqrt{m}}$, each of the following inequalities hold with probability at least $\left(1 - \frac{2(L+1)}{m}\right)$: $\mathcal{L}(\theta_0) \le c_{0,\sigma_1}$ and $\mathcal{L}(\theta) \le c_{\rho_1,\gamma}$ for $\theta \in B_{\rho,\rho_1}^{\text{Spec}}(\theta_0)$, where $c_{a,b} = \frac{2}{n} \sum_{i=1}^n y_i^2 + 2(1+a)^2 |g(b)|^2$ and $g(a) = a^L + |\phi(0)| \sum_{i=1}^L a^i$ for any $a, b \in \mathbb{R}$.*

**Corollary 4.1** (**Loss gradient bound**). *Consider the square loss. Under Assumptions 1 and 2, for $\theta \in B_{\rho,\rho_1}^{\text{Spec}}(\theta_0)$, with probability at least $\left(1 - \frac{2(L+1)}{m}\right)$, we have $\|\nabla_\theta \mathcal{L}(\theta)\|_2 \le 2\sqrt{\mathcal{L}(\theta)} \varrho \le 2\sqrt{c_{\rho_1,\gamma}} \varrho$, with $\varrho$ as in Lemma 4.1 and $c_{\rho_1,\gamma}$ as in Lemma 4.2.*

## 5 Optimization Guarantees with Restricted Strong Convexity

We focus on minimizing the empirical loss $\mathcal{L}(\theta)$ over $\theta \in B^{\mathrm{Spec}}_{\rho,\rho_1}(\theta_0)$, the layerwise spectral norm ball in (3). Our analysis is based on Restricted Strong Convexity (RSC) (Negahban et al., 2012; Banerjee et al., 2014; Chen & Banerjee, 2015; Wainwright, 2019), which relaxes the definition of strong convexity by only needing strong convexity in certain directions or over a subset of the ambient space. We introduce the following specific definition of RSC with respect to a tuple $(S, \theta)$.

**Definition 5.1** (**Restricted Strong Convexity (RSC)**). *A function $\mathcal{L}$ is said to satisfy $\alpha$-restricted strong convexity ($\alpha$-RSC) with respect to the tuple $(S, \theta)$ if for any $\theta' \in S \subseteq \mathbb{R}^p$ and some fixed $\theta \in \mathbb{R}^p$, we have $\mathcal{L}(\theta') \geq \mathcal{L}(\theta) + \langle \theta' - \theta, \nabla_\theta \mathcal{L}(\theta) \rangle + \frac{\alpha}{2} \|\theta' - \theta\|_2^2$, with $\alpha > 0$.*

Note that $\mathcal{L}$ being $\alpha$-RSC w.r.t. $(S, \theta)$ does not need $\mathcal{L}$ to be convex on $\mathbb{R}^p$. Let us consider a sequence of iterates $\{\theta_t\}_{t \geq 0} \subset \mathbb{R}^p$. Our RSC analysis will rely on the following $Q_\kappa^t$-sets at step $t$, which avoid directions almost orthogonal to the average gradient of the predictor. We define the following notation: for two vectors $\pi$ and $\bar{\pi}$, $\cos(\pi, \bar{\pi})$ denotes the cosine of the angle between $\pi$ and $\bar{\pi}$.

**Definition 5.2** ($\mathbf{Q_\kappa^t}$ **sets**). *For iterate $\theta_t \in \mathbb{R}^p$, let $\bar{\mathbf{g}}_t = \frac{1}{n} \sum_{i=1}^n \nabla_\theta f(\theta_t; \mathbf{x}_i)$. For any $\kappa \in (0, 1]$, define $Q_\kappa^t := \{\theta \in \mathbb{R}^p \mid |\cos(\theta - \theta_t, \bar{\mathbf{g}}_t)| \geq \kappa\}$.*

We define the set $B_t := Q_\kappa^t \cap B^{\mathrm{Spec}}_{\rho,\rho_1}(\theta_0) \cap B^{\mathrm{Euc}}_{\rho_2}(\theta_t)$. We focus on establishing RSC w.r.t. the tuple $(B_t, \theta_t)$, where $B^{\mathrm{Spec}}_{\rho,\rho_1}(\theta_0)$ becomes the feasible set for the optimization and $B^{\mathrm{Euc}}_{\rho_2}(\theta_t)$ is a Euclidean ball around the current iterate.

**Assumption 3** (**Loss function**). *The loss $\ell_i$, $i \in [n]$, is (i) strongly convex, i.e., $\ell_i'' \geq a > 0$ and (ii) smooth, i.e., $\ell_i'' \leq b$.*

Assumption 3 is satisfied by commonly used loss functions such as square loss, where $a = b = 2$. We state the RSC result for square loss; the result for other losses and proofs of all technical results in this section are in Appendix B.

**Theorem 5.1** (**RSC for Square Loss**). *For square loss, under Assumptions 1 and 2, with probability at least $(1 - \frac{2(L+1)}{m})$, $\forall \theta' \in Q_\kappa^t \cap B^{\mathrm{Spec}}_{\rho,\rho_1}(\theta_0) \cap B^{\mathrm{Euc}}_{\rho_2}(\theta_t)$ with $\theta_t \in B^{\mathrm{Spec}}_{\rho,\rho_1}(\theta_0)$,*

$$\mathcal{L}(\theta') \geq \mathcal{L}(\theta_t) + \langle \theta' - \theta_t, \nabla_\theta \mathcal{L}(\theta_t) \rangle + \frac{\alpha_t}{2} \|\theta' - \theta_t\|_2^2, \quad \text{with} \quad \alpha_t = c_1 \|\bar{\mathbf{g}}_t\|_2^2 - \frac{c_2}{\sqrt{m}}, \tag{10}$$

*where $\bar{\mathbf{g}}_t = \frac{1}{n} \sum_{i=1}^n \nabla_\theta f(\theta_t; \mathbf{x}_i)$, $c_1 = 2\kappa^2$ and $c_2 = 2c_H(2\varrho\rho_2 + \sqrt{c_{\rho_1,\gamma}})$, with $c_H$ as in Theorem 4.1, $\varrho$ as in Lemma 4.1, and $c_{\rho_1,\gamma}$ as in Lemma 4.2. Consequently, $\mathcal{L}$ satisfies RSC w.r.t. $(Q_\kappa^t \cap B^{\mathrm{Spec}}_{\rho,\rho_1}(\theta_0) \cap B^{\mathrm{Euc}}_{\rho_2}(\theta_t), \theta_t)$ whenever $\alpha_t > 0$.*

**Remark 5.1.** The RSC condition $\alpha_t > 0$ is satisfied at iteration $t$ as long as $\|\bar{\mathbf{g}}_t\|_2^2 > \frac{c_2}{c_1 \sqrt{m}}$ where $c_1, c_2$ are exactly specified in Theorem 5.1. Indeed, if $\gamma$ (and so $\sigma_1$ and $\rho$) is chosen according to the desirable operating regimes (see Remark 4.1), $\rho_1 = O(\mathrm{poly}(L))$ and $\rho_2 = O(\mathrm{poly}(L))$, then we can use the bounds from Lemma 4.2 and obtain that the RSC condition is satisfied when $\|\bar{\mathbf{g}}_t\|_2^2 > \frac{O(\mathrm{poly}(L))}{\sqrt{m}}$. The condition is arguably mild, does not need the NTK condition $\lambda_{\min}(K_{\mathrm{ntk}}(\cdot; \theta_t)) > 0$, and is expected to hold till convergence (see Remark 5.3). Moreover, it is a local condition at step $t$ and has no dependence on being "near initialization" in the sense of $\theta_t \in B^{\mathrm{Euc}}_\rho(\theta_0)$ for $\rho = O(1)$ as in (Liu et al., 2020; 2022). $\qquad\square$

For the convergence analysis, we also need to establish a smoothness property of the total loss.

**Theorem 5.2** (**Local Smoothness for Square Loss**). *For square loss, under Assumptions 1 and 2, with probability at least $(1 - \frac{2(L+1)}{m})$, $\forall \theta, \theta' \in B^{\mathrm{Spec}}_{\rho,\rho_1}(\theta_0)$,*

$$\mathcal{L}(\theta') \leq \mathcal{L}(\theta) + \langle \theta' - \theta, \nabla_\theta \mathcal{L}(\theta) \rangle + \frac{\beta}{2} \|\theta' - \theta\|_2^2, \quad \text{with} \quad \beta = 2\varrho^2 + \frac{2c_H \sqrt{c_{\rho_1,\gamma}}}{\sqrt{m}}, \tag{11}$$

*with $c_H$ as in Theorem 4.1, $\varrho$ as in Lemma 4.1, and $c_{\rho_1,\gamma}$ as in Lemma 4.2. Consequently, $\mathcal{L}$ is locally $\beta$-smooth. Moreover, if $\gamma$ (and so $\sigma_1$ and $\rho$) is chosen according to the desirable operating regimes (see Remark 4.1) and $\rho_1 = O(\mathrm{poly}(L))$, then $\beta = O(\mathrm{poly}(L))$.*

**Remark 5.2.** Similar to the case of the standard strong convexity and smoothness, the RSC and smoothness parameters respectively in Theorems 5.1 and 5.2 satisfy $\alpha_t < \beta$. To see this note that $\alpha_t < 2\kappa^2 \|\bar{\mathbf{g}}_t\|_2^2 \le 2\varrho^2 \le \beta$, where the second inequality follows since $\kappa \le 1$, and $\|\bar{\mathbf{g}}_t\|_2^2 \le \varrho^2$ using Lemma 4.1. $\square$

Next, we show that the RSC condition w.r.t. the tuple $(B_t, \theta_t)$ implies a *restricted* Polyak-Łojasiewicz (RPL) condition w.r.t. the tuple $(B_t, \theta_t)$, unlike standard PL which holds without restrictions (Karimi et al., 2016).

**Lemma 5.1 (RSC $\Rightarrow$ RPL).** *Let $B_t := Q_\kappa^t \cap B_{\rho,\rho_1}^{\mathrm{Spec}}(\theta_0) \cap B_{\rho_2}^{\mathrm{Euc}}(\theta_t)$. In the setting of Theorem 5.1, if $\alpha_t > 0$, then the tuple $(B_t, \theta_t)$ satisfies the Restricted Polyak-Łojasiewicz (RPL) condition, i.e.,*

$$\mathcal{L}(\theta_t) - \inf_{\theta \in B_t} \mathcal{L}(\theta) \le \frac{1}{2\alpha_t}\|\nabla_\theta \mathcal{L}(\theta_t)\|_2^2 , \tag{12}$$

*with probability at least $(1 - \frac{2(L+1)}{m})$.*

For the rest of the convergence analysis, we make the following assumption where $T$ can be viewed as the stopping time so the convergence analysis holds given the assumptions are satisfied.

**Assumption 4 (Iterates' conditions).** *For iterates $\{\theta_t\}_{t=0,1,\ldots,T}$:* **(A4.1)** $\alpha_t > 0$*;* **(A4.2)** $\theta_t \in B_{\rho,\rho_1}^{\mathrm{Spec}}(\theta_0)$.

**Remark 5.3 (Assumption (A4.1)).** From Remark 5.1, (A4.1) is satisfied as long as $\|\bar{\mathbf{g}}_t\|_2^2 > \frac{c_2}{c_1\sqrt{m}}$ where $c_1, c_2$ are as in Theorem 5.1, which is arguably a mild condition. In Section 6 we will present some empirical findings that show that this condition on $\|\bar{\mathbf{g}}_t\|_2^2$ behaves well empirically. $\square$

We now consider the particular case of gradient descent (GD) for the iterates: $\theta_{t+1} = \theta_t - \eta_t \nabla \mathcal{L}(\theta_t)$, where $\eta_t$ is chosen so that $\theta_{t+1} \in B_{\rho,\rho_1}^{\mathrm{Spec}}(\theta_0)$ and $\rho_2$ is chosen so that $\theta_{t+1} \in B_{\rho_2}^{\mathrm{Euc}}(\theta_t)$, which are sufficient for the analysis of Theorem 5.1 — we specify suitable choices in the sequel (see Remark 5.4).

Given RPL w.r.t. $(B_t, \theta_t)$, gradient descent leads to a strict decrease of loss in $B_t$.

**Lemma 5.2 (Local Loss Reduction in $B_t$).** *Let $\alpha_t, \beta$ be as in Theorems 5.1 and 5.2 respectively, and $B_t := Q_\kappa^t \cap B_{\rho,\rho_1}^{\mathrm{Spec}}(\theta_0) \cap B_{\rho_2}^{\mathrm{Euc}}(\theta_t)$. Consider Assumptions 1, 2, and 4, and gradient descent with step size $\eta_t = \frac{\omega_t}{\beta}, \omega_t \in (0, 2)$. Then, for any $\overline{\theta}_{t+1} \in \mathrm{arginf}_{\theta \in B_t} \mathcal{L}(\theta)$, we have with probability at least $(1 - \frac{2(L+1)}{m})$,*

$$\mathcal{L}(\theta_{t+1}) - \mathcal{L}(\overline{\theta}_{t+1}) \le \left(1 - \frac{\alpha_t \omega_t}{\beta}(2 - \omega_t)\right)\left(\mathcal{L}(\theta_t) - \mathcal{L}(\overline{\theta}_{t+1})\right) . \tag{13}$$

Building on Lemma 5.2, we show that GD in fact leads to a geometric decrease in the loss relative to the minimum value of $\mathcal{L}(\cdot)$ in the set $B_{\rho,\rho_1}^{\mathrm{Spec}}(\theta_0)$.

**Theorem 5.3 (Global Loss Reduction in $B_{\rho,\rho_1}^{\mathrm{Spec}}(\theta_0)$).** *Let $\alpha_t, \beta$ be as in Theorems 5.1 and 5.2 respectively, and $B_t := Q_\kappa^t \cap B_{\rho,\rho_1}^{\mathrm{Spec}}(\theta_0) \cap B_{\rho_2}^{\mathrm{Euc}}(\theta_t)$. Let $\theta^* \in \mathrm{arginf}_{\theta \in B_{\rho,\rho_1}^{\mathrm{Spec}}(\theta_0)} \mathcal{L}(\theta)$, $\overline{\theta}_{t+1} \in \mathrm{arginf}_{\theta \in B_t} \mathcal{L}(\theta)$, and $\gamma_t := \frac{\mathcal{L}(\overline{\theta}_{t+1}) - \mathcal{L}(\theta^*)}{\mathcal{L}(\theta_t) - \mathcal{L}(\theta^*)}$. Let $\alpha_t, \beta$ be as in Theorems 5.1 and 5.2 respectively, and $B_t := Q_\kappa^t \cap B_{\rho,\rho_1}^{\mathrm{Spec}}(\theta_0) \cap B_{\rho_2}^{\mathrm{Euc}}(\theta_t)$. Consider Assumptions 1, 2, and 4, and gradient descent with step size $\eta_t = \frac{\omega_t}{\beta}, \omega_t \in (0, 2)$. Then, with probability at least $(1 - \frac{2(L+1)}{m})$, we have we have $\gamma_t \in [0, 1)$ and*

$$\mathcal{L}(\theta_{t+1}) - \mathcal{L}(\theta^*) \le \left(1 - \frac{\alpha_t \omega_t}{\beta}(1 - \gamma_t)(2 - \omega_t)\right)\left(\mathcal{L}(\theta_t) - \mathcal{L}(\theta^*)\right) . \tag{14}$$

As long as the conditions in Theorem 5.3 are kept across iterations, there will be a geometric decrease in loss. For Assumption 4, we have discussed (A4.1) in Remark 5.1, and we discuss (A4.2) next.

**Remark 5.4 (Assumption (A4.2)).** Consider we run gradient descent iterations until some stopping time $T > 0$. Given radius $\rho < \sqrt{m}$, Assumption (A4.2) $\theta_t \in B_{\rho,\rho_1}^{\mathrm{Spec}}(\theta_0)$, $t = 0, \ldots, T$, can be verified empirically. Alternatively, we can choose suitable step sizes $\eta_t$ to ensure the property using the geometric convergence from Theorem 5.3. Assume that our goal is to get $\mathcal{L}(\theta_T) - \mathcal{L}(\theta^*) \leq \epsilon$. Then, with $\chi_T := \min_{t \in [T]} \frac{\alpha_t \omega_t}{\beta}(1 - \gamma_t)(2 - \omega_t)$, Assumption (A4.1) along with Remark 5.2 ensures $\chi_T < 1$. Then, it suffices to have $T = \lceil \log(\frac{\mathcal{L}(\theta_0) - \mathcal{L}(\theta^*)}{\epsilon}) / \log \frac{1}{1 - \chi_T} \rceil = \Theta(\log \frac{1}{\epsilon})$. Then, to ensure $\theta_t \in B_{\rho,\rho_1}^{\mathrm{Spec}}(\theta_0), t \in [T]$, in the case of the square loss, since $\|\nabla \mathcal{L}(\theta_t)\|_2 \leq c$ for some constant $c$ (see Corollary 4.1), it suffices to have $\eta_t \leq \frac{\min\{\rho,\rho_1\}}{\Theta(\log \frac{1}{\epsilon})}$. Moreover, we point out that having $\rho_2 \geq \eta_t c$ ensures $\|\theta_{t+1} - \theta_t\|_2 \leq \rho_2 \Rightarrow \theta_{t+1} \in B_{\rho_2}^{\mathrm{Euc}}(\theta_t)$, which in this case can be guaranteed if $\rho_2 \geq \frac{\min\{\rho,\rho_1\}}{\Theta(\log \frac{1}{\epsilon})}$. The argument above is informal, but illustrates that Assumption (A4.1) along with suitable constant step sizes $\eta_t$ would ensure (A4.2). Thus, Assumption (A4.1), which ensures the RSC condition, is the main assumption behind the analysis. $\square$

The conditions in Assumption 4 (see Remarks 5.1 and 5.4) along with Theorem 5.3 imply that the RSC based convergence analysis holds for a much larger layerwise spectral radius norm ball $B_{\rho,\rho_1}^{\mathrm{Spec}}(\theta_0)$ with any radius $\rho < \sqrt{m}$ and $\rho_1 = O(\mathrm{poly}(L))$.

**Remark 5.5 (RSC and NTK).** In the context of square loss, the *NTK condition* for geometric convergence needs $\lambda_{\min}(K_{\mathrm{ntk}}(\cdot; \theta_t)) \geq c_0 > 0$ for every $t$, i.e., uniformly bounded away from 0 by a constant $c_0 > 0$. The NTK condition can also be written as

$$\inf_{v: \|v\|_2 = 1} \left\| \sum_{i=1}^{n} v_i \nabla_\theta f(\theta_t; \mathbf{x}_i) \right\|_2^2 \geq c_0 > 0 \,. \tag{15}$$

In contrast, the proposed RSC condition (Theorem 5.1) needs

$$\left\| \frac{1}{n} \sum_{i=1}^{n} \nabla_\theta f(\theta_t; \mathbf{x}_i) \right\|_2^2 \geq \frac{\bar{c}_0}{\sqrt{m}} \,, \tag{16}$$

where $m$ is the width and $\bar{c}_0 = \frac{c_2}{c_1}$ where $c_1, c_2$ are constants defined in Theorem 5.1. As a quadratic form on the NTK, the RSC condition can be viewed as using a specific $v$ in (15), i.e., $v_i = \frac{1}{\sqrt{n}}$ for $i \in [n]$, since the RSC condition is $\left\| \sum_{i=1}^{n} \frac{1}{\sqrt{n}} \nabla_\theta f(\theta_t; \mathbf{x}_i) \right\|_2^2 \geq \frac{\bar{c}_0 n}{\sqrt{m}}$. For $m = \Omega(n^2)$, the RSC condition is more general since NTK $\Rightarrow$ RSC, but the converse is not necessarily true. $\square$

**Remark 5.6 (RSC covers different settings than NTK).** The NTK condition may be violated in certain settings, e.g., $\nabla_\theta f(\theta_t; \mathbf{x}_i), i = 1, \ldots, n$ are linearly dependent, $\mathbf{x}_i \approx \mathbf{x}_j$ for some $i \neq j$, layer widths are small $m_l < n$, etc., but the optimization may work in practice. The RSC condition provides a way to analyze convergence in such settings. The RSC condition gets violated when $\frac{1}{n} \sum_{i=1}^{n} \nabla_\theta f(\theta_t; \mathbf{x}_i) \approx 0$, which does not seem to happen in practice (see Section 6), and future work will focus on understanding the phenomena. Finally, note that it is possible to construct a set of gradient vectors which satisfy the NTK condition but violates the RSC condition. Our perspective is to view the NTK and the RSC as two different *sufficient* conditions and geometric convergence of gradient descent (GD) is guaranteed as long as one of them is satisfied in any step. $\square$

## 6 RSC CONDITION: EXPERIMENTAL RESULTS

In this section, we present experimental results verifying the RSC condition $\left\| \frac{1}{n} \sum_{i=1}^{n} \nabla_\theta f(\theta_t; \mathbf{x}_i) \right\|_2^2 = \Omega\left( \frac{\mathrm{poly}(L)}{\sqrt{m}} \right)$, $t = 1, \ldots, T$, on standard benchmarks: CIFAR-10, MNIST, and Fashion-MNIST. For simplicity, as before, we use $\bar{\mathbf{g}}_t = \frac{1}{n} \sum_{i=1}^{n} \nabla_\theta f(\theta_t; \mathbf{x}_i)$.

In Figure 1(a), we consider CIFAR-10 and show the trajectory of $\|\bar{\mathbf{g}}_t\|_2$ over iterations $t$, for different values of the network width $m$. For any width, the value of $\|\bar{\mathbf{g}}_t\|_2$ stabilizes to a constant value over iterations, empirically validating the RSC condition $\|\bar{\mathbf{g}}_t\|_2^2 = \Omega(\mathrm{poly}(L)/\sqrt{m})$. Interestingly, the smallest value of $\|\bar{\mathbf{g}}_t\|_2$ seems to increase with the width. To study the width dependence further, in Figure 1(b), we plot $\min_{t \in [T]} \|\bar{\mathbf{g}}_t\|_2$ as a function of width $m$ for several values of the width.

The plot shows that $\min_{t\in[T]} \|\bar{\mathbf{g}}_t\|_2$ increases steadily with $m$ illustrating that the RSC condition is empirically satisfied more comfortably for wider networks. In Figure 1(c) and (d), we show similar plots for MNIST and Fashion-MNIST illustrating the same phenomena of $\min_{t\in[T]} \|\bar{\mathbf{g}}_t\|_2$ increasing with $m$.

For the experiments, the network architecture we used had 3-layer fully connected neural network with tanh activation function. The training algorithm is gradient descent (GD) width constant learning rate, chosen appropriately to keep the training in NTK regime. Since we are using GD, we use 512 randomly chosen training points for the experiments. The stopping criteria is either training loss $< 10^{-3}$ or number of iterations larger than 3000.

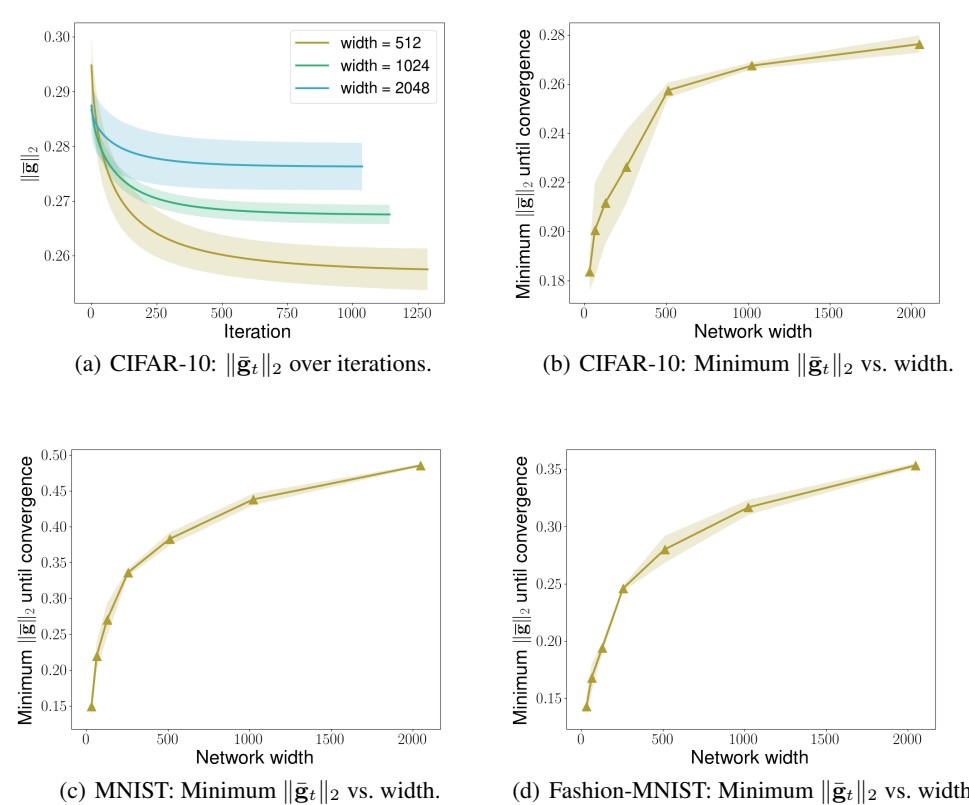

(a) CIFAR-10: $\|\bar{\mathbf{g}}_t\|_2$ over iterations.

(b) CIFAR-10: Minimum $\|\bar{\mathbf{g}}_t\|_2$ vs. width.

(c) MNIST: Minimum $\|\bar{\mathbf{g}}_t\|_2$ vs. width.

(d) Fashion-MNIST: Minimum $\|\bar{\mathbf{g}}_t\|_2$ vs. width.

Figure 1: Experiments on CIFAR-10: (a) $\|\bar{\mathbf{g}}_t\|_2$ over iterations for different network widths $m$; (b) minimum $\|\bar{\mathbf{g}}_t\|_2$ over all iterations, i.e., $\min_{t\in[T]} \|\bar{\mathbf{g}}_t\|_2$, as a function of network width $m$. Experiments on (c) MNIST and (d) Fashion-MNIST. The experiments validates the RSC condition empirically, and illustrates that the condition is more comfortably satisfied for wider networks. Each curve is the average of 3 independent runs.

## 7 CONCLUSIONS

In this paper, we revisit deep learning optimization for feedforward models with smooth activations, and make two technical contributions. First, we bound the spectral norm of the Hessian over a large layerwise spectral norm radius ball, highlighting the role of initialization in such analysis. Second, we introduce a new approach to showing geometric convergence in deep learning optimization using restricted strong convexity (RSC). Our analysis sheds considerably new light on deep learning optimization problems, underscores the importance of initialization variance, and introduces a RSC based alternative to the prevailing NTK based analysis, which may fuel future work.

ACKNOWLEDGMENTS

AB is grateful for support from the National Science Foundation (NSF) through awards IIS 21-31335, OAC 21-30835, DBI 20-21898, as well as a C3.ai research award. MB and LZ are grateful for support from the National Science Foundation (NSF) and the Simons Foundation for the Collaboration on the Theoretical Foundations of Deep Learning2 through awards DMS-2031883 and 814639 as well as NSF IIS-1815697 and the TILOS institute (NSF CCF-2112665).

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

# A  SPECTRAL NORM OF THE HESSIAN

We establish the main theorem from Section 4 in this Appendix.

**Theorem 4.1** (**Hessian Spectral Norm Bound**). *Under Assumptions 1 and 2, for $\theta \in B^{\mathrm{Spec}}_{\rho,\rho_1}(\theta_0)$, with probability at least $(1 - \frac{2(L+1)}{m})$, for any $\mathbf{x}_i, i \in [n]$, we have*

$$\left\| \nabla^2_\theta f(\theta; \mathbf{x}_i) \right\|_2 \leq \frac{c_H}{\sqrt{m}} , \tag{5}$$

*with $c_H = O(L^5(1 + \gamma^{6L})(1 + \rho_1))$ where $\gamma := \sigma_1 + \frac{\rho}{\sqrt{m}}$.*

## A.1  ANALYSIS OUTLINE

Our analysis follows that of Liu et al. (2020) and sharpens the analysis to get better dependence on the depth $L$ of the neural network.

We start by defining the following quantities:

$$\mathcal{Q}_\infty(f) := \max_{1 \leq l \leq L} \left\{ \left\| \frac{\partial f}{\partial \alpha^{(l)}} \right\|_\infty \right\} , \quad \frac{\partial f}{\partial \alpha^{(l)}} \in \mathbb{R}^m , \tag{17}$$

$$\mathcal{Q}_2(f) := \max_{1 \leq l \leq L} \left\{ \left\| \frac{\partial \alpha^{(l)}}{\partial \mathbf{w}^{(l)}} \right\|_2 \right\} , \quad \mathbf{w}^{(l)} := \mathrm{vec}(W^{(l)}) , \quad \frac{\partial \alpha^{(l)}}{\partial \mathbf{w}^{(l)}} \in \mathbb{R}^{m \times m^2} , \tag{18}$$

$$\mathcal{Q}_{2,2,1}(f) := \max_{1 \leq l_1 < l_2 < l_3 \leq L} \left\{ \left\| \frac{\partial^2 \alpha^{(l_1)}}{\partial \mathbf{w}^{(l_1)^2}} \right\|_{2,2,1} , \left\| \frac{\partial \alpha^{(l_1)}}{\partial \mathbf{w}^{(l_1)}} \right\|_2 \left\| \frac{\partial^2 \alpha^{(l_2)}}{\partial \alpha^{(l_2-1)} \partial \mathbf{w}^{(l_2)}} \right\|_{2,2,1} , \right. \tag{19}$$

$$\left. \left\| \frac{\partial \alpha^{(l_1)}}{\partial \mathbf{w}^{(l_1)}} \right\|_2 \left\| \frac{\partial \alpha^{(l_2)}}{\partial \mathbf{w}^{(l_2)}} \right\|_2 \left\| \frac{\partial^2 \alpha^{(l_3)}}{\partial \alpha^{(l_3-1)^2}} \right\|_{2,2,1} \right\} \tag{20}$$

where for an order-3 tensor $T \in \mathbb{R}^{d_1 \times d_2 \times d_3}$ we define the $(2,2,1)-$norm as follows,

$$\|T\|_{2,2,1} := \sup_{\|\mathbf{x}\|_2 = \|\mathbf{z}\|_2 = 1} \sum_{k=1}^{d_3} \left| \sum_{i=1}^{d_1} \sum_{j=1}^{d_2} T_{ijk} x_i z_j \right| , \quad \mathbf{x} \in \mathbb{R}^{d_1}, \mathbf{z} \in \mathbb{R}^{d_2} . \tag{21}$$

We will also use the notation $W^{(L+1)} := \mathbf{v}$.

A key result established in Liu et al. (2020) provides an upper bound to the spectral norm of the Hessian:

**Theorem 4.2** (Liu et al. (2020), Theorem 3.1). *Under Assumptions 1, assuming there is $\delta$ such that $\left\| \frac{\partial \alpha^{(l)}}{\partial \alpha^{(l-1)}} \right\|_2 \leq \delta$, with $C_1 \leq L^2 \delta^{2L} + L\delta^L + L$ and $C_2 \leq L\delta^L$, we have*

$$\left\| \nabla^2_\theta f(\theta; \mathbf{x}) \right\|_2 \leq 2C_1 \mathcal{Q}_{2,2,1}(f) \mathcal{Q}_\infty(f) + \frac{2}{\sqrt{m}} C_2 \mathcal{Q}_2(f) , \tag{8}$$

In order to prove Theorem 4.1, we prove that Theorem 4.2 holds with high-probability where

- $\delta = \gamma$ follows from Lemma A.3,

- $\mathcal{Q}_2(f) = O(L(1 + \gamma^L))$ follows from Lemma A.4,

- $\mathcal{Q}_{2,2,1}(f) = O(L^3(1 + \gamma^{3L}))$ follows from Lemma A.4 and Lemma A.5, and

- $\mathcal{Q}_\infty(f) = O\left( \frac{(1+\gamma^L)(1+\rho_1)}{\sqrt{m}} \right)$ follows from Lemma A.7 ,

while also establishing precise constants to get a proper form for the constant $c_H$ in Theorem 4.1. As a result, $c_H \leq O(\frac{L^5(1+\gamma^{6L})(1+\rho_1)}{\sqrt{m}})$.

## A.2 Spectral norms of $W^{(l)}$ and $L_2$ norms of $\alpha^{(l)}$

We start by bounding the spectral norm of the layer-wise matrices at initialization.

**Lemma A.1.** *Consider any $l \in [L]$. If the parameters are initialized as $w_{0,ij}^{(l)} \sim \mathcal{N}(0, \sigma_0^2)$ where $\sigma_0 = \frac{\sigma_1}{2(1+\sqrt{\frac{\log m}{2m}})}$ as in Assumption 2, then with probability at least $\left(1 - \frac{2}{m}\right)$, we have*

$$\|W_0^{(l)}\|_2 \le \sigma_1 \sqrt{m} \,. \tag{22}$$

*Proof.* For a $(m_l \times m_{l-1})$ random matrix $W_0^{(l)}$ with i.i.d. entries $w_{0,ij}^{(l)} \in \mathcal{N}(0, \sigma_0^2)$, with probability at least $(1 - 2\exp(-t^2/2\sigma_0^2))$, the largest singular value of $W_0$ is bounded by

$$\sigma_{\max}(W_0^{(\ell)}) \le \sigma_0(\sqrt{m_l} + \sqrt{m_{l-1}}) + t \,. \tag{23}$$

This concentration result can be easily derived as follows: notice that $W_0 = \sigma_0 \bar{W}_0^{(\ell)}$, where $\bar{w}_{0,ij}^{(\ell)} \sim N(0,1)$, thus we can use the expectation $\mathbb{E}[\|W_0\|_2^{(\ell)}] = \sigma_0 \mathbb{E}[\|\bar{W}_0\|_2^{(\ell)}] = \sigma_0(\sqrt{m_\ell} + \sqrt{m_{\ell-1}})$ from Gordon's Theorem for Gaussian matrices (Vershynin, 2012, Theorem 5.32) in the Gaussian concentration result for Lipschitz functions (Vershynin, 2012, Proposition 3.4) considering that $B \mapsto \|\sigma_0 B\|_2$ is a $\sigma_0$-Lipschitz function when the matrix $B$ is treated as a vector. Let us choose $t = \sigma_0 \sqrt{2 \log m}$ so that (23) holds with probability at least $(1 - \frac{2}{m})$. Then, to obtain (22),

**Case 1:** $l = 1$. With $m_0 = d$ and $m_1 = m$,

$$\|W_0^{(1)}\|_2 \le \sigma_0(\sqrt{d} + \sqrt{m} + \sqrt{2\log m}) \le \sigma_0(2\sqrt{m} + \sqrt{2\log m})$$

since we are in the over-parameterized regime $m \ge d$.

**Case 2:** $2 \le l \le L$. With $m_l = m_{l-1} = m$,

$$\|W_0^{(l)}\|_2 \le \sigma_0(2\sqrt{m} + \sqrt{2\log m}) \,.$$

Now, using $\sigma_0 = \frac{\sigma_1}{2(1+\sqrt{\frac{\log m}{2m}})}$ in both cases completes the proof. $\square$

Next we bound the spectral norm of layerwise matrices.

**Proposition A.1.** *Under Assumptions 2, for $\theta \in B_{\rho,\rho_1}^{\mathrm{Spec}}(\theta_0)$, with probability at least $\left(1 - \frac{2}{m}\right)$,*

$$\|W^{(l)}\|_2 \le \left(\sigma_1 + \frac{\rho}{\sqrt{m}}\right)\sqrt{m} \qquad, l \in [L].$$

*Proof.* By triangle inequality, for $l \in [L]$,

$$\|W^{(l)}\|_2 \le \|W_0^{(l)}\|_2 + \|W^{(l)} - W_0^{(l)}\|_2 \overset{(a)}{\le} \sigma_1\sqrt{m} + \rho \,,$$

where (a) follows from Lemma A.1. This completes the proof. $\square$

Next, we show that the output $\alpha^{(l)}$ of layer $l$ has an $L_2$ norm bounded by $O(\sqrt{m})$.

**Lemma A.2.** *Consider any $l \in [L]$. Under Assumptions 1 and 2, for $\theta \in B_{\rho,\rho_1}^{\mathrm{Spec}}(\theta_0)$, with probability at least $\left(1 - \frac{2l}{m}\right)$, we have*

$$\|\alpha^{(l)}\|_2 \le \sqrt{m}\left(\sigma_1 + \frac{\rho}{\sqrt{m}}\right)^l + \sqrt{m}\sum_{i=1}^{l}\left(\sigma_1 + \frac{\rho}{\sqrt{m}}\right)^{i-1}|\phi(0)| = \left(\gamma^l + |\phi(0)|\sum_{i=1}^{l}\gamma^{i-1}\right)\sqrt{m} \,.$$

*Proof.* Following Allen-Zhu et al. (2019); Liu et al. (2020), we prove the result by recursion. First, recall that since $\|\mathbf{x}\|_2^2 = d$, we have $\|\alpha^{(0)}\|_2 = \sqrt{d}$. Then, since $m_0 = d$ and $\phi$ is 1-Lipschitz,

$$\left\|\phi\left(\frac{1}{\sqrt{d}}W^{(1)}\alpha^{(0)}\right)\right\|_2 - \|\phi(\mathbf{0})\|_2 \leq \left\|\phi\left(\frac{1}{\sqrt{d}}W^{(1)}\alpha^{(0)}\right) - \phi(\mathbf{0})\right\|_2 \leq \left\|\frac{1}{\sqrt{d}}W^{(1)}\alpha^{(0)}\right\|_2,$$

so that

$$\begin{aligned}
\|\alpha^{(1)}\|_2 = \left\|\phi\left(\frac{1}{\sqrt{d}}W^{(1)}\alpha^{(0)}\right)\right\|_2 &\leq \left\|\frac{1}{\sqrt{d}}W^{(1)}\alpha^{(0)}\right\|_2 + \|\phi(\mathbf{0})\|_2 \\
&\leq \frac{1}{\sqrt{d}}\|W^{(1)}\|_2\|\alpha^{(0)}\|_2 + |\phi(0)|\sqrt{m} \\
&\leq \left(\sigma_1 + \frac{\rho}{\sqrt{m}}\right)\sqrt{m} + |\phi(0)|\sqrt{m},
\end{aligned}$$

where we used Proposition A.1 in the last inequality, which holds with probability at least $1 - \frac{2}{m}$. For the inductive step, we assume that for some $l-1$, we have

$$\|\alpha^{(l-1)}\|_2 \leq \sqrt{m}\left(\sigma_1 + \frac{\rho}{\sqrt{m}}\right)^{l-1} + \sqrt{m}\sum_{i=1}^{l-1}\left(\sigma_1 + \frac{\rho}{\sqrt{m}}\right)^{i-1}|\phi(0)|,$$

which holds with the probability at least $1 - \frac{2(l-1)}{m}$. Since $\phi$ is 1-Lipschitz, for layer $l$, we have

$$\left\|\phi\left(\frac{1}{\sqrt{m}}W^{(l)}\alpha^{(l-1)}\right)\right\|_2 - \|\phi(\mathbf{0})\|_2 \leq \left\|\phi\left(\frac{1}{\sqrt{m}}W^{(l)}\alpha^{(l-1)}\right) - \phi(\mathbf{0})\right\|_2$$
$$\leq \left\|\frac{1}{\sqrt{m}}W^{(l)}\alpha^{(l-1)}\right\|_2,$$

so that

$$\begin{aligned}
\|\alpha^{(l)}\|_2 = \left\|\phi\left(\frac{1}{\sqrt{m}}W^{(l)}\alpha^{(l-1)}\right)\right\|_2 &\leq \left\|\frac{1}{\sqrt{m}}W^{(l)}\alpha^{(l-1)}\right\|_2 + \|\phi(\mathbf{0})\|_2 \\
&\leq \frac{1}{\sqrt{m}}\|W^{(l)}\|_2\|\alpha^{(l-1)}\|_2 + \sqrt{m}|\phi(0)| \\
&\overset{(a)}{\leq} \left(\sigma_1 + \frac{\rho}{\sqrt{m}}\right)\|\alpha^{(l-1)}\|_2 + \sqrt{m}|\phi(0)| \\
&\overset{(b)}{=} \sqrt{m}\left(\sigma_1 + \frac{\rho}{\sqrt{m}}\right)^l + \sqrt{m}\sum_{i=1}^{l}\left(\sigma_1 + \frac{\rho}{\sqrt{m}}\right)^{i-1}|\phi(0)|,
\end{aligned}$$

where (a) follows from Proposition A.1 and (b) from the inductive step. Since we have used Proposition A.1 $l$ times, after a union bound, our result would hold with probability at least $1 - \frac{2l}{m}$. This completes the proof. □

## A.3 SPECTRAL NORMS OF $\frac{\partial\alpha^{(l)}}{\partial\mathbf{w}^{(l)}}$ AND $\frac{\partial\alpha^{(l)}}{\partial\alpha^{(l-1)}}$

Recall that in our setup, the layerwise outputs and pre-activations are respectively given by:

$$\alpha^{(l)} = \phi\left(\tilde{\alpha}^{(l)}\right), \quad \tilde{\alpha}^{(l)} := \frac{1}{\sqrt{m_{l-1}}}W^{(l)}\alpha^{(l-1)}. \tag{24}$$

**Lemma A.3.** *Consider any $l \in \{2, \dots, L\}$. Under Assumptions 1 and 2, for $\theta \in B_{\rho,\rho_1}^{\mathrm{Spec}}(\theta_0)$, with probability at least $\left(1 - \frac{2}{m}\right)$,*

$$\left\|\frac{\partial\alpha^{(l)}}{\partial\alpha^{(l-1)}}\right\|_2^2 \leq \left(\sigma_1 + \frac{\rho}{\sqrt{m}}\right)^2 = \gamma^2. \tag{25}$$

*Proof.* By definition, we have

$$\left[\frac{\partial \alpha^{(l)}}{\partial \alpha^{(l-1)}}\right]_{i,j} = \frac{1}{\sqrt{m}}\phi'(\tilde{\alpha}_i^{(l)})W_{ij}^{(l)} . \tag{26}$$

Since $\|A\|_2 = \sup_{\|\mathbf{v}\|_2=1}\|A\mathbf{v}\|_2$, so that $\|A\|_2^2 = \sup_{\|\mathbf{v}\|_2=1}\sum_i\langle\mathbf{a}_i,\mathbf{v}\rangle^2$, we have that for $2 \le l \le L$,

$$\left\|\frac{\partial \alpha^{(l)}}{\partial \alpha^{(l-1)}}\right\|_2^2 = \sup_{\|\mathbf{v}\|_2=1}\frac{1}{m}\sum_{i=1}^m\left(\phi'(\tilde{\alpha}_i^{(l)})\sum_{j=1}^m W_{ij}^{(l)}v_j\right)^2$$

$$\overset{(a)}{\le} \sup_{\|\mathbf{v}\|_2=1}\frac{1}{m}\|W^{(l)}\mathbf{v}\|_2^2$$

$$= \frac{1}{m}\|W^{(l)}\|_2^2$$

$$\overset{(b)}{\le} \gamma^2 ,$$

where (a) follows from $\phi$ being 1-Lipschitz by Assumption 1 and (b) from Proposition A.1. This completes the proof. $\qquad\square$

**Lemma A.4.** *Consider any $l \in [L]$. Under Assumptions 1 and 2, for $\theta \in B_{\rho,\rho_1}^{\mathrm{Spec}}(\theta_0)$, with probability at least $\left(1 - \frac{2l}{m}\right)$,*

$$\left\|\frac{\partial \alpha^{(l)}}{\partial \mathbf{w}^{(l)}}\right\|_2^2 \le \frac{1}{m}\left[\sqrt{m}\left(\sigma_1 + \frac{\rho}{\sqrt{m}}\right)^{l-1} + \sqrt{m}\sum_{i=1}^{l-1}\left(\sigma_1 + \frac{\rho}{\sqrt{m}}\right)^{i-1}|\phi(0)|\right]^2$$
$$= \left(\gamma^{l-1} + |\phi(0)|\sum_{i=1}^{l-1}\gamma^{i-1}\right)^2 . \tag{27}$$

*Proof.* Note that the parameter vector $\mathbf{w}^{(l)} = \mathrm{vec}(W^{(l)})$ and can be indexed with $j \in [m]$ and $j' \in [d]$ when $l = 1$ and $j' \in [m]$ when $l \ge 2$. Then, we have

$$\left[\frac{\partial \alpha^{(l)}}{\partial \mathbf{w}^{(l)}}\right]_{i,jj'} = \left[\frac{\partial \alpha^{(l)}}{\partial W^{(l)}}\right]_{i,jj'} = \frac{1}{\sqrt{m}}\phi'(\tilde{\alpha}_i^{(l)})\alpha_{j'}^{(l-1)}\mathbb{1}_{[i=j]} . \tag{28}$$

For $l \in \{2,\dots,L\}$, noting that $\frac{\partial \alpha^{(l)}}{\partial \mathbf{w}^{(l)}} \in \mathbb{R}^{m\times m^2}$ and $\|V\|_F = \|\mathrm{vec}(V)\|_2$ for any matrix $V$, we have

$$\left\|\frac{\partial \alpha^{(l)}}{\partial \mathbf{w}^{(l)}}\right\|_2^2 = \sup_{\|V\|_F=1}\frac{1}{m}\sum_{i=1}^m\left(\phi'(\tilde{\alpha}_i^{(l)})\sum_{j,j'=1}^m \alpha_{j'}^{(l-1)}\mathbb{1}_{[i=j]}V_{jj'}\right)^2$$

$$\le \sup_{\|V\|_F=1}\frac{1}{m}\|V\alpha^{(l-1)}\|_2^2$$

$$\le \frac{1}{m}\sup_{\|V\|_F=1}\|V\|_2^2\|\alpha^{(l-1)}\|_2^2$$

$$\overset{(a)}{\le} \frac{1}{m}\|\alpha^{(l-1)}\|_2^2$$

$$\overset{(b)}{\le} \frac{1}{m}\left[\sqrt{m}\left(\sigma_1 + \frac{\rho}{\sqrt{m}}\right)^{l-1} + \sqrt{m}\sum_{i=1}^{l-1}\left(\sigma_1 + \frac{\rho}{\sqrt{m}}\right)^{i-1}|\phi(0)|\right]^2$$

$$= \left(\gamma^{l-1} + |\phi(0)|\sum_{i=1}^{l-1}\gamma^{i-1}\right)^2$$

where (a) follows from $\|V\|_2^2 \leq \|V\|_F^2$ for any matrix $V$, and (b) from Lemma A.2.

The $l = 1$ case follows in a similar manner:

$$\left\|\frac{\partial \alpha^{(1)}}{\partial \mathbf{w}^{(1)}}\right\|_2^2 \leq \frac{1}{d}\|\alpha^{(0)}\|_2^2 = \frac{1}{d}\|\mathbf{x}\|_2^2 = 1$$

which satisfies the form for $l = 1$. That completes the proof. $\qquad \square$

### A.4 $(2, 2, 1)$-NORMS OF ORDER 3 TENSORS

**Lemma A.5.** *Under Assumptions 1 and 2, for $\theta \in B_{\rho,\rho_1}^{\mathrm{Spec}}(\theta_0)$, each of the following inequalities hold with probability at least $\left(1 - \frac{2l}{m}\right)$,*

$$\left\|\frac{\partial^2 \alpha^{(l)}}{(\partial \alpha^{(l-1)})^2}\right\|_{2,2,1} \leq \beta_\phi \gamma^2, \tag{29}$$

$$\left\|\frac{\partial^2 \alpha^{(l)}}{\partial \alpha^{(l-1)} \partial W^{(l)}}\right\|_{2,2,1} \leq \frac{\beta_\phi}{2}\left(\gamma^2 + \left(\gamma^{l-1} + |\phi(0)|\sum_{i=1}^{l-1}\gamma^{i-1}\right)^2\right) + 1, \tag{30}$$

*for $l = 2, \ldots, L$; and*

$$\left\|\frac{\partial^2 \alpha^{(l)}}{(\partial W^{(l)})^2}\right\|_{2,2,1} \leq \beta_\phi\left(\gamma^{l-1} + |\phi(0)|\sum_{i=1}^{l-1}\gamma^{i-1}\right)^2, \tag{31}$$

*for $l \in [L]$.*

*Proof.* For the inequality (29), note that from (26) we obtain $\left(\frac{\partial^2 \alpha^{(l)}}{(\partial \alpha^{(l-1)})^2}\right)_{i,j,k} = \frac{1}{m}\phi''(\tilde{\alpha}_i^{(l)})W_{ik}^{(l)}W_{ij}^{(l)}$, and so

$$\begin{aligned}
\left\|\frac{\partial^2 \alpha^{(l)}}{(\partial \alpha^{(l-1)})^2}\right\|_{2,2,1} &= \sup_{\|\mathbf{v_1}\|_2 = \|\mathbf{v_2}\|_2 = 1} \frac{1}{m}\sum_{i=1}^m \left|\phi''(\tilde{\alpha}_i^{(l)})(W^{(l)}\mathbf{v_1})_i(W^{(l)}\mathbf{v_2})_i\right| \\
&\leq \sup_{\|\mathbf{v_1}\|_2 = \|\mathbf{v_2}\|_2 = 1} \frac{1}{m}\beta_\phi \sum_{i=1}^m \left|(W^{(l)}\mathbf{v_1})_i(W^{(l)}\mathbf{v_2})_i\right| \\
&\overset{(a)}{\leq} \sup_{\|\mathbf{v_1}\|_2 = \|\mathbf{v_2}\|_2 = 1} \frac{1}{2m}\beta_\phi \sum_{i=1}^m (W^{(l)}\mathbf{v_1})_i^2 + (W^{(l)}\mathbf{v_2})_i^2 \\
&\leq \frac{1}{2m}\beta_\phi \sup_{\|\mathbf{v_1}\|_2 = \|\mathbf{v_2}\|_2 = 1} (\|W^{(l)}\mathbf{v_1}\|_2^2 + \|W^{(l)}\mathbf{v_2}\|_2^2) \\
&\leq \frac{1}{2m}\beta_\phi(\|W^{(l)}\|_2^2 + \|W^{(l)}\|_2^2) \\
&\overset{(b)}{\leq} \beta_\phi(\sigma_1 + \rho/\sqrt{m})^2 = \beta_\phi\gamma^2, \tag{32}
\end{aligned}$$

where (a) follows from $2ab \leq a^2 + b^2$ for $a, b \in \mathbb{R}$, and (b) from Proposition A.1, with probability at least $1 - \frac{2}{m}$.

For the inequality (30), carefully following the chain rule in (28) we obtain

$$\left(\frac{\partial^2 \alpha^{(l)}}{\partial \alpha^{(l-1)} \partial W^{(l)}}\right)_{i,jj',k} = \frac{1}{m}\phi''(\tilde{\alpha}_i^{(l)})W_{ik}^{(l)}\alpha_{j'}^{(l-1)}\mathbb{1}_{[j=i]} + \frac{1}{\sqrt{m}}\phi'(\tilde{\alpha}_i^{(l)})\mathbb{1}_{[i=j]}\mathbb{1}_{[j'=k]}.$$

Then, we have

$$
\left\| \frac{\partial^2 \alpha^{(l)}}{\partial \alpha^{(l-1)} \partial W^{(l)}} \right\|_{2,2,1}
$$

$$
= \sup_{\|\mathbf{v}_1\|_2 = \|\mathbf{V}_2\|_F = 1} \sum_{i=1}^{m} \left| \sum_{k=1}^{m} \sum_{j=1}^{m} \sum_{j'=1}^{m} \left( \frac{1}{m} \phi''(\tilde{\alpha}_i^{(l)}) W_{ik}^{(l)} \alpha_{j'}^{(l-1)} \mathbb{1}_{[j=i]} \right. \right.
$$

$$
\left. \left. + \frac{1}{\sqrt{m}} \phi'(\tilde{\alpha}_i^{(l)}) \mathbb{1}_{[i=j]} \mathbb{1}_{[j'=k]} \right) \mathbf{v}_{1,k} \mathbf{V}_{2,jj'} \right|
$$

$$
= \sup_{\|\mathbf{v}_1\|_2 = \|\mathbf{V}_2\|_F = 1} \sum_{i=1}^{m} \left| \frac{1}{m} \sum_{j'=1}^{m} \phi''(\tilde{\alpha}_i^{(l)}) \alpha_{j'}^{(l-1)} \mathbf{V}_{2,ij'} \left( \sum_{k=1}^{m} W_{ik}^{(l)} \mathbf{v}_{1,k} \right) \right.
$$

$$
\left. + \frac{1}{\sqrt{m}} \sum_{k=1}^{m} \phi'(\tilde{\alpha}_i^{(l)}) \mathbf{v}_{1,k} \mathbf{V}_{2,ik} \right|
$$

$$
\leq \sup_{\|\mathbf{v}_1\|_2 = \|\mathbf{V}_2\|_F = 1} \frac{1}{m} \beta_\phi \sum_{i=1}^{m} \left| (W^{(l)} \mathbf{v}_1)_i (\mathbf{V}_2 \alpha^{(l-1)})_i \right| + \frac{1}{\sqrt{m}} \sum_{i=1}^{m} \sum_{k=1}^{m} |\mathbf{v}_{1,k} \mathbf{V}_{2,ik}|
$$

$$
\leq \sup_{\|\mathbf{v}_1\|_2 = \|\mathbf{V}_2\|_F = 1} \frac{1}{2m} \beta_\phi \sum_{i=1}^{m} (W^{(l)} \mathbf{v}_1)_i^2 + (\mathbf{V}_2 \alpha^{(l-1)})_i^2 + \frac{1}{\sqrt{m}} \sum_{i=1}^{m} \|\mathbf{v}_1\|_2 \|\mathbf{V}_{2,i,:}\|_2
$$

$$
= \sup_{\|\mathbf{v}_1\|_2 = \|\mathbf{V}_2\|_F = 1} \frac{1}{2m} \beta_\phi \left( \left\| W^{(l)} \mathbf{v}_1 \right\|_2^2 + \left\| \mathbf{V}_2 \alpha^{(l-1)} \right\|_2^2 \right) + \frac{1}{\sqrt{m}} \sum_{i=1}^{m} \|\mathbf{V}_{2,i,:}\|_2
$$

$$
\overset{(a)}{\leq} \frac{1}{2m} \beta_\phi \left( \left\| W^{(l)} \right\|_2^2 + \left\| \alpha^{(l-1)} \right\|_2^2 \right) + \|\mathbf{V}_2\|_F
$$

$$
\overset{(b)}{\leq} \frac{\beta_\phi}{2} \left( \gamma^2 + \left( \gamma^{l-1} + |\phi(0)| \sum_{i=1}^{l-1} \gamma^{i-1} \right)^2 \right) + 1
$$

where (a) follows from $\left\| \mathbf{V}_2 \alpha^{(l-1)} \right\|_2 \leq \|\mathbf{V}_2\|_2 \left\| \alpha^{l-1} \right\| \leq \|\mathbf{V}_2\|_F \left\| \alpha^{l-1} \right\|_2 = \left\| \alpha^{l-1} \right\|_2$ and $\sum_{i=1}^{m} \|\mathbf{V}_{2,i,:}\|_2 \leq \sqrt{m} \sqrt{\sum_{i=1}^{m} \|\mathbf{V}_{2,i,:}\|_2^2}$, and (b) follows from Proposition A.1 and Lemma A.2, with altogether holds with probability at least $1 - \frac{2l}{m}$.

For the last inequality (31), we start with the analysis for $l \geq 2$. Carefully following the chain rule in (28) we obtain

$$
\left( \frac{\partial^2 \alpha^{(l)}}{(\partial W^{(l)})^2} \right)_{i,jj',kk'} = \frac{1}{m} \phi''(\tilde{\alpha}_i^{(l)}) \alpha_{k'}^{(l-1)} \alpha_{j'}^{(l-1)} \mathbb{1}_{[j=i]} \mathbb{1}_{[k=i]}.
$$

Then, we have

$$
\left\| \frac{\partial^2 \alpha^{(l)}}{(\partial W^{(l)})^2} \right\|_{2,2,1}
$$

$$
= \sup_{\|\mathbf{V}_1\|_F = \|\mathbf{V}_2\|_F = 1} \sum_{i=1}^m \left| \sum_{j,j'=1}^m \sum_{k,k'=1}^m \left( \frac{1}{m} \phi''(\tilde{\alpha}_i^{(l)}) \alpha_{k'}^{(l-1)} \alpha_{j'}^{(l-1)} \mathbb{1}_{[j=i]} \mathbb{1}_{[k=i]} \mathbf{V}_{1,jj'} \mathbf{V}_{2,kk'} \right) \right|
$$

$$
= \sup_{\|\mathbf{V}_1\|_F = \|\mathbf{V}_2\|_F = 1} \sum_{i=1}^m \left| \frac{\phi''(\tilde{\alpha}_i^{(l)})}{m} \sum_{j'=1}^m \left( \alpha_{j'}^{(l-1)} \mathbf{V}_{1,ij'} \right) \sum_{k'=1}^m \left( \alpha_{k'}^{(l-1)} \mathbf{V}_{2,ik'} \right) \right|
$$

$$
\leq \sup_{\|\mathbf{V}_1\|_F = \|\mathbf{V}_2\|_F = 1} \frac{1}{m} \beta_\phi \sum_{i=1}^m \left| (\mathbf{V}_1 \alpha^{(l-1)})_i (\mathbf{V}_2 \alpha^{(l-1)})_i \right|
$$

$$
\leq \sup_{\|\mathbf{V}_1\|_F = \|\mathbf{v}_2\|_F = 1} \frac{1}{2m} \beta_\phi \sum_{i=1}^m (\mathbf{V}_2 \alpha^{(l-1)})_i^2 + (\mathbf{V}_2 \alpha^{(l-1)})_i^2
$$

$$
= \sup_{\|\mathbf{V}_1\|_F = \|\mathbf{V}_2\|_F = 1} \frac{1}{2m} \beta_\phi \left( \left\| \mathbf{V}_2 \alpha^{(l-1)} \right\|_2^2 + \left\| \mathbf{V}_2 \alpha^{(l-1)} \right\|_2^2 \right)
$$

$$
\leq \frac{1}{2m} \beta_\phi \left( \left\| \alpha^{(l-1)} \right\|_2^2 + \left\| \alpha^{(l-1)} \right\|_2^2 \right)
$$

$$
\leq \beta_\phi \left( \gamma^{l-1} + |\phi(0)| \sum_{i=1}^{l-1} \gamma^{i-1} \right)^2 ,
$$

which holds with probability at least $1 - \frac{2(l-1)}{m}$. For the case $l = 1$, it is easy to show that $\left( \frac{\partial^2 \alpha^{(1)}}{(\partial W^{(1)})^2} \right)_{i,jj',kk'} = \frac{1}{d} \phi''(\tilde{\alpha}_i^{(1)}) \mathbf{x}_{k'} \mathbf{x}_{j'} \mathbb{1}_{[j=i]} \mathbb{1}_{[k=i]}$ and so $\left\| \frac{\partial^2 \alpha^{(1)}}{(\partial W^{(1)})^2} \right\|_{2,2,1} \leq \beta_\phi$. This completes the proof. $\qquad\square$

## A.5 $L_\infty$ NORM OF $\frac{\partial f}{\partial \alpha^{(l)}}$

Let $\mathbf{b}^{(l)} := \frac{\partial f}{\partial \alpha^{(l)}} \in \mathbb{R}^m$ for any $l \in [L]$. Let $\mathbf{b}_0^{(l)}$ denote $\mathbf{b}^{(l)}$ at initialization. By a direct calculation, we have

$$
\mathbf{b}^{(l)} = \frac{\partial f}{\partial \alpha^{(l)}} = \left( \prod_{l'=l+1}^L \frac{\partial \alpha^{(l)}}{\partial \alpha^{(l-1)}} \right) \frac{\partial f}{\partial \alpha^{(L)}}
$$

$$
= \left( \prod_{l'=l+1}^L \frac{1}{\sqrt{m}} (W^{(l')})^\top D^{(l')} \right) \frac{1}{\sqrt{m}} \mathbf{v} ,
$$

where $D^{(l')}$ is a diagonal matrix of the gradient of activations, i.e., $D_{ii}^{(l')} = \phi'(\tilde{\alpha}_i^{(l')})$. Note that we also have the following recursion:

$$
\mathbf{b}^{(l)} = \frac{\partial f}{\partial \alpha^{(l)}} = \frac{\partial \alpha^{(l+1)}}{\partial \alpha^{(l)}} \frac{\partial f}{\partial \alpha^{(l+1)}}
$$

$$
= \frac{1}{\sqrt{m}} (W^{(l+1)})^\top D^{(l+1)} \mathbf{b}^{(l+1)} .
$$

**Lemma A.6.** *Consider any $l \in [L]$. Under Assumptions 1 and 2, for $\theta \in B_{\rho,\rho_1}^{\mathrm{Spec}}(\theta_0)$, with probability at least $1 - \frac{2(L-l+1)}{m}$,*

$$
\|\mathbf{b}^{(l)}\|_2 \leq \frac{1}{\sqrt{m}} \left( \sigma_1 + \frac{\rho}{\sqrt{m}} \right)^{L-l} (1 + \rho_1) \tag{33}
$$

*and*

$$
\|\mathbf{b}_0^{(l)}\|_2 \leq \frac{\sigma_1^{L-l}}{\sqrt{m}} \leq \frac{\gamma^{L-l}}{\sqrt{m}} . \tag{34}
$$

*Proof.* First, note that $\left\|\mathbf{b}^{(L)}\right\|_2 = \frac{1}{\sqrt{m}}\left\|\mathbf{v}\right\|_2 \leq \frac{1}{\sqrt{m}}(\left\|\mathbf{v}_0\right\|_2 + \left\|\mathbf{v} - \mathbf{v}_0\right\|_2) \leq \frac{1}{\sqrt{m}}(1 + \rho_1)$, where the inequality follows from from Proposition A.1. Now, for the inductive step, assume $\left\|\mathbf{b}^{(l)}\right\|_2 \leq \left(\sigma_1 + \frac{\rho}{\sqrt{m}}\right)^{L-l} \frac{1}{\sqrt{m}}(1 + \rho_1)$ with probability at least $1 - \frac{2l}{m}$. Then,

$$
\begin{aligned}
\left\|\mathbf{b}^{(l-1)}\right\|_2 &= \left\|\frac{\partial\alpha^{(l)}}{\partial\alpha^{(l-1)}}\mathbf{b}^{(l)}\right\|_2 \leq \left\|\frac{\partial\alpha^{(l)}}{\partial\alpha^{(l-1)}}\right\|_2 \left\|\mathbf{b}^{(l)}\right\|_2 \\
&\leq \left(\sigma_1 + \frac{\rho}{\sqrt{m}}\right)\left(\sigma_1 + \frac{\rho}{\sqrt{m}}\right)^{L-l}\frac{1}{\sqrt{m}}(1 + \rho_1) \\
&= \left(\sigma_1 + \frac{\rho}{\sqrt{m}}\right)^{L-l+1}\frac{1}{\sqrt{m}}(1 + \rho_1)
\end{aligned}
$$

where the last inequality follows from Lemma A.3 with probability at least $1 - \frac{2}{m}(l + 1)$. Since we use Proposition A.1 once at layer $L$ and then Lemma A.3 $(L - l)$ times at layer $l$, then we have that everything holds altogether with probability at least $1 - \frac{2}{m}(L - l + 1)$. We have finished the proof by induction. $\qquad\square$

**Lemma A.7.** *Consider any $l \in [L]$. Under Assumptions 1 and 2, for $\theta \in B_{\rho,\rho_1}^{\mathrm{Spec}}(\theta_0)$, with probability at least $1 - \frac{2(L-l)}{m}$,*

$$
\left\|\mathbf{b}^{(l)}\right\|_\infty \leq \frac{\gamma^{L-l}}{\sqrt{m}}(1 + \rho_1). \tag{35}
$$

*Proof.* For any $l \in [L]$, by definition $i$-th component of $\mathbf{b}^{(l)}$, i.e., $\mathbf{b}_i^{(l)}$, takes the form

$$
\mathbf{b}_i^{(l)} = \frac{\partial\alpha^{(L)}}{\partial\alpha_i^{(l)}}\frac{\partial f}{\partial\alpha^{(L)}} = \frac{\partial\alpha^{(L)}}{\partial\alpha_i^{(l)}}\frac{1}{\sqrt{m}}\mathbf{v}.
$$

Then, with $W_{:,i}^{(l)}$ denoting the $i$-th column of the matrix $W^{(l)}$,

$$
\begin{aligned}
\left\|\frac{\partial\alpha^{(L)}}{\partial\alpha_i^{(l)}}\right\|_2 &\overset{(a)}{=} \left\|\frac{\phi'(\tilde{\alpha}_i^{(l)})}{\sqrt{m}}\left(W_{:,i}^{(l)}\right)^\top \prod_{l'=l+2}^{L}\left(\frac{\partial\alpha^{(l')}}{\partial\alpha^{(l'-1)}}\right)\right\|_2 \overset{(b)}{\leq} \frac{1}{\sqrt{m}}\left\|W_{:,i}^{(l)}\right\|_2 \prod_{l'=l+2}^{L}\left\|\frac{\partial\alpha^{(l')}}{\partial\alpha^{(l'-1)}}\right\|_2 \\
&\overset{(c)}{\leq} \frac{1}{\sqrt{m}}\left\|W_{:,i}^{(l)}\right\|_2 \gamma^{L-l-1} \\
&\overset{(d)}{\leq} \gamma\,\gamma^{L-l-1} \\
&= \gamma^{L-l}
\end{aligned}
$$
$$
\tag{36}
$$

where (a) follows from $\frac{\partial\alpha^{(l+1)}}{\partial\alpha_i^{(l)}} = \frac{1}{\sqrt{m}}\phi'(\tilde{\alpha}_i^{(l)})(W_{:,i}^{(l)})^\top$, (b) from $\phi$ being 1-Lipschitz, (c) from Lemma A.3, and (d) from $\left\|W_{:,i}^{(l)}\right\|_2 \leq \left\|W^{(l)}\right\|_2$ and Proposition A.1, which altogether holds with probability $1 - \frac{2}{m}(L - l)$.

Therefore, for every $i \in [m]$,

$$
\begin{aligned}
\left|\mathbf{b}_i^{(l)}\right| &\leq \left|\frac{1}{\sqrt{m}}\frac{\partial\alpha^{(L)}}{\partial\alpha_i^{(l)}}\mathbf{v}\right| \\
&\leq \frac{1}{\sqrt{m}}\left\|\frac{\partial\alpha^{(L)}}{\partial\alpha_i^{(l)}}\right\|_2 \left\|\mathbf{v}\right\|_2 \\
&\leq \frac{1}{\sqrt{m}}\gamma^{L-l}(1 + \rho_1),
\end{aligned}
$$

where the last inequality follows from (36) and $\left\|\mathbf{v}\right\|_2 \leq \left\|\mathbf{v}_0\right\|_2 + \left\|\mathbf{v} - \mathbf{v}_0\right\|_2 \leq 1 + \rho_1$. This completes the proof. $\qquad\square$

A.6 USEFUL BOUNDS

**Lemma 4.1 (Predictor gradient bounds).** *Under Assumptions 1 and 2, for $\theta \in B_{\rho,\rho_1}^{\mathrm{Spec}}(\theta_0)$, with probability at least $\left(1 - \frac{2(L+1)}{m}\right)$, we have*

$$\|\nabla_\theta f(\theta; \mathbf{x})\|_2 \leq \varrho \qquad \text{and} \qquad \|\nabla_\mathbf{x} f(\theta; \mathbf{x})\|_2 \leq \frac{\gamma^L}{\sqrt{m}}(1 + \rho_1) , \qquad (9)$$

*with $\varrho^2 = (h(L+1))^2 + \frac{1}{m}(1 + \rho_1)^2 \sum_{l=1}^L (h(l))^2 \gamma^{2(L-l)}$, $\gamma = \sigma_1 + \frac{\rho}{\sqrt{m}}$, $h(l) = \gamma^{l-1} + |\phi(0)| \sum_{i=1}^{l-1} \gamma^{i-1}$.*

*Proof.* We first prove the bound on the gradient with respect to the weights. Using the chain rule,

$$\frac{\partial f}{\partial \mathbf{w}^{(l)}} = \frac{\partial \alpha^{(l)}}{\partial w^{(l)}} \prod_{l'=l+1}^L \frac{\partial \alpha^{(l')}}{\partial \alpha^{(l'-1)}} \frac{\partial f}{\partial \alpha^{(L)}}$$

and so

$$\left\|\frac{\partial f}{\partial w^{(l)}}\right\|_2^2 \leq \left\|\frac{\partial \alpha^{(l)}}{\partial w^{(l)}}\right\|_2^2 \left\|\prod_{l'=l+1}^L \frac{\partial \alpha^{(l')}}{\partial \alpha^{(l'-1)}} \frac{\partial f}{\partial \alpha^{(L)}}\right\|_2^2 \overset{(a)}{\leq} \left\|\frac{\partial \alpha^{(l)}}{\partial w^{(l)}}\right\|_2^2 \gamma^{2(L-l)} \cdot \frac{1}{m}(1 + \rho_1)^2$$

$$\overset{(b)}{\leq} \left(\gamma^{l-1} + |\phi(0)| \sum_{i=1}^{l-1} \gamma^{i-1}\right)^2 \gamma^{2(L-l)} \cdot \frac{1}{m}(1 + \rho_1)^2$$

for $l \in [L]$, where (a) follows from Lemma A.6, (b) follows from Lemma A.4. Similarly,

$$\left\|\frac{\partial f}{\partial w^{(L+1)}}\right\|_2^2 = \frac{1}{m}\left\|\alpha^{(L)}\right\|_2^2 \leq \left(\gamma^L + |\phi(0)| \sum_{i=1}^L \gamma^{i-1}\right)^2 ,$$

where we used Lemma A.2 for the inequality. Now,

$$\|\nabla_\theta f\|_2^2 = \sum_{l=1}^{L+1} \left\|\frac{\partial f}{\partial w^{(l)}}\right\|_2^2$$

$$\overset{(a)}{\leq} \left(\gamma^L + |\phi(0)| \sum_{i=1}^L \gamma^{i-1}\right)^2 + \frac{1}{m}(1 + \rho_1)^2 \sum_{l=1}^L \left(\gamma^{l-1} + |\phi(0)| \sum_{i=1}^{l-1} \gamma^{i-1}\right)^2 \gamma^{2(L-l)}$$

$$= \varrho^2 ,$$

where (a) follows with probability $1 - \frac{2}{m}(L+1)$ using a union bound from all the previously used results.

We now prove the bound on the gradient with respect to the input data. Using the chain rule,

$$\frac{\partial f}{\partial \mathbf{x}} = \frac{\partial f}{\partial \alpha^{(0)}} = \frac{\partial \alpha^{(1)}}{\partial \alpha^{(0)}} \left(\prod_{l'=2}^L \frac{\partial \alpha^{(l')}}{\partial \alpha^{(l'-1)}}\right) \frac{\partial f}{\partial \alpha^{(L)}}$$

and so

$$\left\|\frac{\partial f}{\partial \mathbf{x}}\right\|_2 \leq \left\|\frac{\partial \alpha^{(1)}}{\partial \alpha^{(0)}}\right\|_2 \left\|\left(\prod_{l'=2}^L \frac{\partial \alpha^{(l')}}{\partial \alpha^{(l'-1)}}\right) \frac{\partial f}{\partial \alpha^{(L)}}\right\|_2$$

$$\leq \left\|\frac{\partial \alpha^{(1)}}{\partial \alpha^{(0)}}\right\|_2 \left(\prod_{l'=2}^L \left\|\frac{\partial \alpha^{(l')}}{\partial \alpha^{(l'-1)}}\right\|_2\right) \left\|\frac{\partial f}{\partial \alpha^{(L)}}\right\|_2$$

$$\overset{(a)}{\leq} \gamma \cdot \gamma^{L-1} \cdot \frac{1}{\sqrt{m}}(1 + \rho_1)$$

$$= \frac{\gamma^L}{\sqrt{m}}(1 + \rho_1)$$

where (a) follows from Lemma A.3 and Lemma A.6 with probability at least $1 - \frac{2L}{m}$ due to union bound. This completes the proof. $\qquad \square$

**Lemma 4.2 (Loss bounds).** *Consider the square loss. Under Assumptions 1, and 2, for $\gamma = \sigma_1 + \frac{\rho}{\sqrt{m}}$, each of the following inequalities hold with probability at least $\left(1 - \frac{2(L+1)}{m}\right)$: $\mathcal{L}(\theta_0) \leq c_{0,\sigma_1}$ and $\mathcal{L}(\theta) \leq c_{\rho_1,\gamma}$ for $\theta \in B_{\rho,\rho_1}^{\mathrm{Spec}}(\theta_0)$, where $c_{a,b} = \frac{2}{n}\sum_{i=1}^{n} y_i^2 + 2(1+a)^2 |g(b)|^2$ and $g(a) = a^L + |\phi(0)| \sum_{i=1}^{L} a^i$ for any $a, b \in \mathbb{R}$.*

*Proof.* We start by noticing that for $\theta \in B_{\rho,\rho_1}^{\mathrm{Spec}}(\theta_0)$,

$$\mathcal{L}(\theta) = \frac{1}{n}\sum_{i=1}^{n}(y_i - f(\theta; \mathbf{x}_i))^2 \leq \frac{1}{n}\sum_{i=1}^{n}(2y_i^2 + 2|f(\theta; \mathbf{x}_i)|^2). \tag{37}$$

Now, let us consider the particular case $\theta = \theta_0$ and a generic $\|\mathbf{x}\|_2 = \sqrt{d}$. Let $\alpha_o^{(l)}$ be the layerwise output of layer $l$ at initialization. Then,

$$
\begin{aligned}
|f(\theta_0; \mathbf{x})| &= \frac{1}{\sqrt{m}} \mathbf{v}_0^\top \alpha_o^{(L)}(\mathbf{x}) \\
&\leq \frac{1}{\sqrt{m}} \|\mathbf{v}_0\|_2 \left\| \alpha_o^{(L)}(\mathbf{x}) \right\|_2 \\
&\stackrel{(a)}{\leq} \frac{1}{\sqrt{m}} \cdot 1 \cdot \left\| \alpha_o^{(L)}(\mathbf{x}) \right\|_2 \\
&\stackrel{(b)}{\leq} \frac{1}{\sqrt{m}} \left( \sigma_1^L + |\phi(0)| \sum_{i=1}^{L} \sigma_1^{i-1} \right) \sqrt{m} \\
&= g(\sigma_1),
\end{aligned}
$$

where (a) follows by assumption and (b) follows from following the same proof as in Lemma A.2 with the difference that we consider the weights at initialization. Now, replacing this result back in (37) we obtain $\mathcal{L}(\theta_0) \leq c_{0,\sigma_1}$.

Now, let us consider the general case of $\theta \in B_{\rho,\rho_1}^{\mathrm{Spec}}(\theta_0)$,

$$
\begin{aligned}
|f(\theta; \mathbf{x})| &= \frac{1}{\sqrt{m}} \mathbf{v}^\top \alpha^{(L)}(\mathbf{x}) \\
&\leq \frac{1}{\sqrt{m}} \|\mathbf{v}\|_2 \left\| \alpha^{(L)}(\mathbf{x}) \right\|_2 \\
&\stackrel{(a)}{\leq} \frac{1}{\sqrt{m}} (1 + \rho_1) \left\| \alpha^{(L)}(\mathbf{x}) \right\|_2 \\
&\stackrel{(b)}{\leq} \frac{1}{\sqrt{m}} (1 + \rho_1) \left( \gamma^L + |\phi(0)| \sum_{i=1}^{L} \gamma^{i-1} \right) \sqrt{m} \\
&= (1 + \rho_1) g(\gamma),
\end{aligned}
$$

where (a) follows from $\|v\|_2 \leq \|\mathbf{v}_0\|_2 + \|\mathbf{v} - \mathbf{v}_0\|_2 \leq 1 + \rho_1$, and (b) follows from Lemma A.2. Now, replacing this result back in (37) we obtain $\mathcal{L}(\theta_0) \leq c_{\rho_1,\gamma}$.

In either case, a union bound let us obtain the probability with which the results hold. This finishes the proof. $\qquad\square$

**Corollary 4.1 (Loss gradient bound).** *Consider the square loss. Under Assumptions 1 and 2, for $\theta \in B_{\rho,\rho_1}^{\mathrm{Spec}}(\theta_0)$, with probability at least $\left(1 - \frac{2(L+1)}{m}\right)$, we have $\|\nabla_\theta \mathcal{L}(\theta)\|_2 \leq 2\sqrt{\mathcal{L}(\theta)}\varrho \leq 2\sqrt{c_{\rho_1,\gamma}}\varrho$, with $\varrho$ as in Lemma 4.1 and $c_{\rho_1,\gamma}$ as in Lemma 4.2.*

*Proof.* We have that $\|\nabla_\theta \mathcal{L}(\theta)\|_2 = \left\| \frac{1}{n}\sum_{i=1}^{n} \ell_i' \nabla_\theta f \right\|_2 \leq \frac{1}{n}\sum_{i=1}^{n} |\ell_i'| \|\nabla_\theta f\|_2 \stackrel{(a)}{\leq} \frac{2\varrho}{n}\sum_{i=1}^{n} |y_i - \hat{y}_i| \leq 2\varrho\sqrt{\mathcal{L}(\theta)} \stackrel{(b)}{\leq} 2\sqrt{c_{\rho_1,\gamma}}\varrho$ where (a) follows from Lemma 4.1 and (b) from Lemma 4.2. $\qquad\square$

### A.7 REGARDING THE NETWORK ARCHITECTURES IN OUR WORK AND LIU ET AL. (2020)'S

A difference between the neural network used in our work and the one in (Liu et al., 2020) is that we normalize the norm of the last layer's weight at initialization, whereas (Liu et al., 2020) does not. However, if we did not normalize our last layer, then our result on the Hessian spectral norm bound would still hold with $\widetilde{O}$ instead of $O$; consequently, our comparison with (Liu et al., 2020) on the dependence on the network's depth $L$ (our polynomial dependence against their exponential dependence) would still hold as stated in Remark 4.2 .

## B RESTRICTED STRONG CONVEXITY

We establish the results from Section 5 in this appendix.

### B.1 RESTRICTED STRONG CONVEXITY AND SMOOTHNESS

**Theorem 5.1** (**RSC for Square Loss**). *For square loss, under Assumptions 1 and 2, with probability at least* $(1 - \frac{2(L+1)}{m})$, $\forall \theta' \in Q_\kappa^t \cap B_{\rho,\rho_1}^{\mathrm{Spec}}(\theta_0) \cap B_{\rho_2}^{\mathrm{Euc}}(\theta_t)$ *with* $\theta_t \in B_{\rho,\rho_1}^{\mathrm{Spec}}(\theta_0)$,

$$\mathcal{L}(\theta') \geq \mathcal{L}(\theta_t) + \langle \theta' - \theta_t, \nabla_\theta \mathcal{L}(\theta_t) \rangle + \frac{\alpha_t}{2} \|\theta' - \theta_t\|_2^2 , \quad with \quad \alpha_t = c_1 \|\bar{\mathbf{g}}_t\|_2^2 - \frac{c_2}{\sqrt{m}} , \tag{10}$$

*where* $\bar{\mathbf{g}}_t = \frac{1}{n} \sum_{i=1}^n \nabla_\theta f(\theta_t; \mathbf{x}_i)$, $c_1 = 2\kappa^2$ *and* $c_2 = 2c_H(2\varrho\rho_2 + \sqrt{c_{\rho_1,\gamma}})$, *with* $c_H$ *as in Theorem 4.1,* $\varrho$ *as in Lemma 4.1, and* $c_{\rho_1,\gamma}$ *as in Lemma 4.2. Consequently,* $\mathcal{L}$ *satisfies RSC w.r.t.* $(Q_\kappa^t \cap B_{\rho,\rho_1}^{\mathrm{Spec}}(\theta_0) \cap B_{\rho_2}^{\mathrm{Euc}}(\theta_t), \theta_t)$ *whenever* $\alpha_t > 0$.

*Proof.* For any $\theta' \in Q_{\kappa/2}^t \cap B_{\rho,\rho_1}^{\mathrm{Euc}}(\theta_0)$, by the second order Taylor expansion around $\theta_t$, we have

$$\mathcal{L}(\theta') = \mathcal{L}(\theta_t) + \langle \theta' - \theta_t, \nabla_\theta \mathcal{L}(\theta_t) \rangle + \frac{1}{2}(\theta' - \theta_t)^\top \frac{\partial^2 \mathcal{L}(\tilde{\theta}_t)}{\partial \theta^2}(\theta' - \theta_t) ,$$

where $\tilde{\theta}_t = \xi\theta' + (1 - \xi)\theta_t$ for some $\xi \in [0, 1]$. We note that $\tilde{\theta}_t \in B_{\rho,\rho_1}^{\mathrm{Spec}}(\theta_0)$ since,

- $\left\| \tilde{W}_t^{(l)} - W_0^{(l)} \right\|_2 = \left\| \xi W'^{(l)} - \xi W_0^{(l)} + (1 - \xi)W_t^{(l)} - (1 - \xi)W_0^{(l)} \right\|_2 \leq$ $\xi \left\| W'^{(l)} - W_0^{(l)} \right\|_2 + (1 - \xi) \left\| W_t^{(l)} - W_0^{(l)} \right\|_2 \leq \rho$, for any $l \in [L]$, where the last inequality follows from our assumption $\theta', \theta_t \in B_{\rho,\rho_1}^{\mathrm{Spec}}(\theta_0)$; and

- $\left\| \tilde{W}_t^{(L+1)} - W_0^{(L+1)} \right\|_2 = \|\tilde{\mathbf{v}} - \mathbf{v}_0\|_2 \leq \rho_1$, by following a similar derivation as in the previous point.

Focusing on the quadratic form in the Taylor expansion and recalling the form of the Hessian, we get

$$(\theta' - \theta_t)^\top \frac{\partial^2 \mathcal{L}(\tilde{\theta}_t)}{\partial \theta^2}(\theta' - \theta_t)$$

$$= (\theta' - \theta_t)^\top \frac{1}{n} \sum_{i=1}^n \left[ \ell_i'' \frac{\partial f(\tilde{\theta}_t; \mathbf{x}_i)}{\partial \theta} \frac{\partial f(\tilde{\theta}_t; \mathbf{x}_i)}{\partial \theta}^\top + \ell_i' \frac{\partial^2 f(\tilde{\theta}_t; \mathbf{x}_i)}{\partial \theta^2} \right] (\theta' - \theta_t)$$

$$= \underbrace{\frac{1}{n} \sum_{i=1}^n \ell_i'' \left\langle \theta' - \theta_t, \frac{\partial f(\tilde{\theta}_t; \mathbf{x}_i)}{\partial \theta} \right\rangle^2}_{I_1} + \underbrace{\frac{1}{n} \sum_{i=1}^n \ell_i'(\theta' - \theta_t)^\top \frac{\partial^2 f(\tilde{\theta}_t; \mathbf{x}_i)}{\partial \theta^2}(\theta' - \theta_t)}_{I_2} ,$$

where $\ell_i = \ell(y_i, f(\tilde{\theta}_t, \mathbf{x}_i))$, $\ell_i' = \frac{\partial \ell(y_i, z)}{\partial z}\Big|_{z=f(\tilde{\theta}_t, \mathbf{x}_i))}$, and $\ell_i'' = \frac{\partial^2 \ell(y_i, z)}{\partial z^2}\Big|_{z=f(\tilde{\theta}_t, \mathbf{x}_i))}$. Now, note that

$$
I_1 = \frac{1}{n} \sum_{i=1}^n \ell_i'' \left\langle \theta' - \theta_t, \frac{\partial f(\tilde{\theta}_t; \mathbf{x}_i)}{\partial \theta} \right\rangle^2
$$

$$
\geq \frac{2}{n} \sum_{i=1}^n \left\langle \theta' - \theta_t, \frac{\partial f(\theta_t; \mathbf{x}_i)}{\partial \theta} + \left( \frac{\partial f(\tilde{\theta}_t; \mathbf{x}_i)}{\partial \theta} - \frac{\partial f(\theta_t; \mathbf{x}_i)}{\partial \theta} \right) \right\rangle^2
$$

$$
= \frac{2}{n} \sum_{i=1}^n \left\langle \theta' - \theta_t, \frac{\partial f(\theta_t; \mathbf{x}_i)}{\partial \theta} \right\rangle^2 + \frac{2}{n} \sum_{i=1}^n \left\langle \theta' - \theta_t, \frac{\partial f(\tilde{\theta}_t; \mathbf{x}_i)}{\partial \theta} - \frac{\partial f(\theta_t; \mathbf{x}_i)}{\partial \theta} \right\rangle^2
$$

$$
+ \frac{4}{n} \sum_{i=1}^n \left\langle \theta' - \theta_t, \frac{\partial f(\theta_t; \mathbf{x}_i)}{\partial \theta} \right\rangle \left\langle \theta' - \theta_t, \frac{\partial f(\tilde{\theta}_t; \mathbf{x}_i)}{\partial \theta} - \frac{\partial f(\theta_t; \mathbf{x}_i)}{\partial \theta} \right\rangle
$$

$$
\overset{(a)}{\geq} \frac{2}{n} \sum_{i=1}^n \left\langle \theta' - \theta_t, \frac{\partial f(\theta_t; \mathbf{x}_i)}{\partial \theta} \right\rangle^2 - \frac{4}{n} \sum_{i=1}^n \left\| \frac{\partial f(\theta_t; \mathbf{x}_i)}{\partial \theta} \right\|_2 \left\| \frac{\partial f(\tilde{\theta}_t; \mathbf{x}_i)}{\partial \theta} - \frac{\partial f(\theta_t; \mathbf{x}_i)}{\partial \theta} \right\|_2
$$

$$
\times \|\theta' - \theta_t\|_2^2
$$

$$
\overset{(b)}{\geq} 2 \left\langle \theta' - \theta_t, \frac{1}{n} \sum_{i=1}^n \frac{\partial f(\theta_t; \mathbf{x}_i)}{\partial \theta} \right\rangle^2 - \frac{4}{n} \sum_{i=1}^n \varrho \frac{c_H}{\sqrt{m}} \|\tilde{\theta}_t - \theta_t\|_2 \|\theta' - \theta_t\|_2^2
$$

$$
\overset{(c)}{\geq} 2 \left\langle \theta' - \theta_t, \frac{1}{n} \sum_{i=1}^n \frac{\partial f(\theta_t; \mathbf{x}_i)}{\partial \theta} \right\rangle^2 - \frac{4 \varrho c_H}{\sqrt{m}} \|\theta' - \theta_t\|_2^3
$$

$$
\overset{(d)}{\geq} 2\kappa^2 \left\| \frac{1}{n} \sum_{i=1}^n \frac{\partial f(\theta_t; \mathbf{x}_i)}{\partial \theta} \right\|_2^2 \|\theta' - \theta_t\|_2^2 - \frac{4 \varrho c_H}{\sqrt{m}} \|\theta' - \theta_t\|_2^3
$$

$$
= \left( 2\kappa^2 \left\| \frac{1}{n} \sum_{i=1}^n \frac{\partial f(\theta_t; \mathbf{x}_i)}{\partial \theta} \right\|_2^2 - \frac{4 \varrho c_H \|\theta' - \theta_t\|_2}{\sqrt{m}} \right) \|\theta' - \theta_t\|_2^2 ,
$$

where (a) follows by Cauchy-Schwartz inequality; (b) follows by Jensen's inequality (first term) and the use of Theorem 4.1 and Lemma 4.1 due to $\bar{\theta}_t \in B_{\rho, \rho_1}^{\text{Spec}}(\theta_0)$; (c) follows from $\left\| \tilde{\theta}_t - \theta_t \right\|_2 = \|\xi \theta' + (1 - \xi)\theta_t - \theta_t\|_2 = \xi \|\theta' - \theta_t\| \leq \|\theta' - \theta_t\|_2$; (d) follows since $\theta' \in Q_\kappa^t$ and from the fact that $p^\top q = \cos(p, q) \|p\| \|q\|$ for any vectors $p, q$.

For analyzing $I_2$, first note that for square loss, $\ell_{i,t}' = 2(\tilde{y}_{i,t} - y_i)$ with $\tilde{y}_{i,t} = f(\tilde{\theta}_t; \mathbf{x}_i)$, so that for the vector $[\ell_{i,t}']_i$, we have $\frac{1}{n} \|[\ell_{i,t}']_i\|_2^2 = \frac{4}{n} \sum_{i=1}^n (\tilde{y}_{i,t} - y_i)^2 = 4\mathcal{L}(\theta_t)$. Further, with $Q_{t,i} = (\theta' - \theta_t)^\top \frac{\partial^2 f(\tilde{\theta}_t; \mathbf{x}_i)}{\partial \theta^2} (\theta' - \theta_t)$, we have

$$
|Q_{t,i}| = \left| (\theta' - \theta_t)^\top \frac{\partial^2 f(\tilde{\theta}_t; \mathbf{x}_i)}{\partial \theta^2} (\theta' - \theta_t) \right| \leq \|\theta' - \theta_t\|_2^2 \left\| \frac{\partial^2 f(\tilde{\theta}_t; \mathbf{x}_i)}{\partial \theta^2} \right\|_2 \leq \frac{c_H \|\theta' - \theta_t\|_2^2}{\sqrt{m}} .
$$

Now, note that

$$
I_2 = \frac{1}{n} \sum_{i=1}^n \ell_{i,t}' Q_{t,i}
$$

$$
\geq -\left| \sum_{i=1}^n \left( \frac{1}{\sqrt{n}} \ell_{i,t}' \right) \left( \frac{1}{\sqrt{n}} Q_{t,i} \right) \right|
$$

$$
\overset{(a)}{\geq} -\left( \frac{1}{n} \|[\ell_{i,t}']\|_2^2 \right)^{1/2} \left( \frac{1}{n} \sum_{i=1}^n Q_{t,i}^2 \right)^{1/2}
$$

$$
\geq -2\sqrt{\mathcal{L}(\tilde{\theta}_t)} \frac{c_H \|\theta' - \theta_t\|_2^2}{\sqrt{m}} ,
$$

where (a) follows by Cauchy-Schwartz inequality. Putting the lower bounds on $I_1$ and $I_2$ back, with $\bar{\mathbf{g}}_t = \frac{1}{n} \sum_{i=1}^{n} \frac{\partial f(\theta_t; \mathbf{x}_i)}{\partial \theta}$, we have

$$(\theta' - \theta_t)^\top \frac{\partial^2 \mathcal{L}(\tilde{\theta}_t)}{\partial \theta^2}(\theta' - \theta_t) \geq \left( 2\kappa^2 \|\bar{\mathbf{g}}_t\|_2^2 - \frac{4\varrho c_H \|\theta' - \theta_t\|_2 + 2c_H \sqrt{\mathcal{L}(\tilde{\theta}_t)}}{\sqrt{m}} \right) \|\theta' - \theta_t\|_2^2 .$$

Now, since $\theta' \in B_{\rho_2}^{\mathrm{Euc}}(\theta_t)$, $\|\theta' - \theta_t\|_2 \leq \rho_2$, so we have

$$(\theta' - \theta_t)^\top \frac{\partial^2 \mathcal{L}(\tilde{\theta}_t)}{\partial \theta^2}(\theta' - \theta_t) \geq \left( 2\kappa^2 \|\bar{\mathbf{g}}_t\|_2^2 - \frac{4\varrho c_H \rho_2 + 2c_H \sqrt{\mathcal{L}(\tilde{\theta}_t)}}{\sqrt{m}} \right) \|\theta' - \theta_t\|_2^2$$

$$\geq \left( 2\kappa^2 \|\bar{\mathbf{g}}_t\|_2^2 - \frac{4\varrho c_H \rho_2 + 2c_H \sqrt{c_{\rho_1, \gamma}}}{\sqrt{m}} \right) \|\theta' - \theta_t\|_2^2 ,$$

where the last inequality follows from Lemma 4.2. That completes the proof. $\qquad \square$

Next, we state and prove the RSC result for general losses.

**Theorem B.1** (RSC of Loss). *Under Assumptions 1, 2 and 3, with probability at least* $(1 - \frac{2(L+1)}{m})$, $\forall \theta' \in Q_\kappa^t \cap B_{\rho, \rho_1}^{\mathrm{Spec}}(\theta_0) \cap B_{\rho_2}^{\mathrm{Euc}}(\theta_t)$,

$$\mathcal{L}(\theta') \geq \mathcal{L}(\theta_t) + \langle \theta' - \theta_t, \nabla_\theta \mathcal{L}(\theta_t) \rangle + \frac{\alpha_t}{2} \|\theta' - \theta_t\|_2^2 , \quad \text{with} \quad \alpha_t = c_1 \|\bar{\mathbf{g}}_t\|_2^2 - \frac{c_4 + c_{4,t}}{\sqrt{m}} , \quad (38)$$

*where* $\bar{\mathbf{g}}_t = \frac{1}{n} \sum_{i=1}^{n} \nabla_\theta f(\theta_t; \mathbf{x}_i)$, $c_1 = a\kappa^2$, $c_4 = 2a\varrho c_H \rho_2$, $c_H$ *is as in Theorem 4.1, and* $c_{4,t} = c_H \sqrt{\lambda_t}$ *where* $\lambda_t = \frac{1}{n} \sum_{i=1}^{n} (\ell'_{i,t})^2$ *with* $\ell'_i = \frac{\partial \ell(y_i, z)}{\partial z} \Big|_{z = f(\tilde{\theta}_t, \mathbf{x}_i))}$ *and* $\tilde{\theta} \in B_{\rho, \rho_1}^{\mathrm{Spec}}$ *being some point in the segment that joins* $\theta'$ *and* $\theta_t$. *Consequently,* $\mathcal{L}$ *satisfies RSC w.r.t.* $(Q_\kappa^t \cap B_\rho^{\mathrm{Spec}}(\theta_0) \cap B_{\rho_2}^{\mathrm{Euc}}(\theta_t), \theta_t)$ *whenever* $\alpha_t > 0$.

*Proof.* For any $\theta' \in Q_{\kappa/2}^t \cap B_{\rho, \rho_1}^{\mathrm{Euc}}(\theta_0)$, by the second order Taylor expansion around $\theta_t$, we have

$$\mathcal{L}(\theta') = \mathcal{L}(\theta_t) + \langle \theta' - \theta_t, \nabla_\theta \mathcal{L}(\theta_t) \rangle + \frac{1}{2}(\theta' - \theta_t)^\top \frac{\partial^2 \mathcal{L}(\tilde{\theta}_t)}{\partial \theta^2}(\theta' - \theta_t) ,$$

where $\tilde{\theta}_t = \xi \theta' + (1 - \xi)\theta_t$ for some $\xi \in [0, 1]$. We note that $\tilde{\theta}_t \in B_{\rho, \rho_1}^{\mathrm{Spec}}(\theta_0)$ since,

- $\left\| \tilde{W}_t^{(l)} - W_0^{(l)} \right\|_2 = \left\| \xi W'^{(l)} - \xi W_0^{(l)} + (1 - \xi)W_t^{(l)} - (1 - \xi)W_0^{(l)} \right\|_2 \leq \xi \left\| W'^{(l)} - W_0^{(l)} \right\|_2 + (1 - \xi) \left\| W_t^{(l)} - W_0^{(l)} \right\|_2 \leq \rho$, for any $l \in [L]$, where the last inequality follows from our assumption $\theta', \theta_t \in B_{\rho, \rho_1}^{\mathrm{Spec}}(\theta_0)$; and

- $\left\| \tilde{W}_t^{(L+1)} - W_0^{(L+1)} \right\|_2 = \|\tilde{\mathbf{v}} - \mathbf{v}_0\|_2 \leq \rho_1$, by following a similar derivation as in the previous point.

Focusing on the quadratic form in the Taylor expansion and recalling the form of the Hessian, we get

$$(\theta' - \theta_t)^\top \frac{\partial^2 \mathcal{L}(\tilde{\theta}_t)}{\partial \theta^2}(\theta' - \theta_t)$$

$$= (\theta' - \theta_t)^\top \frac{1}{n} \sum_{i=1}^{n} \left[ \ell''_i \frac{\partial f(\tilde{\theta}_t; \mathbf{x}_i)}{\partial \theta} \frac{\partial f(\tilde{\theta}_t; \mathbf{x}_i)}{\partial \theta}^\top + \ell'_i \frac{\partial^2 f(\tilde{\theta}_t; \mathbf{x}_i)}{\partial \theta^2} \right] (\theta' - \theta_t)$$

$$= \underbrace{\frac{1}{n} \sum_{i=1}^{n} \ell''_i \left\langle \theta' - \theta_t, \frac{\partial f(\tilde{\theta}_t; \mathbf{x}_i)}{\partial \theta} \right\rangle^2}_{I_1} + \underbrace{\frac{1}{n} \sum_{i=1}^{n} \ell'_i (\theta' - \theta_t)^\top \frac{\partial^2 f(\tilde{\theta}_t; \mathbf{x}_i)}{\partial \theta^2}(\theta' - \theta_t)}_{I_2} ,$$

where $\ell_i = \ell(y_i, f(\tilde{\theta}_t, \mathbf{x}_i))$, $\ell_i' = \frac{\partial \ell(y_i, z)}{\partial z}\Big|_{z=f(\tilde{\theta}_t, \mathbf{x}_i))}$, and $\ell_i'' = \frac{\partial^2 \ell(y_i, z)}{\partial z^2}\Big|_{z=f(\tilde{\theta}_t, \mathbf{x}_i))}$. Now, note that

$$I_1 = \frac{1}{n} \sum_{i=1}^n \ell_i'' \left\langle \theta' - \theta_t, \frac{\partial f(\tilde{\theta}_t; \mathbf{x}_i)}{\partial \theta} \right\rangle^2$$

$$\geq \frac{2}{n} \sum_{i=1}^n \left\langle \theta' - \theta_t, \frac{\partial f(\theta_t; \mathbf{x}_i)}{\partial \theta} + \left( \frac{\partial f(\tilde{\theta}_t; \mathbf{x}_i)}{\partial \theta} - \frac{\partial f(\theta_t; \mathbf{x}_i)}{\partial \theta} \right) \right\rangle^2$$

$$= \frac{2}{n} \sum_{i=1}^n \left\langle \theta' - \theta_t, \frac{\partial f(\theta_t; \mathbf{x}_i)}{\partial \theta} \right\rangle^2 + \frac{2}{n} \sum_{i=1}^n \left\langle \theta' - \theta_t, \frac{\partial f(\tilde{\theta}_t; \mathbf{x}_i)}{\partial \theta} - \frac{\partial f(\theta_t; \mathbf{x}_i)}{\partial \theta} \right\rangle^2$$

$$\quad + \frac{4}{n} \sum_{i=1}^n \left\langle \theta' - \theta_t, \frac{\partial f(\theta_t; \mathbf{x}_i)}{\partial \theta} \right\rangle \left\langle \theta' - \theta_t, \frac{\partial f(\tilde{\theta}_t; \mathbf{x}_i)}{\partial \theta} - \frac{\partial f(\theta_t; \mathbf{x}_i)}{\partial \theta} \right\rangle$$

$$\overset{(a)}{\geq} \frac{2}{n} \sum_{i=1}^n \left\langle \theta' - \theta_t, \frac{\partial f(\theta_t; \mathbf{x}_i)}{\partial \theta} \right\rangle^2$$

$$\quad - \frac{4}{n} \sum_{i=1}^n \left\| \frac{\partial f(\theta_t; \mathbf{x}_i)}{\partial \theta} \right\|_2 \left\| \frac{\partial f(\tilde{\theta}_t; \mathbf{x}_i)}{\partial \theta} - \frac{\partial f(\theta_t; \mathbf{x}_i)}{\partial \theta} \right\|_2 \|\theta' - \theta_t\|_2^2$$

$$\overset{(b)}{\geq} a \left\langle \theta' - \theta_t, \frac{1}{n} \sum_{i=1}^n \frac{\partial f(\theta_t; \mathbf{x}_i)}{\partial \theta} \right\rangle^2 - \frac{4}{n} \sum_{i=1}^n \varrho \frac{c_H}{\sqrt{m}} \|\tilde{\theta}_t - \theta_t\|_2 \|\theta' - \theta_t\|_2^2$$

$$\overset{(c)}{\geq} a \left\langle \theta' - \theta_t, \frac{1}{n} \sum_{i=1}^n \frac{\partial f(\theta_t; \mathbf{x}_i)}{\partial \theta} \right\rangle^2 - \frac{2a\varrho c_H}{\sqrt{m}} \|\theta' - \theta_t\|_2^3$$

$$\overset{(d)}{\geq} a\kappa^2 \left\| \frac{1}{n} \sum_{i=1}^n \frac{\partial f(\theta_t; \mathbf{x}_i)}{\partial \theta} \right\|_2^2 \|\theta' - \theta_t\|_2^2 - \frac{2a\varrho c_H}{\sqrt{m}} \|\theta' - \theta_t\|_2^3$$

$$= \left( a\kappa^2 \left\| \frac{1}{n} \sum_{i=1}^n \frac{\partial f(\theta_t; \mathbf{x}_i)}{\partial \theta} \right\|_2^2 - \frac{2a\varrho c_H \|\theta' - \theta_t\|_2}{\sqrt{m}} \right) \|\theta' - \theta_t\|_2^2,$$

where (a) follows by Cauchy-Schwartz inequality; (b) follows by Jensen's inequality (first term) and the use of Theorem 4.1 and Lemma 4.1 due to $\tilde{\theta}_t \in B_{\rho,\rho_1}^{\text{Spec}}(\theta_0)$; (c) follows from $\left\| \tilde{\theta}_t - \theta_t \right\|_2 = \|\xi\theta' + (1-\xi)\theta_t - \theta_t\|_2 = \xi \|\theta' - \theta_t\| \leq \|\theta' - \theta_t\|_2$; (d) follows since $\theta' \in Q_\kappa^t$ and from the fact that $p^\top q = \cos(p, q) \|p\| \|q\|$ for any vectors $p, q$.

For analyzing $I_2$, let $\lambda_t := \frac{1}{n} \sum_{i=1}^n (\ell_{i,t}')^2$. As before, with $Q_{t,i} = (\theta' - \theta_t)^\top \frac{\partial^2 f(\tilde{\theta}_t; \mathbf{x}_i)}{\partial \theta^2} (\theta' - \theta_t)$, we have

$$|Q_{t,i}| = \left| (\theta' - \theta_t)^\top \frac{\partial^2 f(\tilde{\theta}_t; \mathbf{x}_i)}{\partial \theta^2} (\theta' - \theta_t) \right| \leq \|\theta' - \theta_t\|_2^2 \left\| \frac{\partial^2 f(\tilde{\theta}_t; \mathbf{x}_i)}{\partial \theta^2} \right\|_2 \leq \frac{c_H \|\theta' - \theta_t\|_2^2}{\sqrt{m}}.$$

Now, note that

$$I_2 = \frac{1}{n} \sum_{i=1}^n \ell_{i,t}' Q_{t,i}$$

$$\geq - \left| \sum_{i=1}^n \left( \frac{1}{\sqrt{n}} \ell_{i,t}' \right) \left( \frac{1}{\sqrt{n}} Q_{t,i} \right) \right|$$

$$\overset{(a)}{\geq} - \left( \frac{1}{n} \|[\ell_{i,t}']\|_2^2 \right)^{1/2} \left( \frac{1}{n} \sum_{i=1}^n Q_{t,i}^2 \right)^{1/2}$$

$$\geq - \sqrt{\lambda_t} \frac{c_H \|\theta' - \theta_t\|_2^2}{\sqrt{m}},$$

where (a) follows by Cauchy-Schwartz inequality. Putting the lower bounds on $I_1$ and $I_2$ back, with $\bar{\mathbf{g}}_t = \frac{1}{n} \sum_{i=1}^{n} \frac{\partial f(\theta_t; \mathbf{x}_i)}{\partial \theta}$, we have

$$(\theta' - \theta_t)^\top \frac{\partial^2 \mathcal{L}(\tilde{\theta}_t)}{\partial \theta^2} (\theta' - \theta_t) \geq \left( a\kappa^2 \|\bar{\mathbf{g}}_t\|_2^2 - \frac{2a\varrho c_H \|\theta' - \theta_t\|_2 + c_H \sqrt{\lambda_t}}{\sqrt{m}} \right) \|\theta' - \theta_t\|_2^2 \, .$$

Now, since $\theta' \in B_{\rho_2}^{\mathrm{Euc}}(\theta_t)$, $\|\theta' - \theta_t\|_2 \leq \rho_2$, so we have

$$(\theta' - \theta_t)^\top \frac{\partial^2 \mathcal{L}(\tilde{\theta}_t)}{\partial \theta^2} (\theta' - \theta_t) \geq \left( a\kappa^2 \|\bar{\mathbf{g}}_t\|_2^2 - \frac{2a\varrho c_H \rho_2 + c_H \sqrt{\lambda_t}}{\sqrt{m}} \right) \|\theta' - \theta_t\|_2^2 \, .$$

That completes the proof. $\qquad\qquad\qquad\qquad\qquad\qquad\qquad\qquad\qquad\qquad\qquad\qquad\square$

**Theorem 5.2** (**Local Smoothness for Square Loss**). *For square loss, under Assumptions 1 and 2, with probability at least* $(1 - \frac{2(L+1)}{m})$, $\forall \theta, \theta' \in B_{\rho,\rho_1}^{\mathrm{Spec}}(\theta_0)$,

$$\mathcal{L}(\theta') \leq \mathcal{L}(\theta) + \langle \theta' - \theta, \nabla_\theta \mathcal{L}(\theta) \rangle + \frac{\beta}{2} \|\theta' - \theta\|_2^2 \, , \quad \text{with} \quad \beta = 2\varrho^2 + \frac{2c_H \sqrt{c_{\rho_1,\gamma}}}{\sqrt{m}} \, , \qquad (11)$$

*with $c_H$ as in Theorem 4.1, $\varrho$ as in Lemma 4.1, and $c_{\rho_1,\gamma}$ as in Lemma 4.2. Consequently, $\mathcal{L}$ is locally $\beta$-smooth. Moreover, if $\gamma$ (and so $\sigma_1$ and $\rho$) is chosen according to the desirable operating regimes (see Remark 4.1) and $\rho_1 = O(\mathrm{poly}(L))$, then $\beta = O(\mathrm{poly}(L))$.*

*Proof.* By the second order Taylor expansion about $\bar{\theta} \in B_{\rho,\rho_1}^{\mathrm{Spec}}(\theta_0)$, we have $\mathcal{L}(\theta') = \mathcal{L}(\bar{\theta}) + \langle \theta' - \bar{\theta}, \nabla_\theta \mathcal{L}(\bar{\theta}) \rangle + \frac{1}{2} (\theta' - \bar{\theta})^\top \frac{\partial^2 \mathcal{L}(\tilde{\theta})}{\partial \theta^2} (\theta' - \bar{\theta})$, where $\tilde{\theta} = \xi \theta' + (1 - \xi)\bar{\theta}$ for some $\xi \in [0,1]$. Then,

$$(\theta' - \bar{\theta})^\top \frac{\partial^2 \mathcal{L}(\tilde{\theta})}{\partial \theta^2} (\theta' - \bar{\theta})$$

$$= (\theta' - \bar{\theta})^\top \frac{1}{n} \sum_{i=1}^{n} \left[ \ell_i'' \frac{\partial f(\tilde{\theta}; \mathbf{x}_i)}{\partial \theta} \frac{\partial f(\tilde{\theta}; \mathbf{x}_i)}{\partial \theta}^\top + \ell_i' \frac{\partial^2 f(\tilde{\theta}; \mathbf{x}_i)}{\partial \theta^2} \right] (\theta' - \bar{\theta})$$

$$= \underbrace{\frac{1}{n} \sum_{i=1}^{n} \ell_i'' \left\langle \theta' - \bar{\theta}, \frac{\partial f(\tilde{\theta}; \mathbf{x}_i)}{\partial \theta} \right\rangle^2}_{I_1} + \underbrace{\frac{1}{n} \sum_{i=1}^{n} \ell_i' (\theta' - \bar{\theta})^\top \frac{\partial^2 f(\tilde{\theta}; \mathbf{x}_i)}{\partial \theta^2} (\theta' - \bar{\theta})}_{I_2} \, ,$$

where $\ell_i = \ell(y_i, f(\tilde{\theta}, \mathbf{x}_i))$, $\ell_i' = \left. \frac{\partial \ell(y_i, z)}{\partial z} \right|_{z = f(\tilde{\theta}, \mathbf{x}_i))}$, and $\ell_i'' = \left. \frac{\partial^2 \ell(y_i, z)}{\partial z^2} \right|_{z = f(\tilde{\theta}, \mathbf{x}_i))}$. Now, note that

$$I_1 = \frac{1}{n} \sum_{i=1}^{n} \ell_i'' \left\langle \theta' - \bar{\theta}, \frac{\partial f(\tilde{\theta}; \mathbf{x}_i)}{\partial \theta} \right\rangle^2$$

$$\overset{(a)}{\leq} \frac{2}{n} \sum_{i=1}^{n} \left\| \frac{\partial f(\tilde{\theta}; \mathbf{x}_i)}{\partial \theta} \right\|_2^2 \|\theta' - \bar{\theta}\|_2^2$$

$$\overset{(b)}{\leq} 2\varrho^2 \|\theta' - \bar{\theta}\|_2^2 \, ,$$

where (a) follows by the Cauchy-Schwartz inequality and (b) from Lemma 4.1.

For $I_2$, first note that for square loss, $\ell_i' = 2(\tilde{y}_i - y_i)$ with $\tilde{y}_i = f(\tilde{\theta}; \mathbf{x}_i)$, so that for the vector $[\ell_i']_i$, we have $\frac{1}{n} \|[\ell_i']_i\|_2^2 = \frac{4}{n} \sum_{i=1}^{n} (\tilde{y}_i - y_i)^2 = 4\mathcal{L}(\tilde{\theta})$. Further, with

$$Q_i = (\theta' - \bar{\theta})^\top \frac{\partial^2 f(\tilde{\theta}_t; \mathbf{x}_i)}{\partial \theta^2} (\theta' - \bar{\theta}),$$

we have

$$|Q_i| = \left| (\theta' - \bar{\theta})^\top \frac{\partial^2 f(\tilde{\theta}; \mathbf{x}_i)}{\partial \theta^2} (\theta' - \bar{\theta}) \right| \leq \|\theta' - \bar{\theta}\|_2^2 \left\| \frac{\partial^2 f(\tilde{\theta}; \mathbf{x}_i)}{\partial \theta^2} \right\|_2 \leq \frac{c_H \|\theta' - \bar{\theta}\|_2^2}{\sqrt{m}} \, .$$

Then, we have

$$
\begin{aligned}
I_2 &= \frac{1}{n}\sum_{i=1}^{n}\ell_i'(\theta'-\bar{\theta})^\top\frac{\partial^2 f(\tilde{\theta};\mathbf{x}_i)}{\partial\theta^2}(\theta'-\bar{\theta}) \\
&\leq \left|\sum_{i=1}^{n}\left(\frac{1}{\sqrt{n}}\ell_i'\right)\left(\frac{1}{\sqrt{n}}Q_i\right)\right| \\
&\overset{(a)}{\leq} \left(\frac{1}{n}\|[\ell_i']_i\|_2^2\right)^{1/2}\left(\frac{1}{n}\sum_{i=1}^{n}Q_i^2\right)^{1/2} \\
&\leq 2\sqrt{\mathcal{L}(\tilde{\theta})}\frac{c_H\|\theta'-\bar{\theta}\|_2^2}{\sqrt{m}} \;,
\end{aligned}
$$

where (a) follows by Cauchy-Schwartz. Putting the upper bounds on $I_1$ and $I_2$ back, we have

$$
\begin{aligned}
(\theta'-\bar{\theta})^\top\frac{\partial^2\mathcal{L}(\tilde{\theta})}{\partial\theta^2}(\theta'-\bar{\theta}) &\leq \left[2\varrho^2 + \frac{2c_H\sqrt{\mathcal{L}(\tilde{\theta})}}{\sqrt{m}}\right]\|\theta'-\bar{\theta}\|_2^2 \\
&\leq \left[2\varrho^2 + \frac{2c_H\sqrt{c_{\rho_1,\gamma}}}{\sqrt{m}}\right]\|\theta'-\bar{\theta}\|_2^2 \;,
\end{aligned}
$$

where the last inequality follows from Lemma 4.2. This proves the first statement of the theorem. Now, the second statement simply follows from the fact that by choosing $\gamma$ (and so $\rho$ and $\sigma_1$) according to the desirable operating regimes (see Remark 4.1) and by choosing $\rho_1$ according to Theorem 4.1, we obtain $c_H = O(\mathrm{poly}(L))$, $\rho^2 = O(\mathrm{poly}(L))$ and $c_{\rho_1,\gamma} = O(\mathrm{poly}(L))$. This completes the proof. $\qquad\square$

Next, we state and prove the smoothness result for general losses.

**Theorem B.2** (**Local Smoothness of Loss**). *Under Assumptions 3, 1, and 2, with probability at least* $(1 - \frac{2(L+1)}{m})$, $\mathcal{L}(\theta), \theta \in B^{\mathrm{Spec}}_{\rho,\rho_1}(\theta_0)$, *is $\beta$-smooth with $\beta = b\varrho^2 + \frac{c_H\sqrt{\lambda_t}}{\sqrt{m}}$ with $\varrho$ as in Lemma 4.1 and* $\lambda_t = \frac{1}{n}\sum_{i=1}^{n}(\ell_i')^2$ *with* $\ell_i' = \frac{\partial\ell(y_i,z)}{\partial z}\Big|_{z=f(\tilde{\theta}_t,\mathbf{x}_i))}$ *and $\tilde{\theta} \in B^{\mathrm{Spec}}_{\rho,\rho_1}$ being some point in the segment that joins $\theta'$ and $\theta$ as in (11).*

*Proof.* By the second order Taylor expansion about $\bar{\theta}$, we have $\mathcal{L}(\theta') = \mathcal{L}(\bar{\theta}) + \langle\theta'-\bar{\theta}, \nabla_\theta\mathcal{L}(\bar{\theta})\rangle + \frac{1}{2}(\theta'-\bar{\theta})^\top\frac{\partial^2\mathcal{L}(\tilde{\theta})}{\partial\theta^2}(\theta'-\bar{\theta})$, where $\tilde{\theta} = \xi\theta' + (1-\xi)\bar{\theta}$ for some $\xi \in [0,1]$. Then,

$$
\begin{aligned}
&(\theta'-\bar{\theta})^\top\frac{\partial^2\mathcal{L}(\tilde{\theta})}{\partial\theta^2}(\theta'-\bar{\theta}) \\
&= (\theta'-\bar{\theta})^\top\frac{1}{n}\sum_{i=1}^{n}\left[\ell_i''\frac{\partial f(\tilde{\theta};\mathbf{x}_i)}{\partial\theta}\frac{\partial f(\tilde{\theta};\mathbf{x}_i)}{\partial\theta}^\top + \ell_i'\frac{\partial^2 f(\tilde{\theta};\mathbf{x}_i)}{\partial\theta^2}\right](\theta'-\bar{\theta}) \\
&= \underbrace{\frac{1}{n}\sum_{i=1}^{n}\ell_i''\left\langle\theta'-\bar{\theta}, \frac{\partial f(\tilde{\theta};\mathbf{x}_i)}{\partial\theta}\right\rangle^2}_{I_1} + \underbrace{\frac{1}{n}\sum_{i=1}^{n}\ell_i'(\theta'-\bar{\theta})^\top\frac{\partial^2 f(\tilde{\theta};\mathbf{x}_i)}{\partial\theta^2}(\theta'-\bar{\theta})}_{I_2} \;.
\end{aligned}
$$

where $\ell_i = \ell(y_i, f(\tilde{\theta},\mathbf{x}_i))$, $\ell_i' = \frac{\partial\ell(y_i,z)}{\partial z}\Big|_{z=f(\tilde{\theta},\mathbf{x}_i))}$, and $\ell_i'' = \frac{\partial^2\ell(y_i,z)}{\partial z^2}\Big|_{z=f(\tilde{\theta},\mathbf{x}_i))}$. Now, note that

$$
\begin{aligned}
I_1 &= \frac{1}{n}\sum_{i=1}^{n}\ell_i''\left\langle\theta'-\bar{\theta}, \frac{\partial f(\tilde{\theta};\mathbf{x}_i)}{\partial\theta}\right\rangle^2 \\
&\overset{(a)}{\leq} \frac{b}{n}\sum_{i=1}^{n}\left\|\frac{\partial f(\tilde{\theta};\mathbf{x}_i)}{\partial\theta}\right\|_2^2\|\theta'-\bar{\theta}\|_2^2 \\
&\overset{(b)}{\leq} b\varrho^2\|\theta'-\bar{\theta}\|_2^2 \;,
\end{aligned}
$$

where (a) follows by the Cauchy-Schwartz inequality and (b) from Lemma 4.1.

For $I_2$, let $\lambda_t = \frac{1}{n}\|[\ell_i']_i\|_2^2$. Further, with $Q_{t,i} = (\theta' - \bar{\theta})^\top \frac{\partial^2 f(\tilde{\theta}; \mathbf{x}_i)}{\partial \theta^2}(\theta' - \bar{\theta})$, we have

$$|Q_{t,i}| = \left|(\theta' - \bar{\theta})^\top \frac{\partial^2 f(\tilde{\theta}_t; \mathbf{x}_i)}{\partial \theta^2}(\theta' - \bar{\theta})\right| \leq \|\theta' - \bar{\theta}\|_2^2 \left\|\frac{\partial^2 f(\tilde{\theta}_t; \mathbf{x}_i)}{\partial \theta^2}\right\|_2 \leq \frac{c_H\|\theta' - \bar{\theta}\|_2^2}{\sqrt{m}}.$$

Then, we have

$$\begin{aligned}
I_2 &= \frac{1}{n}\sum_{i=1}^n \ell_i'(\theta' - \bar{\theta})^\top \frac{\partial^2 f(\tilde{\theta}; \mathbf{x}_i)}{\partial \theta^2}(\theta' - \bar{\theta}) \\
&\leq \left|\sum_{i=1}^n \left(\frac{1}{\sqrt{n}}\ell_i'\right)\left(\frac{1}{\sqrt{n}}Q_i\right)\right| \\
&\stackrel{(a)}{\leq} \left(\frac{1}{n}\|[\ell_i']_i\|_2^2\right)^{1/2}\left(\frac{1}{n}\sum_{i=1}^n Q_i^2\right)^{1/2} \\
&\leq \sqrt{\lambda_t}\frac{c_H\|\theta' - \bar{\theta}\|_2^2}{\sqrt{m}},
\end{aligned}$$

where (a) follows by Cauchy-Schwartz. Putting the upper bounds on $I_1$ and $I_2$ back, we have

$$(\theta' - \bar{\theta})^\top \frac{\partial^2 \mathcal{L}(\tilde{\theta})}{\partial \theta^2}(\theta' - \bar{\theta}) \leq \left[b\varrho^2 + \frac{c_H\sqrt{\lambda_t}}{\sqrt{m}}\right]\|\theta' - \bar{\theta}\|_2^2.$$

This completes the proof. $\qquad\square$

**Lemma 5.1 (RSC $\Rightarrow$ RPL).** *Let $B_t := Q_\kappa^t \cap B_{\rho,\rho_1}^{\mathrm{Spec}}(\theta_0) \cap B_{\rho_2}^{\mathrm{Euc}}(\theta_t)$. In the setting of Theorem 5.1, if $\alpha_t > 0$, then the tuple $(B_t, \theta_t)$ satisfies the Restricted Polyak-Łojasiewicz (RPL) condition, i.e.,*

$$\mathcal{L}(\theta_t) - \inf_{\theta \in B_t}\mathcal{L}(\theta) \leq \frac{1}{2\alpha_t}\|\nabla_\theta \mathcal{L}(\theta_t)\|_2^2, \tag{12}$$

*with probability at least $(1 - \frac{2(L+1)}{m})$.*

*Proof.* Define

$$\hat{\mathcal{L}}_{\theta_t}(\theta) := \mathcal{L}(\theta_t) + \langle \theta - \theta_t, \nabla_\theta \mathcal{L}(\theta_t)\rangle + \frac{\alpha_t}{2}\|\theta - \theta_t\|_2^2.$$

By Theorem 5.1, $\forall \theta' \in B_t$, we have

$$\mathcal{L}(\theta') \geq \hat{\mathcal{L}}_{\theta_t}(\theta'). \tag{39}$$

Further, note that $\hat{\mathcal{L}}_{\theta_t}(\theta)$ is minimized at $\hat{\theta}_{t+1} := \theta_t - \nabla_\theta \mathcal{L}(\theta_t)/\alpha_t$ and the minimum value is:

$$\inf_\theta \hat{\mathcal{L}}_{\theta_t}(\theta) = \hat{\mathcal{L}}_{\theta_t}(\hat{\theta}_{t+1}) = \mathcal{L}(\theta_t) - \frac{1}{2\alpha_t}\|\nabla_\theta \mathcal{L}(\theta_t)\|_2^2.$$

Then, we have

$$\inf_{\theta \in B_t}\mathcal{L}(\theta) \stackrel{(a)}{\geq} \inf_{\theta \in B_t}\hat{\mathcal{L}}_{\theta_t}(\theta) \geq \inf_\theta \hat{\mathcal{L}}_{\theta_t}(\theta) = \mathcal{L}(\theta_t) - \frac{1}{2\alpha_t}\|\nabla_\theta \mathcal{L}(\theta_t)\|_2^2,$$

where (a) follows from (39). Rearranging terms completes the proof. $\qquad\square$

### B.2 Convergence with Restricted Strong Convexity

**Lemma 5.2 (Local Loss Reduction in $B_t$).** *Let $\alpha_t, \beta$ be as in Theorems 5.1 and 5.2 respectively, and $B_t := Q_\kappa^t \cap B_{\rho,\rho_1}^{\mathrm{Spec}}(\theta_0) \cap B_{\rho_2}^{\mathrm{Euc}}(\theta_t)$. Consider Assumptions 1, 2, and 4, and gradient descent with step size $\eta_t = \frac{\omega_t}{\beta}, \omega_t \in (0, 2)$. Then, for any $\bar{\theta}_{t+1} \in \arginf_{\theta \in B_t}\mathcal{L}(\theta)$, we have with probability at least $(1 - \frac{2(L+1)}{m})$,*

$$\mathcal{L}(\theta_{t+1}) - \mathcal{L}(\bar{\theta}_{t+1}) \leq \left(1 - \frac{\alpha_t \omega_t}{\beta}(2 - \omega_t)\right)(\mathcal{L}(\theta_t) - \mathcal{L}(\bar{\theta}_{t+1})). \tag{13}$$

*Proof.* Since $\mathcal{L}$ is $\beta$-smooth by Theorem 5.2, we have

$$
\begin{aligned}
\mathcal{L}(\theta_{t+1}) &\leq \mathcal{L}(\theta_t) + \langle \theta_{t+1} - \theta_t, \nabla_\theta \mathcal{L}(\theta_t) \rangle + \frac{\beta}{2} \|\theta_{t+1} - \theta_t\|_2^2 \\
&= \mathcal{L}(\theta_t) - \eta_t \|\nabla_\theta \mathcal{L}(\theta_t)\|_2^2 + \frac{\beta \eta_t^2}{2} \|\nabla_\theta \mathcal{L}(\theta_t)\|_2^2 \qquad (40) \\
&= \mathcal{L}(\theta_t) - \eta_t \left( 1 - \frac{\beta \eta_t}{2} \right) \|\nabla_\theta \mathcal{L}(\theta_t)\|_2^2
\end{aligned}
$$

Since $\bar{\theta}_{t+1} \in \operatorname{arginf}_{\theta \in B_t} \mathcal{L}(\theta)$ and $\alpha_t > 0$ by assumption, from Lemma 5.1 we obtain

$$
-\|\nabla_\theta \mathcal{L}(\theta_t)\|_2^2 \leq -2\alpha_t (\mathcal{L}(\theta_t) - \mathcal{L}(\bar{\theta}_{t+1})) .
$$

Hence

$$
\begin{aligned}
\mathcal{L}(\theta_{t+1}) - \mathcal{L}(\bar{\theta}_{t+1}) &\leq \mathcal{L}(\theta_t) - \mathcal{L}(\bar{\theta}_{t+1}) - \eta_t \left( 1 - \frac{\beta \eta_t}{2} \right) \|\nabla_\theta \mathcal{L}(\theta_t)\|_2^2 \\
&\overset{(a)}{\leq} \mathcal{L}(\theta_t) - \mathcal{L}(\bar{\theta}_{t+1}) - \eta_t \left( 1 - \frac{\beta \eta_t}{2} \right) 2\alpha_t (\mathcal{L}(\theta_t) - \mathcal{L}(\bar{\theta}_{t+1})) \\
&= \left( 1 - 2\alpha_t \eta_t \left( 1 - \frac{\beta \eta_t}{2} \right) \right) (\mathcal{L}(\theta_t) - \mathcal{L}(\bar{\theta}_{t+1}))
\end{aligned}
$$

where (a) follows for any $\eta_t \leq \frac{2}{\beta}$ because this implies $1 - \frac{\beta \eta_t}{2} \geq 0$. Choosing $\eta_t = \frac{\omega_t}{\beta}, \omega_t \in (0, 2)$,

$$
\mathcal{L}(\theta_{t+1}) - \mathcal{L}(\bar{\theta}_{t+1}) \leq \left( 1 - \frac{\alpha_t \omega_t}{\beta} (2 - \omega_t) \right) (\mathcal{L}(\theta_t) - \mathcal{L}(\bar{\theta}_{t+1})) .
$$

This completes the proof. $\qquad \square$

**Theorem 5.3** (**Global Loss Reduction in** $B_{\rho,\rho_1}^{\mathrm{Spec}}(\theta_0)$). *Let $\alpha_t, \beta$ be as in Theorems 5.1 and 5.2 respectively, and $B_t := Q_\kappa^t \cap B_{\rho,\rho_1}^{\mathrm{Spec}}(\theta_0) \cap B_{\rho_2}^{\mathrm{Euc}}(\theta_t)$. Let $\theta^* \in \operatorname{arginf}_{\theta \in B_{\rho,\rho_1}^{\mathrm{Spec}}(\theta_0)} \mathcal{L}(\theta), \bar{\theta}_{t+1} \in \operatorname{arginf}_{\theta \in B_t} \mathcal{L}(\theta)$, and $\gamma_t := \frac{\mathcal{L}(\bar{\theta}_{t+1}) - \mathcal{L}(\theta^*)}{\mathcal{L}(\theta_t) - \mathcal{L}(\theta^*)}$. Let $\alpha_t, \beta$ be as in Theorems 5.1 and 5.2 respectively, and $B_t := Q_\kappa^t \cap B_{\rho,\rho_1}^{\mathrm{Spec}}(\theta_0) \cap B_{\rho_2}^{\mathrm{Euc}}(\theta_t)$. Consider Assumptions 1, 2, and 4, and gradient descent with step size $\eta_t = \frac{\omega_t}{\beta}, \omega_t \in (0, 2)$. Then, with probability at least $(1 - \frac{2(L+1)}{m})$, we have we have $\gamma_t \in [0, 1)$ and*

$$
\mathcal{L}(\theta_{t+1}) - \mathcal{L}(\theta^*) \leq \left( 1 - \frac{\alpha_t \omega_t}{\beta} (1 - \gamma_t)(2 - \omega_t) \right) (\mathcal{L}(\theta_t) - \mathcal{L}(\theta^*)) . \qquad (14)
$$

*Proof.* We start by showing $\gamma_t = \frac{\mathcal{L}(\bar{\theta}_{t+1}) - \mathcal{L}(\theta^*)}{\mathcal{L}(\theta_t) - \mathcal{L}(\theta^*)}$ satisfies $0 \leq \gamma_t < 1$. Since $\theta^* \in \operatorname{arginf}_{\theta \in B_{\rho,\rho_1}^{\mathrm{Spec}}(\theta_0)} \mathcal{L}(\theta)$ and $\theta_{t+1} \in B_{\rho,\rho_1}^{\mathrm{Spec}}(\theta_0)$ by the definition of gradient descent and Assumption 4, we have

$$
\mathcal{L}(\theta^*) \leq \mathcal{L}(\theta_{t+1}) \overset{(a)}{\leq} \mathcal{L}(\theta_t) - \frac{\omega_t}{\beta} \left( 1 - \frac{\omega_t}{2} \right) \|\nabla_\theta \mathcal{L}(\theta_t)\|_2^2 < \mathcal{L}(\theta_t) ,
$$

where (a) follows from (40). Since $\mathcal{L}(\bar{\theta}_{t+1}) \geq \mathcal{L}(\theta^*)$ and $\mathcal{L}(\theta_t) > \mathcal{L}(\theta^*)$, we have $\gamma_t \geq 0$. Further, we have $\mathcal{L}(\bar{\theta}_{t+1}) < \mathcal{L}(\theta_t)$, and so we have $\gamma_t < 1$. To see this, consider two cases: (i) $\theta_{t+1} \in B_t$ and (ii) $\theta_{t+1} \notin B_t$. When $\theta_{t+1} \in B_t$, we have $\mathcal{L}(\bar{\theta}_{t+1}) \leq \mathcal{L}(\theta_{t+1}) < \mathcal{L}(\theta_t)$. When $\theta_{t+1} \notin B_t$, we only consider the case $\mathcal{L}(\theta_{t+1}) < \mathcal{L}(\bar{\theta}_{t+1})$; otherwise, if $\mathcal{L}(\theta_{t+1}) \geq \mathcal{L}(\bar{\theta}_{t+1})$ then it follows $\mathcal{L}(\bar{\theta}_{t+1}) < \mathcal{L}(\theta_t)$ by (40). So, let us consider level sets of the loss between $\mathcal{L}(\theta_{t+1})$ and $\mathcal{L}(\theta_t)$. Because of the definition of $Q_\kappa^t$ (which defines a cone), the RSC property due to $\theta' \in B_t$, and the smoothness of the loss, we will have some $\theta' \in B_t$ living in one of those level sets such that $\mathcal{L}(\theta_{t+1}) \leq \mathcal{L}(\theta') < \mathcal{L}(\theta_t)$, but then $\mathcal{L}(\bar{\theta}_{t+1}) \leq \mathcal{L}(\theta')$ by definition, implying $\mathcal{L}(\bar{\theta}_{t+1}) < \mathcal{L}(\theta_t)$. Hence, $\gamma_t < 1$.

Now, with $\omega_t \in (0, 2)$, we have

$$
\begin{aligned}
& \mathcal{L}(\theta_{t+1}) - \mathcal{L}(\theta^*) \\
& = \mathcal{L}(\theta_{t+1}) - \mathcal{L}(\bar{\theta}_{t+1}) + \mathcal{L}(\bar{\theta}_{t+1}) - \mathcal{L}(\theta^*) \\
& \leq \left(1 - \frac{\alpha_t \omega_t}{\beta}(2 - \omega_t)\right)(\mathcal{L}(\theta_t) - \mathcal{L}(\bar{\theta}_{t+1})) + \left(1 - \frac{\alpha_t \omega_t}{\beta}(2 - \omega_t)\right)(\mathcal{L}(\bar{\theta}_{t+1}) - \mathcal{L}(\theta^*)) \\
& \quad + \left(\mathcal{L}(\bar{\theta}_{t+1}) - \left(1 - \frac{\alpha_t \omega_t}{\beta}(2 - \omega_t)\right)\mathcal{L}(\bar{\theta}_{t+1})\right) - \left(\mathcal{L}(\theta^*) - \left(1 - \frac{\alpha_t \omega_t}{\beta}(2 - \omega_t)\right)\mathcal{L}(\theta^*)\right) \\
& = \left(1 - \frac{\alpha_t \omega_t}{\beta}(2 - \omega_t)\right)(\mathcal{L}(\theta_t) - \mathcal{L}(\theta^*)) + \frac{\alpha_t \omega_t}{\beta}(2 - \omega_t)(\mathcal{L}(\bar{\theta}_{t+1}) - \mathcal{L}(\theta^*)) \\
& = \left(1 - \frac{\alpha_t \omega_t}{\beta}(2 - \omega_t)\right)(\mathcal{L}(\theta_t) - \mathcal{L}(\theta^*)) + \frac{\alpha_t \omega_t}{\beta}(2 - \omega_t)\gamma_t(\mathcal{L}(\theta_t) - \mathcal{L}(\theta^*)) \\
& = \left(1 - \frac{\alpha_t \omega_t}{\beta}(1 - \gamma_t)(2 - \omega_t)\right)(\mathcal{L}(\theta_t) - \mathcal{L}(\theta^*)) .
\end{aligned}
$$

That completes the proof. $\square$

## C   ANALYSIS FOR MODELS WITH $k$ OUTPUTS

In this section, we illustrate that our results can be extended to neural models with $k$ outputs.

### C.1   OPTIMIZATION SETUP WITH $k$ OUTPUTS

Consider a training set $\mathcal{D} = \{\mathbf{x}_i, \mathbf{y}_i\}_{i=1}^n, \mathbf{x}_i \in \mathcal{X} \subseteq \mathbb{R}^d, \mathbf{y}_i \in \mathcal{Y} \subseteq \mathbb{R}^k, k \geq 1$. We will denote by $X \in \mathbb{R}^{n \times d}$ the matrix whose $i$th row is $\mathbf{x}_i^\top$. The goal is to minimize the square loss:

$$
\mathcal{L}(\theta) = \frac{1}{n} \sum_{i=1}^n \|\mathbf{y}_i - \hat{\mathbf{y}}_i\|_2^2 = \frac{1}{n} \sum_{i=1}^n \sum_{h=1}^k (y_{ih} - f_h(\theta; \mathbf{x}_i))^2 ,
$$

where the prediction $\hat{\mathbf{y}}_i := \mathbf{f}(\theta; \mathbf{x}_i) \in \mathbb{R}^k$ is from a deep model, $f_h(\theta; \mathbf{x}_i), h \in [k]$ denotes the $h^{th}$ output, and $\theta \in \mathbb{R}^p$ denotes the parameter vector. In our setting $\mathbf{f}$ is a feed-forward multi-layer (fully-connected) neural network with depth $L$ and widths $m_l, l \in [L] := \{1, \dots, L\}$ given by

$$
\begin{aligned}
\alpha^{(0)}(\mathbf{x}) &= \mathbf{x} , \\
\alpha^{(l)}(\mathbf{x}) &= \phi\left(\frac{1}{\sqrt{m_{l-1}}} W^{(l)} \alpha^{(l-1)}(\mathbf{x})\right) , \quad l = 1, \dots, L , \\
\mathbf{f}(\theta; \mathbf{x}) = \alpha^{(L+1)}(\mathbf{x}) &= \frac{1}{\sqrt{m_L}} V^\top \alpha^{(L)}(\mathbf{x}) ,
\end{aligned} \tag{41}
$$

where $W^{(l)} \in \mathbb{R}^{m_l \times m_{l-1}}, l \in [L]$ are layer-wise weight matrices, $V \in \mathbb{R}^{m_L \times k}$ is the last layer matrix, $\phi(\cdot)$ is the smooth (pointwise) activation function, and the total set of parameters

$$
\theta := (\text{vec}(W^{(1)})^\top, \dots, \text{vec}(W^{(L)})^\top, V^\top)^\top \in \mathbb{R}^{\sum_{l=1}^L m_l m_{l-1} + k m_L} , \tag{42}
$$

with $m_0 = d$. Note that the total number of parameters is $p = \sum_{l=1}^L m_l m_{l-1} + k m_L$. For simplicity, we will assume that the width of all the layers is the same, i.e., $m_l = m, l \in [L]$.

Define the pointwise loss $\ell_{ih} := (y_{ih} - \hat{y}_{ih})^2$ and denote its first- and second-derivative w.r.t. $\hat{y}_{ih}$ as $\ell'_{ih} := -2(y_{ih} - \hat{y}_{ih})$ and $\ell''_{ih} := 2$. Let $f_h(\theta; \mathbf{x}), h \in [k]$ denote the $h^{th}$ output, and let the gradient and Hessian of $f_h(\theta; \mathbf{x}_i)$ w.r.t. $\theta$ be denoted as

$$
\nabla f_{ih} := \frac{\partial f_h(\theta; \mathbf{x}_i)}{\partial \theta} , \quad \text{and} \quad \nabla^2 f_{ih} := \frac{\partial^2 f_h(\theta; \mathbf{x}_i)}{\partial \theta^2} .
$$

By chain rule, the gradient and Hessian of the empirical loss w.r.t. $\theta$ are given by

$$\frac{\partial \mathcal{L}(\theta)}{\partial \theta} = \frac{1}{n} \sum_{i=1}^{n} \sum_{h=1}^{k} \ell'_{ih} \nabla f_{ih} ,$$

$$\frac{\partial^2 \mathcal{L}(\theta)}{\partial \theta^2} = \frac{1}{n} \sum_{i=1}^{n} \sum_{h=1}^{k} \left[ \ell''_{ih} \nabla f_{ih} \nabla f_{ih}^\top + \ell'_{ih} \nabla^2 f_{ih} \right] .$$

For the last layer, note that

$$f_h(\theta; \mathbf{x}_i) = \frac{1}{\sqrt{m}} \mathbf{v}_h^T \alpha^{(L)}(\mathbf{x})$$

where $\mathbf{v}_h \in \mathbb{R}^m$ is the last layer vector corresponding to output $f_h$ and $V = [\mathbf{v}_h] \in \mathbb{R}^{m \times k}$ is the last layer matrix. For the analysis, we update the definition of the spectral norm ball to work with each last layer vector $\mathbf{v}_h$:

$$B_{\rho,\rho_1}^{\mathrm{Spec}}(\bar{\theta}) := \left\{ \theta \in \mathbb{R}^p \text{ as in (42) } \mid \|W^{(\ell)} - \overline{W}^{(\ell)}\|_2 \le \rho, \ell \in [L], \|\mathbf{v}_h - \bar{\mathbf{v}}_h\|_2 \le \rho_1, h \in [k] \right\} ,$$
(43)

Similarly, we update the initialization so each of the last layer vectors are unit norm.

**Assumption 5 (Initialization).** *The initialization weights* $w_{0,ij}^{(l)} \sim \mathcal{N}(0, \sigma_0^2)$ *for* $l \in [L]$ *where* $\sigma_0 = \frac{\sigma_1}{2\left(1 + \frac{\sqrt{\log m}}{\sqrt{2m}}\right)}, \sigma_1 > 0$, *and* $\mathbf{v}_{0,h}, h \in [k]$ *are random unit vectors with* $\|\mathbf{v}_{0,h}\|_2 = 1$. *Further, we assume the input data satisfies:* $\|\mathbf{x}_i\|_2 = \sqrt{d}, i \in [n]$.

Based on the setup, following Theorem 4.1, we get the following result for the Hessian of each $f_h$:

**Theorem C.1 (Hessian Spectral Norm Bound).** *Under Assumptions 1, and 5, for* $\theta \in B_{\rho,\rho_1}^{\mathrm{Spec}}(\theta_0)$ *as in (43),* $\rho_1 = O(\mathrm{poly}(L))$, *with probability at least* $(1 - \frac{2(L+1)}{m})$, *for any* $\mathbf{x}_i, i \in [n]$, *we have*

$$\left\| \nabla_\theta^2 f(\theta; \mathbf{x}_i) \right\|_2 \le \frac{c_H}{\sqrt{m}} ,$$
(44)

*with* $c_H = O(\mathrm{poly}(L)(1 + \gamma^{2L}))$ *where* $\gamma := \sigma_1 + \frac{\rho}{\sqrt{m}}$.

## C.2 RESTRICTED STRONG CONVEXITY AND SMOOTHNESS

Let us assume we have a sequence of iterates $\{\theta_t\}_{t \ge 0}$ from gradient descent. Our RSC analysis will rely on the following $\tilde{Q}_\kappa^t$-sets at step $t$, which avoids directions almost orthogonal to the average gradient of the predictor.

**Definition C.1 ($\tilde{\mathbf{Q}}_\kappa^\mathbf{t}$ sets).** *For iterate* $\theta_t \in \mathbb{R}^p$, *let* $\bar{\mathbf{g}}_t = \frac{1}{nk} \sum_{i=1}^{n} \sum_{h=1}^{k} \nabla_\theta f_h(\theta_t; \mathbf{x}_i)$. *For any* $\kappa \in (0,1]$, *define* $\tilde{Q}_\kappa^t := \{\theta \mid |\cos(\theta - \theta_t, \bar{\mathbf{g}}_t)| \ge \kappa\}$.

We define the set $B_t := \tilde{Q}_\kappa^t \cap B_{\rho,\rho_1}^{\mathrm{Spec}}(\theta_0) \cap B_{\rho_2}^{\mathrm{Euc}}(\theta_t)$. We focus on establishing RSC w.r.t. the tuple $(B_t, \theta_t)$, where $B_{\rho,\rho_1}^{\mathrm{Spec}}(\theta_0)$ becomes the feasible set for the optimization and $B_{\rho_2}^{\mathrm{Euc}}(\theta_t)$ is an Euclidean ball around the current iterate.

**Theorem C.2 (RSC for $k$-output Square Loss).** *For square loss, under Assumptions 1, 5, and 3, with probability at least* $(1 - \frac{2(L+1)}{m})$, $\forall \theta' \in \tilde{Q}_\kappa^t \cap B_{\rho,\rho_1}^{\mathrm{Spec}}(\theta_0) \cap B_{\rho_2}^{\mathrm{Euc}}(\theta_t)$,

$$\mathcal{L}(\theta') \ge \mathcal{L}(\theta_t) + \langle \theta' - \theta_t, \nabla_\theta \mathcal{L}(\theta_t) \rangle + \frac{\alpha_t}{2} \|\theta' - \theta_t\|_2^2 ,$$

$$\text{with} \quad \alpha_t = c_1 k \|\bar{\mathbf{g}}_t\|_2^2 - \frac{kc_2}{\sqrt{m}}$$
(45)

*where* $\bar{\mathbf{g}}_t = \frac{1}{nk} \sum_{i=1}^{n} \sum_{h=1}^{k} \nabla_\theta f_h(\theta_t; \mathbf{x}_i)$, $c_1 = 2\kappa^2$ *and* $c_2 = 2c_H(2\varrho\rho_2 + \sqrt{kc_{\rho_1,\gamma}})$, *with* $c_H$ *is as in Theorem 4.1,* $\varrho$ *as in Lemma 4.1, and* $c_{\rho_1,\gamma}$ *as in Lemma 4.2. Consequently,* $\mathcal{L}$ *satisfies RSC w.r.t.* $(\tilde{Q}_\kappa^t \cap B_{\rho,\rho_1}^{\mathrm{Spec}}(\theta_0) \cap B_{\rho_2}^{\mathrm{Euc}}(\theta_t), \theta_t)$ *whenever* $\alpha_t > 0$.

*Proof.* For any $\theta' \in \tilde{Q}_\kappa^t \cap B_\rho(\theta_0)$, by the second order Taylor expansion around $\theta_t$, we have

$$\mathcal{L}(\theta') = \mathcal{L}(\theta_t) + \langle \theta' - \theta_t, \nabla_\theta \mathcal{L}(\theta_t) \rangle + \frac{1}{2}(\theta' - \theta_t)^\top \frac{\partial^2 \mathcal{L}(\tilde{\theta}_t)}{\partial \theta^2}(\theta' - \theta_t) \,,$$

where $\tilde{\theta}_t = \xi \theta' + (1 - \xi)\theta_t$ for some $\xi \in [0, 1]$. We note that $\tilde{\theta}_t \in B_{\rho,\rho_1}^{\mathrm{Spec}}(\theta_0)$ since,

- $\left\| \tilde{\theta}_t^{(l)} - \theta_0^{(l)} \right\|_2 = \left\| \xi \theta'^{(l)} - \xi \theta_0^{(l)} + (1 - \xi)\theta_t^{(l)} - (1 - \xi)\theta_0^{(l)} \right\|_2 \leq \xi \left\| \theta'^{(l)} - \theta_0^{(l)} \right\|_2 + (1 - \xi) \left\| \theta_t^{(l)} - \theta_0^{(l)} \right\|_2 \leq \rho$, for any $l \in [L]$, where the last inequality follows from our assumption $\theta_t'^{(l)} \theta_t^{(l)} \in B_{\rho,\rho_1}^{\mathrm{Spec}}(\theta_0)$; and

- $\left\| \tilde{\theta}_t^{(L+1)} - \theta_0^{(L+1)} \right\|_2 = \left\| \tilde{V} - V_0 \right\|_F \leq k\rho_1$, by following a similar derivation as in the previous point.

Focusing on the quadratic form in the Taylor expansion and recalling the form of the Hessian, we get

$$(\theta' - \theta_t)^\top \frac{\partial^2 \mathcal{L}(\tilde{\theta}_t)}{\partial \theta^2}(\theta' - \theta_t)$$

$$= (\theta' - \theta_t)^\top \frac{1}{n} \sum_{i=1}^n \sum_{h=1}^k \left[ \ell_{ih}'' \frac{\partial f_h(\tilde{\theta}_t; \mathbf{x}_i)}{\partial \theta} \frac{\partial f_h(\tilde{\theta}_t; \mathbf{x}_i)}{\partial \theta}^\top + \ell_{ih}' \frac{\partial^2 f_h(\tilde{\theta}_t; \mathbf{x}_i)}{\partial \theta^2} \right] (\theta' - \theta_t)$$

$$= \underbrace{\frac{1}{n} \sum_{i=1}^n \sum_{h=1}^k \ell_{ih}'' \left\langle \theta' - \theta_t, \frac{\partial f_h(\tilde{\theta}_t; \mathbf{x}_i)}{\partial \theta} \right\rangle^2}_{I_1} + \underbrace{\frac{1}{n} \sum_{i=1}^n \sum_{h=1}^k \ell_{ih}'(\theta' - \theta_t)^\top \frac{\partial^2 f_h(\tilde{\theta}_t; \mathbf{x}_i)}{\partial \theta^2}(\theta' - \theta_t)}_{I_2} \,,$$

where $\ell_{ih} = \ell(y_{ih}, f_h(\tilde{\theta}_t, \mathbf{x}_i))$, $\ell_{ih}' = \frac{\partial \ell(y_{ih}, z)}{\partial z}\Big|_{z=f_h(\tilde{\theta}_t, \mathbf{x}_i))}$, and $\ell_{ih}'' = \frac{\partial^2 \ell(y_{ih}, z)}{\partial z^2}\Big|_{z=f_h(\tilde{\theta}_t, \mathbf{x}_i))}$.

Now, note that

$$
\begin{aligned}
I_1 &= \frac{1}{n} \sum_{i=1}^{n} \sum_{h=1}^{k} \ell_{ih}'' \left\langle \theta' - \theta_t, \frac{\partial f_h(\tilde{\theta}_t; \mathbf{x}_i)}{\partial \theta} \right\rangle^2 \\
&\geq \frac{2}{n} \sum_{i=1}^{n} \sum_{h=1}^{k} \left\langle \theta' - \theta_t, \frac{\partial f_h(\theta_t; \mathbf{x}_i)}{\partial \theta} + \left( \frac{\partial f_h(\tilde{\theta}_t; \mathbf{x}_i)}{\partial \theta} - \frac{\partial f_h(\theta_t; \mathbf{x}_i)}{\partial \theta} \right) \right\rangle^2 \\
&= \frac{2}{n} \sum_{i=1}^{n} \sum_{h=1}^{k} \left\langle \theta' - \theta_t, \frac{\partial f_h(\theta_t; \mathbf{x}_i)}{\partial \theta} \right\rangle^2 + \frac{2}{n} \sum_{i=1}^{n} \sum_{h=1}^{k} \left\langle \theta' - \theta_t, \frac{\partial f_h(\tilde{\theta}_t; \mathbf{x}_i)}{\partial \theta} - \frac{\partial f_h(\theta_t; \mathbf{x}_i)}{\partial \theta} \right\rangle^2 \\
&\quad + \frac{4}{n} \sum_{i=1}^{n} \sum_{h=1}^{k} \left\langle \theta' - \theta_t, \frac{\partial f_h(\theta_t; \mathbf{x}_i)}{\partial \theta} \right\rangle \left\langle \theta' - \theta_t, \frac{\partial f_h(\tilde{\theta}_t; \mathbf{x}_i)}{\partial \theta} - \frac{\partial f_h(\theta_t; \mathbf{x}_i)}{\partial \theta} \right\rangle \\
&\overset{(a)}{\geq} \frac{2}{n} \sum_{i=1}^{n} \sum_{h=1}^{k} \left\langle \theta' - \theta_t, \frac{\partial f_h(\theta_t; \mathbf{x}_i)}{\partial \theta} \right\rangle^2 - \frac{4}{n} \sum_{i=1}^{n} \sum_{h=1}^{k} \left\| \frac{\partial f_h(\theta_t; \mathbf{x}_i)}{\partial \theta} \right\|_2 \left\| \frac{\partial f_h(\tilde{\theta}_t; \mathbf{x}_i)}{\partial \theta} - \frac{\partial f_h(\theta_t; \mathbf{x}_i)}{\partial \theta} \right\|_2 \|\theta' - \theta_t\|_2^2 \\
&\overset{(b)}{\geq} 2k \left\langle \theta' - \theta_t, \frac{1}{nk} \sum_{i=1}^{n} \sum_{h=1}^{k} \frac{\partial f_h(\theta_t; \mathbf{x}_i)}{\partial \theta} \right\rangle^2 - \frac{4}{n} \sum_{i=1}^{n} \sum_{h=1}^{k} \varrho \frac{c_H}{\sqrt{m}} \|\tilde{\theta}_t - \theta_t\|_2 \|\theta' - \theta_t\|_2^2 \\
&\overset{(c)}{\geq} 2k \left\langle \theta' - \theta_t, \frac{1}{nk} \sum_{i=1}^{n} \sum_{h=1}^{k} \frac{\partial f_h(\theta_t; \mathbf{x}_i)}{\partial \theta} \right\rangle^2 - \frac{4k\varrho c_H}{\sqrt{m}} \|\theta' - \theta_t\|_2^3 \\
&\overset{(d)}{\geq} 2\kappa^2 k \left\| \frac{2}{nk} \sum_{i=1}^{n} \sum_{h=1}^{k} \frac{\partial f_h(\theta_t; \mathbf{x}_i)}{\partial \theta} \right\|_2^2 \|\theta' - \theta_t\|_2^2 - \frac{4k\varrho c_H}{\sqrt{m}} \|\theta' - \theta_t\|_2^3 \\
&= k \left( 2\kappa^2 \left\| \frac{1}{nk} \sum_{i=1}^{n} \sum_{h=1}^{k} \frac{\partial f_h(\theta_t; \mathbf{x}_i)}{\partial \theta} \right\|_2^2 - \frac{4\varrho c_H \|\theta' - \theta_t\|_2}{\sqrt{m}} \right) \|\theta' - \theta_t\|_2^2,
\end{aligned}
$$

where (a) follows by Cauchy-Schwartz inequality; (b) follows by Jensen's inequality (first term) and the use of Theorem 4.1 and Lemma 4.1 due to $\tilde{\theta}_t \in B_{\rho,\rho_1}^{\text{Spec}}(\theta_0)$; (c) follows from $\left\| \tilde{\theta}_t - \theta_t \right\|_2 = \|\xi\theta' + (1-\xi)\theta_t - \theta_t\|_2 = \xi \|\theta' - \theta_t\| \leq \|\theta' - \theta_t\|_2$; (d) follows since $\theta' \in Q_\kappa^t$ and from the fact that $p^\top q = \cos(p, q) \|p\| \|q\|$ for any vectors $p, q$.

For analyzing $I_2$, first note that for square loss, $\ell_{ih,t}' = 2(\hat{y}_{ih,t} - y_{ih})$ with $\hat{y}_{ih,t} = f_h(\theta_t; \mathbf{x}_i)$, so that for the vector $[\ell_{ih,t}']$, we have $\frac{1}{n}\|[\ell_{ih,t}']\|_2^2 = \frac{4}{n} \sum_{i=1}^{n} \sum_{h=1}^{k} (\hat{y}_{ih,t} - y_{ih})^2 = 4\mathcal{L}(\theta_t)$. Further, with $Q_{ih,t} = (\theta' - \theta_t)^\top \frac{\partial^2 f_h(\tilde{\theta}_t; \mathbf{x}_i)}{\partial \theta^2}(\theta' - \theta_t)$, we have

$$
|Q_{ih,t}| = \left| (\theta' - \theta_t)^\top \frac{\partial^2 f_h(\tilde{\theta}_t; \mathbf{x}_i)}{\partial \theta^2}(\theta' - \theta_t) \right| \leq \|\theta' - \theta_t\|_2^2 \left\| \frac{\partial^2 f_h(\tilde{\theta}_t; \mathbf{x}_i)}{\partial \theta^2} \right\|_2 \leq \frac{c_H \|\theta' - \theta_t\|_2^2}{\sqrt{m}}.
$$

Now, note that

$$
\begin{aligned}
I_2 &= \frac{1}{n} \sum_{i=1}^{n} \sum_{h=1}^{k} \ell_{ih,t}' Q_{ih,t} \\
&\geq -\left| \sum_{i=1}^{n} \sum_{h=1}^{k} \left( \frac{1}{\sqrt{n}} \ell_{ih,t}' \right) \left( \frac{1}{\sqrt{n}} Q_{ih,t} \right) \right| \\
&\overset{(a)}{\geq} -\left( \frac{1}{n} \sum_{i=1}^{n} \sum_{h=1}^{k} \ell_{ih,t}'^2 \right)^{1/2} \left( \frac{1}{n} \sum_{i=1}^{n} \sum_{h=1}^{k} Q_{ih,t}^2 \right)^{1/2} \\
&\geq -2\sqrt{\mathcal{L}(\tilde{\theta}_t)} \frac{c_H k \|\theta' - \theta_t\|_2^2}{\sqrt{m}},
\end{aligned}
$$

where (a) follows by Cauchy-Schwartz inequality.

Putting the lower bounds on $I_1$ and $I_2$ back, with $\bar{\mathbf{g}}_t = \frac{1}{nk} \sum_{i=1}^{n} \sum_{h=1}^{k} \frac{\partial f_h(\theta_t; \mathbf{x}_i)}{\partial \theta}$, we have

$$(\theta' - \theta_t)^\top \frac{\partial^2 \mathcal{L}(\tilde{\theta}_t)}{\partial \theta^2} (\theta' - \theta_t) \geq k \left( 2\kappa^2 \|\bar{\mathbf{g}}_t\|_2^2 - \frac{4\varrho c_H \|\theta' - \theta_t\|_2 + 2c_H \sqrt{\mathcal{L}(\tilde{\theta}_t)}}{\sqrt{m}} \right) \|\theta' - \theta_t\|_2^2 .$$

Now, since $\theta' \in B_{\rho_2}^{\mathrm{Euc}}(\theta_t)$, $\|\theta' - \theta_t\|_2 \leq \rho_2$, so we have

$$(\theta' - \theta_t)^\top \frac{\partial^2 \mathcal{L}(\tilde{\theta}_t)}{\partial \theta^2} (\theta' - \theta_t) \geq k \left( 2\kappa^2 \|\bar{\mathbf{g}}_t\|_2^2 - \frac{4\varrho c_H \rho_2 + 2c_H \sqrt{\mathcal{L}(\tilde{\theta}_t)}}{\sqrt{m}} \right) \|\theta' - \theta_t\|_2^2$$

$$\geq k \left( 2\kappa^2 \|\bar{\mathbf{g}}_t\|_2^2 - \frac{4\varrho c_H \rho_2 + 2c_H \sqrt{kc_{\rho_1,\gamma}}}{\sqrt{m}} \right) \|\theta' - \theta_t\|_2^2 ,$$

where the last inequality follows from Lemma 4.2. That completes the proof. $\qquad \square$

**Theorem C.3** (**Local Smoothness for $k$–output Square Loss**). *Under Assumptions 1, 5, and 3, with probability at least $(1 - \frac{2(L+1)}{m})$, $\forall \theta, \theta' \in B_{\rho,\rho_1}^{\mathrm{Spec}}(\theta_0)$,*

$$\mathcal{L}(\theta') \leq \mathcal{L}(\theta) + \langle \theta' - \theta, \nabla_\theta \mathcal{L}(\theta) \rangle + \frac{\beta}{2} \|\theta' - \theta\|_2^2 , \quad with \quad \beta = 2k\varrho^2 + \frac{2kc_H \sqrt{kc_{\rho_1,\gamma}}}{\sqrt{m}} , \qquad (46)$$

*with $c_H$ as in Theorem 4.1, $\varrho$ as in Lemma 4.1, and $c_{\rho_1,\gamma}$ as in Lemma 4.2. Consequently, $\mathcal{L}$ is locally $\beta$-smooth.*

*Proof.* By the second order Taylor expansion about $\bar{\theta} \in B_{\rho,\rho_1}^{\mathrm{Spec}}(\theta_0)$, we have $\mathcal{L}(\theta') = \mathcal{L}(\bar{\theta}) + \langle \theta' - \bar{\theta}, \nabla_\theta \mathcal{L}(\bar{\theta}) \rangle + \frac{1}{2}(\theta' - \bar{\theta})^\top \frac{\partial^2 \mathcal{L}(\tilde{\theta})}{\partial \theta^2}(\theta' - \bar{\theta})$, where $\tilde{\theta} = \xi \theta' + (1 - \xi)\bar{\theta}$ for some $\xi \in [0, 1]$. Then,

$$(\theta' - \bar{\theta})^\top \frac{\partial^2 \mathcal{L}(\tilde{\theta})}{\partial \theta^2} (\theta' - \bar{\theta})$$

$$= (\theta' - \bar{\theta})^\top \frac{1}{n} \sum_{i=1}^{n} \sum_{h=1}^{k} \left[ \ell_{ih}'' \frac{\partial f_h(\tilde{\theta}; \mathbf{x}_i)}{\partial \theta} \frac{\partial f_h(\tilde{\theta}; \mathbf{x}_i)}{\partial \theta}^\top + \ell_{ih}' \frac{\partial^2 f_h(\tilde{\theta}; \mathbf{x}_i)}{\partial \theta^2} \right] (\theta' - \bar{\theta})$$

$$= \underbrace{\frac{1}{n} \sum_{i=1}^{n} \sum_{h=1}^{k} \ell_{ih}'' \left\langle \theta' - \bar{\theta}, \frac{\partial f_h(\tilde{\theta}; \mathbf{x}_i)}{\partial \theta} \right\rangle^2}_{I_1} + \underbrace{\frac{1}{n} \sum_{i=1}^{n} \sum_{h=1}^{k} \ell_{ih}' (\theta' - \bar{\theta})^\top \frac{\partial^2 f_h(\tilde{\theta}; \mathbf{x}_i)}{\partial \theta^2} (\theta' - \bar{\theta})}_{I_2} ,$$

where $\ell_{ih} = \ell(y_i, f_h(\tilde{\theta}, \mathbf{x}_i))$, $\ell_{ih}' = \left. \frac{\partial \ell(y_{ih}, z)}{\partial z} \right|_{z = f_h(\tilde{\theta}, \mathbf{x}_i))}$, and $\ell_{ih}'' = \left. \frac{\partial^2 \ell(y_{ih}, z)}{\partial z^2} \right|_{z = f_h(\tilde{\theta}, \mathbf{x}_i))}$.

Now, note that

$$I_1 = \frac{1}{n} \sum_{i=1}^{n} \sum_{h=1}^{k} \ell_{ih}'' \left\langle \theta' - \bar{\theta}, \frac{\partial f_h(\tilde{\theta}; \mathbf{x}_i)}{\partial \theta} \right\rangle^2$$

$$\overset{(a)}{\leq} \frac{2}{n} \sum_{i=1}^{n} \sum_{h=1}^{k} \left\| \frac{\partial f_h(\tilde{\theta}; \mathbf{x}_i)}{\partial \theta} \right\|_2^2 \|\theta' - \bar{\theta}\|_2^2$$

$$\overset{(b)}{\leq} 2k\varrho^2 \|\theta' - \bar{\theta}\|_2^2 ,$$

where (a) follows by the Cauchy-Schwartz inequality and (b) from Lemma 4.1.

For $I_2$, first note that for square loss, $\ell_{ih}' = 2(\hat{y}_{ih} - y_{ih})$ with $\hat{y}_{ih} = f_h(\tilde{\theta}; \mathbf{x}_i)$, so that for the vector $[\ell_{ih}']$, we have $\frac{1}{n} \|[\ell_{ih}']\|_2^2 = \frac{4}{n} \sum_{i=1}^{n} \sum_{h=1}^{k} (\hat{y}_{ih} - y_{ih})^2 = 4\mathcal{L}(\tilde{\theta})$. Further, with

$$Q_{ih} = (\theta' - \bar{\theta})^\top \frac{\partial^2 f_h(\tilde{\theta}_t; \mathbf{x}_i)}{\partial \theta^2} (\theta' - \bar{\theta}),$$

we have

$$|Q_{ih}| = \left| (\theta' - \bar\theta)^\top \frac{\partial^2 f_h(\tilde\theta; \mathbf{x}_i)}{\partial \theta^2} (\theta' - \bar\theta) \right| \le \|\theta' - \bar\theta\|_2^2 \left\| \frac{\partial^2 f_h(\tilde\theta; \mathbf{x}_i)}{\partial \theta^2} \right\|_2 \le \frac{c_H \|\theta' - \bar\theta\|_2^2}{\sqrt{m}} .$$

Then, we have

$$I_2 = \frac{1}{n} \sum_{i=1}^{n} \sum_{h=1}^{k} \ell'_{ih} (\theta' - \bar\theta)^\top \frac{\partial^2 f_h(\tilde\theta; \mathbf{x}_i)}{\partial \theta^2} (\theta' - \bar\theta)$$

$$\le \left| \sum_{i=1}^{n} \sum_{h=1}^{k} \left( \frac{1}{\sqrt{n}} \ell'_{ih} \right) \left( \frac{1}{\sqrt{n}} Q_{ih} \right) \right|$$

$$\overset{(a)}{\le} \left( \frac{1}{n} \sum_{h=1}^{k} \| [\ell'_{ih}] \|_2^2 \right)^{1/2} \left( \frac{1}{n} \sum_{i=1}^{n} \sum_{h=1}^{k} Q_{ih}^2 \right)^{1/2}$$

$$\le 2 \sqrt{\mathcal{L}(\tilde\theta)} \frac{k c_H \|\theta' - \bar\theta\|_2^2}{\sqrt{m}} ,$$

where (a) follows by Cauchy-Schwartz. Putting the upper bounds on $I_1$ and $I_2$ back, we have

$$(\theta' - \bar\theta)^\top \frac{\partial^2 \mathcal{L}(\tilde\theta)}{\partial \theta^2} (\theta' - \bar\theta) \le \left[ 2k\varrho^2 + \frac{2k c_H \sqrt{\mathcal{L}(\tilde\theta)}}{\sqrt{m}} \right] \|\theta' - \bar\theta\|_2^2$$

$$\le \left[ 2k\varrho^2 + \frac{2k c_H \sqrt{k c_{\rho_1,\gamma}}}{\sqrt{m}} \right] \|\theta' - \bar\theta\|_2^2 ,$$

where the last inequality follows from Lemma 4.2. That completes the proof. $\qquad\square$

Note that we now have the RSC and smoothness results for the $k$-output case similar to the single output case. With these properties, the rest of the convergence analysis for the $k$-output case stays the same as before.

