# OpenReview forum: "Restricted Strong Convexity of Deep Learning Models with Smooth Activations"
_ICLR.cc/2023/Conference — ICLR 2023 poster_

### Official Review · Reviewer_GhrZ · 2022-10-24

**Confidence:** 3
**Correctness:** 3
**Technical Novelty And Significance:** 3
**Empirical Novelty And Significance:** Not applicable
**Recommendation:** 8

**Clarity, Quality, Novelty And Reproducibility:**

_Clarity_:
The paper has some issues with clarity (as pointed out in the weaknesses section above), but overall I found it clear enough.

_Originality_:
This paper uses a considerably finer analysis than Liu et al, and obtain interesting results that are strictly better than the prior work. The work also establishes poly-time guarantees without the NTK assumption.



**Strength And Weaknesses:**

_Strengths_:
- This paper considerably improves the results in Liu et. al. with polynomial factors wherever the prior work had exponential factors.

- The use of RSC matches empirical observations that the Hessian of neural networks have many flat directions with eigenvalues $=0$.

- The gradient descent iterates are allowed to be in a much larger ball when compared to prior work.

_Weaknesses_:
- The flow of the paper is a little confusing. The authors first show an upper bound on the spectral norm of the Hessian (in an appropriate ball around the initialization), and then in the second paragraph of page 2, say that this will be used to show Restricted Strong Convexity. This doesn't make much sense when phrased this way, since RSC would require a lower bound on the smallest eigenvalue of the Hessian within the restricted set. My understanding of the paper is that the Hessian spectral bound has nothing to do with RSC, and is instead used for showing smoothness of the loss function, as in Theorem 5.2.

- In Theorem 4.1, the role of $\sigma_1$ is not clear. What stops you from choosing an extremely small $\sigma_1$? It seems like this would give a good value for $\gamma$. Assumption 2 requires it to bounded away from 0, but its significance in Theorem 4.1 is unclear.

- I'm a little concerned about the connections between the Lipschitz properties of the network (via the Lipschitz activations in Assumption 1) and the Restricted Strong Convexity properties. These two are conflicting properties, and the choice of $\rho, \rho_1$ would dictate whether the Lipschitz bound in Assumption 1 can be satisfied. Any clarification on this point would be appreciated.

- The geometric convergence in Theorem 5.3 should also come with the disclaimer that the loss cannot be made arbitrarily small. In order for A4.1 to hold, the average gradient needs to be sufficiently large, which implies that the algorithm cannot be arbitrarily close to the minimum (in which case the gradient would be arbitrarily close to zero). THis would also contradict the claim in Remark 5.4, where the authors say that there exist parameters that can drive the loss to an arbitrary $\epsilon$.

- Assumption A4.1 should appear before Theorem 5.1. Currently it is buried in the last sentence of Theorem 5.1.

_Clarifications_:

- This is not a complaint, but perhaps the choice of $\sigma_0$ can be compared with common initialization schemes, such as, Xavier initialization.

- The line below Assumption 3 says that cross-entropy is strongly convex. Is this true? (the loss in logistic regression for e.g. is certainly not strongly convex) If your results in Theorem 5.1 are for the square loss, why mention cross-entropy at all?

_Minor comments_:
- Para above lemma 4.1; "to established" -> "to establish"

**Summary Of The Paper:**

This paper analyzes the convergence of gradient descent on overparameterized neural networks with smooth activation functions. The authors show that the Hessian of the neural network has a small spectral norm within a special ball of the initialization, and this ball is allowed to be significantly larger than those allowed by prior works, such as those that use the NTK framework.

The authors then show that the square loss is strongly convex within a restricted set around the initialization (under certain assumptions). This along with the spectral norm bound (which implies smoothness for the loss function) gives a geometric convergence rate for gradient descent.

**Summary Of The Review:**

This result gives a poly-time (almost) geometric convergence for gradient descent on deep neural nets with smooth activations, and does not rely on the NTK framework.

The reasons for my score are:
1. the GD iterates are allowed to lie in a significantly larger set than prior work.
2. the paper has the added benefit that it closely resembles empirical observations -- it is well known that Hessians of neural nets have many directions in which the eigenvalues are zero, and the use of RSC implies that these directions can be accounted for.
3. the analysis is much sharper than prior work in Liu et al, and obtains poly factors wherever prior work obtains exponential factors.

---

> ### Author Response · Authors · 2022-11-13
> **Initialization scheme, Flow**
>
> **Comparison of initialization with other schemes, such as, Xavier initialization**
>
> For the initialization scheme, note that we assume $w^{(l)}_{0,ij} \sim {\cal N}(0,\sigma_0^2)$ where $\sigma_0 = \Theta(1)$. Since our model in equation (1) has a $\frac{1}{\sqrt{m}} W$ layerwise scaling, appllying the $1/\sqrt{m}$ scaling to elements of $W$, we note that our initialization is equivalent to having $O(1/m)$ variance entries in $W_0$ without any additional scaling. So, it is related to the Xavier initialization, with a constant possibly different from 1.
>
> Our scaling and model is also related to what has been studied in (Jakot et al., NeurIPS 2018)[Section 2] and (Du et al., ICML 2019)[Section 3.3].
>
> **Quote: ``In Theorem 4.1, the role of $\sigma_1$ is not clear. What stops you from choosing an extremely small $\sigma_1$?...''**
>
> To understand the effect of $\sigma_1$, we can look at how the NTK gets affected. For ReLU networks, (Nguyen et al., ICML 2021) show in their equation (7) that minimum eigenvalue of the NTK is effectively $\lambda_0 = \Theta(m^L\sigma_1^{2L})$, adopting to our scaling and notation. For convergence, (Du et al., ICML 2019) show that the width required for geometric convergence is $m = \Omega(\frac{n^4}{\lambda_0^4})$. Thus, as a function of $\sigma_1$, the width needed is $m = \Omega(\sigma_1^{-8L})$, ignoring other dependencies. Thus, making $\sigma_1$ small will end up needing a significantly larger width $m$. Something similar is expected to happen for smooth activations, though the analysis will need work.
>
> **Quote: ``Assumption A4.1 should appear before Theorem 5.1. Currently it is buried in the last sentence of Theorem 5.1''**
>
> Placing Assumption A4.1 before Theorem 5.1 could provide better visibility to it, but we hesitate because of two reasons.
> * Assumption A4.1 talks about the RSC parameter $\alpha_t$, a parameter which is defined for the first time in Theorem 5.1.
> * None of the results before Lemma 5.2 (which is the Lemma right after Assumption A4.1 is introduced) makes use of assumption A4.1, and thus placing our assumption much earlier can lead to confusion whether this is being used or not.

---

> ### Author Response · Authors · 2022-11-13
> **Lipschitz vs. RSC, gradient of loss vs. predictor**
>
>
> **Lipschitz and (R)SC are conflicting properties**
>
> You are right that Lipschitz and strong convexity are conflicting properties, but the issue can be understood in our setting. We assumed the layerwise activation functions to have a Lipschitz constant of 1, which hid the dependence. If the layerwise activations have Lipschitz constant $\zeta_{\phi}$, the network effectively has a $O(\zeta_{\phi}^{L})$ Lipschitz constant, and the Hessian spectral norm bound would be $\frac{c_H O(\zeta_{\phi}^{2L})}{\sqrt{m}}$, following Liu et al., NeurIPS 2021, Theorem 3.1.
>
> As a consequence, the RSC condition will become $|| \bar{\mathbf{g}} ||_2^2 = \Omega(\frac{\zeta^{2L}}{\sqrt{m}} )$.
>
> Thus, for RSC to hold, the width $m$ has to depend on $\zeta_{\phi}^{4L}$ to neutralize the term due to the Lipschitz constant, i.e., we will need a *much* wider network.
>
> Most activation functions used in practice have $\zeta_\phi =1$ or $\zeta_{\phi} = 1 + o(1)$ with $\zeta_{\phi}^{2L} = O(\text{poly}(L))$, so width dependence stays reasonable. Further, for constant depth networks $L=O(1)$, the dependence $\zeta_{\phi}^{2L} = O(1)$, so the width $m$ stays reasonable.
>
>
> **Quote: ``The geometric convergence in Theorem 5.3 should also come with the disclaimer that the loss cannot be made arbitrarily small. In order for A4.1 to hold, the average gradient needs to be sufficiently large, which implies that the algorithm cannot be arbitrarily close to the minimum (in which case the gradient would be arbitrarily close to zero).**
>
> Note that the RSC parameter $\alpha_t$ (in A4.1) is defined in terms of the norm of the **average gradient of the predictor**
>
> $$
> \bar{\mathbf{g_t}} = \frac{1}{n} \sum_{i=1}^n \nabla_\theta f(\theta_t, \mathbf{x}_i)
> $$
>
> not the **average gradient of the loss**. The difference is important in our context, especially in addressing your comment.
>
> When the value of the loss decreases, the norm of the *average gradient of the loss* is expected to decrease, but the norm of the *average of the gradient of the predictor* $|| \bar{\mathbf{g}}_t ||_2$ need not decrease. In fact, in Figure 1(b), (c) and (d) from our experimental results, we observe that $|| \bar{\mathbf{g}}_t ||_2$ stays bounded away from zero throughout the training, as the loss decreases.

---

> ### Author Response · Authors · 2022-11-13
> **Hessian Spectral Norm, RSC**
>
> Thank you for your comment on the upper and lower bounds on the eigenvalues of the Hessian, which is at the core of our analysis. Our first result (Section 4) is for the **Hessian of the predictor** $\frac{\partial^2 f(\theta;\mathbf{x}_i)}{\partial \theta^2}$ and we upper bound its spectral norm, whereas our second result (Section 5) relies on lower bounding the smallest eigenvalue of the **Hessian of the loss** $\frac{\partial^2 {\mathcal{L}}(\theta)}{\partial \theta^2}$ within the restricted set. As we briefly show below (details are in the proofs in the Appendix), to get the lower bound on the restricted minimum eigenvalue of $\frac{\partial^2 {\mathcal{L}}(\theta)}{\partial \theta^2}$ in Section 5, we in fact used the upper bound on the spectral norm of $\frac{\partial^2 f(\theta;\mathbf{x}_i)}{\partial \theta^2}$.
>
>
> To see the high level, we start with the Taylor expansion
>
>
> ${\cal L}(\theta') = {\cal L}(\theta_t) + \langle \theta' - \theta_t, \nabla_\theta {\cal L}(\theta_t) \rangle + \frac{1}{2} (\theta'-\theta_t)^\top \frac{\partial^2 {\cal L}(\tilde{\theta}_t)}{\partial \theta^2} (\theta'-\theta_t)~.$
>
> Focusing on the quadratic form in the Taylor expansion and recalling the form of the Hessian,  we can write
>
> $$
> (\theta'-\theta_t)^\top \frac{\partial^2 {\cal L}(\tilde{\theta}_t)}{\partial \theta^2} (\theta'-\theta_t)
> $$
>
> as
>
> $$
> (\theta'-\theta_t)^\top \frac{1}{n} \sum_{i=1}^n \left[ \ell''_i \frac{\partial f(\tilde{\theta}_t;\mathbf{x}_i)}{\partial \theta}   \frac{\partial f(\tilde{\theta}_t;\mathbf{x}_i)}{\partial \theta}^\top  + \ell'_i   \frac{\partial^2 f(\tilde{\theta}_t;\mathbf{x}_i)}{\partial \theta^2}   \right] (\theta'-\theta_t) = I_1 + I_2
> $$
>
> where
>
> $$
> I_1 = \frac{1}{n} \sum_{i=1}^n  \ell''_i \left\langle \theta'-\theta_t, \frac{\partial f(\tilde{\theta}_t;\mathbf{x}_i)}{\partial \theta} \right\rangle^2
> $$
>
> and
>
> $$
> I_2 =  \frac{1}{n} \sum_{i=1}^n \ell'_i  (\theta'-\theta_t)^\top \frac{\partial^2 f(\tilde{\theta}_t;\mathbf{x}_i)}{\partial \theta^2}  (\theta'-\theta_t)
> $$
>
> with $\ell_i=\ell(y_i,\hat{y}_i)$, $\hat{y}_i = f(\tilde{\theta}_t,\mathbf{x}_i)$ and $\ell'_i=\ell'(y_i,\hat{y}_i)$, $\ell''_i=\ell''(y_i,\hat{y}_i)$ where the derivates are w.r.t. the second argument.
>
> The RSC condition analysis focuses on $\theta' \in Q^t_{\kappa} \cap B_{\rho}(\theta_0)$, the restricted set, and constructs a lower bound of the quadratic form, which exactly corresponds to the smallest eigenvalue of the loss Hessian $\frac{\partial^2 {\cal L}(\tilde{\theta}_t)}{\partial \theta^2}$ on the restricted set.
>
> In the analysis, the $I_1$ term is lower bounded by a scaled version of $|| \bar{\mathbf{g}}_t||_2^2$, where
>
> $\bar{\mathbf{g_t}} = \frac{1}{n} \sum_{i=1}^n \nabla_{\theta} f(\theta_t;\mathbf{x}_i)$.
>
> For the $I_2$ term, the quadratic form can be negative since the Hessian of the predictor $\frac{\partial^2 f(\tilde{\theta}_t;\mathbf{x}_i)}{\partial \theta^2}$ is indefinite, and can have both positive and negative eigenvalues. We use the bound on the Hessian of the predictor from Section 4 is lower bound $I_2$. We also have a similar analysis establishing smoothness where both $I_1, I_2$ are upper bounded, and we use the result from Section 4 for upper bounding $I_2$.
>
> Thus, the spectral norm bound in Section 4 plays a central role in establishing RSC and smoothness in Section 5, which are in turn used for the geometric convergence analysis.

---

### Official Review · Reviewer_uzLy · 2022-10-24

**Confidence:** 1
**Clarity, Quality, Novelty And Reproducibility:** It is clear and has good writting qua…
**Correctness:** 3
**Technical Novelty And Significance:** 3
**Empirical Novelty And Significance:** 3
**Recommendation:** 6

**Strength And Weaknesses:**

The paper provides a good litterature review. It is well written and easy to follow. However, it is a dense paper with lots of theoretical proof to check. The ICLR review period is too short to allow a careful theoretical review. It would be better as a journal paper.



**Summary Of The Paper:**

This paper provides another sufficient condition besides NTK to ensure geometric convergence of deep learning.

**Summary Of The Review:**

This is seemingly a good paper, but too heavy to review for a ML conference with so short review periods. Hence, you should not consider my recommendation as meaningful.

---

> ### Author Response · Authors · 2022-11-13
> **Well written, lots of theoretical results**
>
> We are happy that you found our paper to be well-written and easy to follow. We do acknowledge that by its theoretical nature and technical depth, it is a paper that contains many theoretical results. For this reason, we have kept many remarks in the main paper to improve its exposition and moved all of the proofs to the Appendix. We hope that these choices were helpful for you when reading our paper.

---

### Official Review · Reviewer_oVua · 2022-10-25

**Confidence:** 3
**Correctness:** 3
**Technical Novelty And Significance:** 3
**Empirical Novelty And Significance:** Not applicable
**Recommendation:** 6

**Clarity, Quality, Novelty And Reproducibility:**

- The paper is generally well-written. The authors make particular effort to make the presentation less 'dry' with several remarks on intuitions and interpretations of their results
- The contributions are made clear and it is clarified what the novelties are compared to previous works

**Strength And Weaknesses:**

Strengths:

- The paper is generally well-written. The authors make particular effort to make the presentation less 'dry' with several remarks on intuitions and interpretations of their results
- The contributions are made clear and it is clarified what the novelties are compared to previous works
- Thm 1 improves on a previous result by (Liu et al.)
- Thm 5.2 provides an alternative a criterion for linear convergence that is different compared to the now common uniform minimum eigenvalue bound of the NTK

Weaknesses:

- The two contributions of the paper are not well connected. Have I missed something on this? Perhaps the authors can clarify how the content of Sec. 4 informs that of Sec. 5
- How is Lemma 5.1 different from what is already shown in Karimi et al. that RSC=>RPL other than here you are talking about restricted PL?
- The result on RSC requires (essentially) two assumptions.
(a)The first is that GD travels a path in which iterates are not orthogonal to the average gradient. While it is mentioned, that this is something observed in practice, no further evidence is given. Also, it would be useful to add some intuition on why this is needed.
(b) The second needed condition is that the squared norm of the average gradient is greater than c/\sqrt{m}, where m is the network's width. Remark 5.5 is nice, but it would strengthen the argument if the authors could show an example of concrete architecture where (20) holds but (19) does not.
- In Remark 5.6 they authors mention: "our perspective is to view the NTK and RSC as two different sufficient conditions ...". This remark is somewhat contradictory to the flavor of comment in the rest of the paper, e.g. Remark 5.5 also earlier that the layer-wise spectral ball does not require to be in NTK regime
- In the experiments: what does it mean, step-size is chosen appropriately to keep training in NTK regimes?
- minor: above Thm 1, cross-entropy is not strongly convex






**Summary Of The Paper:**

The paper makes two contributions for single-output deep learning models with smooth activations:
1) Upper bounds the spectral norm of the model's Hessian over a layer-wise spectral norm, which is larger than the euclidean ball that was studied in previous work. Hence, the result holds with relaxed conditions on how close it is to initialization.
2) proves a restricted strong convexity property along the path of GD iterates that avoid directions orthogonal to the average gradient and that stay close to initialization with respect to the layer-wise spectral norm.



**Summary Of The Review:**

Disclaimer: I am not super familiar with the details of some of the closely related works, but the authors appear to have done a good job in comparing their results to them and clarifying the novelties in a way that is convincing. Also, given the reviewing load, I was unable to go through the proofs. Because of these hesitations, I recommend 'marginal accept' for now, but I am willing to improve my score further down in the discussion period.

---

> ### Author Response · Authors · 2022-11-13
> **RSC, RPL, NTK**
>
> **RSC, RPL, and what is known as SC, PL (Karimi et al.)**
>
> As you wrote, SC $\implies$ PL is well known, e.g., Karimi et al., in $\mathbb{R}^p$. The main difference is indeed that we are talking about *restricted SC* and *restricted PL*, which needs to hold on a cone (the set $Q_\kappa^t$) intersecting a ball, and not on the entire $\mathbb{R}^p$ as in standard SC and PL. The concept of RPL seems new, it generalizes PL similar to how RSC generalizes SC. The math (Lemma 5.1) works out by paying attention to the restricted sets and using standard tools. The result plays a central role in how we build the overall proof.
>
>
> **Quote: ``RSC requires (essentially) two assumptions. (a) ... GD travels a path in which iterates are not orthogonal to the average gradient. ... (b) ... the squared norm of the average gradient is greater than $c/\sqrt{m}$''**
>
> For the RSC based convergence analysis, we only need (b) ``square norm of the average gradient is greater than $c/\sqrt{m}$.''
>
> We do not need (a) --- although we start off the analysis on the restricted set, our final result Theorem 5.3 is on the entire spectral norm ball $B_{\rho,\rho_1}^{\text{spec}}(\theta_0)$, i.e., equation (18) in final Theorem 5.3 implies decrease on the entire spectral norm ball, whereas the intermediate previous result in Lemma 5.2 implies decrease in a subset with paths that are not orthogonal to the average gradient.
>
> **RSC and NTK: ``Remark 5.5 is nice'', examples will help**
>
> We appreciate your comment on Remark 5.5, which shows the crux of the difference between the NTK and the RSC based approaches. For the difference between the two settings, we discuss this briefly in Remark 5.6. As a simple example, when $\mathbf{x}_i = \mathbf{x}_j, i \neq j$, the NTK analysis breaks down, whereas the RSC analysis would still work. Specifically, in the NTK analysis, the convergence speed of the loss is controlled by the minimum eigenvalue of NTK (e.g., see Theorem 5.1 in [Du et al., 2019]). The NTK analysis fails when the minimum eigenvalue is $0$, as it is in the case when $\mathbf{x}_i = \mathbf{x}_j, i \neq j$. However, in our RSC analysis, we allow the minimum eigenvalue of the NTK to be $0$ as we only need the norm of average gradient to be bounded from $0$.
> Further, for square loss and smooth activations, if $m = \Omega(n^2)$,  NTL $\implies$ RSC, so RSC is a more general condition for wide networks.
>
> **Quote: ``In Remark 5.6 they authors mention: "our perspective is to view the NTK and RSC as two different sufficient conditions ...". This remark is somewhat contradictory to the flavor of comment in the rest of the paper, e.g. Remark 5.5 also earlier that the layer-wise spectral ball does not require to be in NTK regime"**
>
> * Regarding Remark 5.6, as you stated, our use of laywerwise *spectral norm* ball is indeed an improvement over related results in the NTK literature.
> * In the infinite width limit $m \rightarrow \infty$, the RSC condition (in equation (20) in Remark 5.5) is always satisfied. We realized that we never remarked on the infinite width case in the manuscript.
> * If the average of gradients is zero, i.e., $\frac{1}{n}\sum_{i=1}^n \nabla_{\theta} f(\theta_t;\mathbf{x}_i) = 0$ so the RSC condition in (20) is violated, then the NTK condition in (19) is also violated since the gradients are linearly dependent.
>
> The reason behind Remark 5.6 is: for finite width $m$, one may be able to construct a set of gradient vectors such that the RSC condition in (20) is violated but the NTK condition in (19) is not violated. The good news is: the loss will decrease geometrically as long as one of RSC or NTK is satisfied in a step of the optimization. This is why we view them as two different sufficient conditions, and progress can be made as long as one of them is satisfied.
>
>
>
> **Experiments: how is step-size ``chosen appropriately to keep training in NTK regime''?**
>
> For wide neural networks, there is a critical learning rate $2/(\lambda_{\min}(K)+\lambda_{\max}(K))$, where $K$ is the NTK,  such that when training with learning rates smaller than the critical one, the network function evolves like a linear model in the NTK regime (Lee et al., 2019). For each experiment, we computed the critical learning rate and made sure the learning rate is smaller than that and thus the training is in NTK regime.
>
> Lee, Jaehoon, et al. "Wide neural networks of any depth evolve as linear models under gradient descent." Advances in neural information processing systems 32 (2019).
>
> **Above Thm 1, cross-entropy is not strongly convex**
>
> Thank you for catching this --- we have removed the statement.

---

> ### Author Response · Authors · 2022-11-13
> **Connection between two contributions -- Sections 4 and 5**
>
>
> The two contributions in Sections 4 and 5 are indeed well connected. The Hessian spectral norm bound in Section 4 plays a central role in deriving the RSC condition in Section 5. The proofs of Theorem 5.1 and 5.2 in Appendix B.2 has the details. To see the high level, we start with the Taylor expansion
>
>
> ${\cal L}(\theta') = {\cal L}(\theta_t) + \langle \theta' - \theta_t, \nabla_\theta {\cal L}(\theta_t) \rangle + \frac{1}{2} (\theta'-\theta_t)^\top \frac{\partial^2 {\cal L}(\tilde{\theta}_t)}{\partial \theta^2} (\theta'-\theta_t)~.$
>
> Focusing on the quadratic form in the Taylor expansion and recalling the form of the Hessian,  we can write
>
> $$
> (\theta'-\theta_t)^\top \frac{\partial^2 {\cal L}(\tilde{\theta}_t)}{\partial \theta^2} (\theta'-\theta_t)
> $$
>
> as
>
> $$
> (\theta'-\theta_t)^\top \frac{1}{n} \sum_{i=1}^n \left[ \ell''_i \frac{\partial f(\tilde{\theta}_t;\mathbf{x}_i)}{\partial \theta}   \frac{\partial f(\tilde{\theta}_t;\mathbf{x}_i)}{\partial \theta}^\top  + \ell'_i   \frac{\partial^2 f(\tilde{\theta}_t;\mathbf{x}_i)}{\partial \theta^2}   \right] (\theta'-\theta_t) = I_1 + I_2
> $$
>
> where
>
> $$
> I_1 = \frac{1}{n} \sum_{i=1}^n  \ell''_i \left\langle \theta'-\theta_t, \frac{\partial f(\tilde{\theta}_t;\mathbf{x}_i)}{\partial \theta} \right\rangle^2
> $$
>
> and
>
> $$
> I_2 =  \frac{1}{n} \sum_{i=1}^n \ell'_i  (\theta'-\theta_t)^\top \frac{\partial^2 f(\tilde{\theta}_t;\mathbf{x}_i)}{\partial \theta^2}  (\theta'-\theta_t)
> $$
>
> with $\ell_i=\ell(y_i,\hat{y}_i)$, $\hat{y}_i = f(\tilde{\theta}_t,\mathbf{x}_i)$ and $\ell'_i=\ell'(y_i,\hat{y}_i)$, $\ell''_i=\ell''(y_i,\hat{y}_i)$ where the derivates are w.r.t. the second argument.
>
> The RSC condition analysis focuses on $\theta' \in Q^t_{\kappa} \cap B_{\rho}(\theta_0)$, the restricted set, and constructs a lower bound of the quadratic form, which exactly corresponds to the smallest eigenvalue of the loss Hessian $\frac{\partial^2 {\cal L}(\tilde{\theta}_t)}{\partial \theta^2}$ on the restricted set.
>
> In the analysis, the $I_1$ term is lower bounded by a scaled version of $|| \bar{\mathbf{g}}_t||_2^2$, where
>
> $\bar{\mathbf{g_t}} = \frac{1}{n} \sum_{i=1}^n \nabla_{\theta} f(\theta_t;\mathbf{x}_i)$.
>
> For the $I_2$ term, the quadratic form can be negative since the Hessian of the predictor $\frac{\partial^2 f(\tilde{\theta}_t;\mathbf{x}_i)}{\partial \theta^2}$ is indefinite, and can have both positive and negative eigenvalues. We use the bound on the Hessian of the predictor from Section 4 is lower bound $I_2$. We also have a similar analysis establishing smoothness where both $I_1, I_2$ are upper bounded, and we use the result from Section 4 for upper bounding $I_2$.
>
> Thus, the spectral norm bound in Section 4 plays a central role in establishing RSC and smoothness in Section 5, which are in turn used for the geometric convergence analysis.

---

### Official Review · Reviewer_Qo6j · 2022-11-03

**Confidence:** 2
**Correctness:** 4
**Technical Novelty And Significance:** 2
**Empirical Novelty And Significance:** 2
**Recommendation:** 6

**Clarity, Quality, Novelty And Reproducibility:**

**Clarity**
The paper is well written and the main paper provides sufficient clarity on the main proof ideas. However some terms are used without properly introducing them.


**Novelty and quality**
The paper provides (and according to the authors, they are the first to do so) an convergence analysis that uses the Restricted Strong Convexity argument in the context of deep learning models. However the scope of the paper is limited to particular neural networks (see above comment) and not enough comparison is provided between their result and NTK.


**Strength And Weaknesses:**

Overall the paper provides a new approach to analyze the behavior of gradient descent when training deep learning models. Below I highlight my concerns and include some minor comments.

**Weakness**

-  The analysis presented is for NN with same width for all layers and with the output dimension of 1. The authors point to another paper for extension to multidimensional output. It would be beneficial to provide a remark, at the least, on the central steps required for extending the result.
-  Another restriction to the network is the activation function. Does the analysis extend to networks with smooth activation in last layer? Does it extend to non-smooth activation function?
- In my understanding, the result presented is for convergence to a neighborhood near the initialization. The main point made in the paper is that the networks does not have to be as wide for convergence to hold, unlike for NTK approaches. A comparison between the two methods on the required width size is missing.
- The experimental result presented is very limited. Following the results presented in the paper, the average gradient satisfies $\bar{g}_t = \Omega(\frac{L}{\sqrt{m}})$. However, figure 1b shows that the norm of the average gradient grows with network width. What accounts for this difference in theoretical vs experimental results.
- In the abstract, the authors state "...norm of the Hessian of such models...". This should be reworded to indicate that Hessian is of the loss function.
- In page 2, the authors state the set $Q_k^t$ without an explanation of what the set is (or referencing to where it is defined).
- In assumption 2, the initialization for $v_0$ is stated in a roundabout way. I believe it is simply sampled uniformly at random from a unit sphere.
- In Theorem 4.1, the bound on the maximum of the Hessian should either depend on $n$ or the bound holds for all $x_i$. This should be made clear.
- In Definition 5.2, missing inner product in $\cos(\theta -\theta_t, \bar{g}_t)$.

**Summary Of The Paper:**

In the paper, the authors analyze the convergence of gradient descent (GD) on training a deep learning model with smooth activation function. The paper is presented for fully connected neural network with linear last layer, output dimension of 1, and equal layer width for all layers. Under reasonable assumptions on the loss function, activation function, randomness of the initialization, etc., the authors show that GD will converge (geometrically) to the minimizer in the neighborhood around the initialization. The analysis establishes an upper bound on the Hessian of the loss function in the neighborhood and then using Restricted Strong Convexity of the loss function to argue for convergence to the minimizer. The approach for showing convergence differs from NTK approaches which are generally restricted to wide networks.


**Summary Of The Review:**

The stated problem is interesting and the authors provide an insight on the behavior of gradient descent for neural networks without large width. Overall, the paper is easy to read but contains some unclear parts.

---

> ### Author Response · Authors · 2022-11-13
> **Extensions**
>
> **Extension to k output neural networks**
>
> We have added Section C in the Appendix where we expand our results to a neural network with $k$ outputs. As expected, an extra factor of $k$ appears in the RSC and smoothness parameters. With the RSC and smoothness results, the optimization analysis, i.e., Lemma 5.1 onwards, stays the same, so we have not repeated those results.
>
>
> **Extension to smooth activation in the last layer**
>
> We believe our results could be extended to smooth activations in the last layer, but such extension may require extra assumptions on such non-linearities. For example, in [Liu et al., 2022], the introduction of an activation function on the output layer required the introduction of an additional assumption: a uniform lower bound on the first derivative. Due to the additional and potentially involved analysis this extension would imply in our specific setting, we will pursue this direction as future work.
>
>
> **Extension to non-smooth activation function**
>
> Extending our results directly to non-smooth activation functions such as ReLU is tricky, and at least will need a different proof technique. Our current analysis depends on explicitly upper bounding the spectral norm of the Hessian of the model and the RSC condition uses this bound. Since the Hessian is not well-defined for non-smooth activation functions, the challenge will be to find some alternative mechanism that would allow us to prove the RSC condition without the use of Hessian bounds for the model.

---

> ### Author Response · Authors · 2022-11-13
> **Clarifications**
>
>
> **Comparison with NTK, implications for width**
>
> The core comparison between RSC and NTK is in Remark 5.5, with (19) for NTK and (20) for RSC. Note that the RSC condition in (20) needs  $|| \bar{\mathbf{g}}_t ||_2^2 = \Omega(1/\sqrt{m})$   where
>
> $\bar{\mathbf{g_t}} = \frac{1}{n} \sum_{i=1}^n \nabla_{\theta} f(\theta_t;\mathbf{x}_i)$.
>
> The condition is based on the *gradient of the predictor* not the *gradient of the loss*. The RSC condition is more geometric and needs the predictor gradients to not cancel out. In terms of width, if $m = \Omega(n^2)$, then NTK condition implies the RSC condition, but RSC condition may be satisfied with a much smaller width.
>
>
> **Figure 1b, potential difference in theoretical vs experimental results?**
>
> Thank you for the opportunity to clarify our results in Figure 1b. The plot does not really show a difference between theoretical vs.~experimental results. The RSC condition (see Remark 5.1) does not provide a theoretical *lower bound* on the norm of the average gradient, but instead provides a *lower bound* that the norm of the average gradient would need to satisfy in order to ensure RSC. What Figure 1b shows is that as the width grows, the minimum value of the norm of the average gradient also grows, and thus becomes more likely to satisfy the RSC condition in all the training trajectory.
>
>
>
> **Abstract: Hessian of the model vs. Hessian of the loss**
>
> The statement in the abstract is correct--we establish an upper bound on the spectral norm of the Hessian of the neural network model itself (see Theorem 4.1 and Remark 4.2), not the Hessian of the loss function. We analyze the Hessian of the loss later in Section 5, leading to the RSC, smoothness, and convergence guarantees.
>
>
>
> **Page 2 discussion of the set $Q^t_\kappa$**
>
> Since the first mention of $Q^t_\kappa$ is in page 2, Section 1, we do not want to unpack the details here. Based on your comment, we have added a forward reference to the precise definition in Section 5.
>
>
> **Assumption 2, the initialization for $\mathbf{v}_0$**
>
> For the initialization for $\mathbf{v}_0$, our results go through as long as $||\mathbf{v}_0||_2 = 1$, without making any specific assumptions on the distribution on the unit ball. A uniform distribution on the unit ball will satisfy the condition, but is a special case of our result.
>
> **Theorem 4.1, the bound on the maximum of the Hessian should depend on $n$ or hold uniformly for all $\mathbf{x}_i$**
>
> You are making a good point, we agree that the expression $\max_{i \in [n]}$ in Theorem 4.1 can lead to confusion. Theorem 4.1 in fact holds uniformly for any $\mathbf{x}$ with $|| \mathbf{x}||_2 = \sqrt{d}$, which is the scaling we mention at the beginning of Section 4 (after Assumption 2) but which we did not include in the statement of the Theorem 4.1. We have now fixed this by including the scaling of the input data in Assumption 1. We have also removed the max to avoid confusion.
>
> **Definition 5.2, missing inner product in $\cos(\theta - \theta_t, \bar{\mathbf{g}}_t)$**
>
> Definition 5.2 is not missing any inner product, it just happens that we have not properly defined what the symbol $\text{cos}(\cdot,\cdot)$ on two vectors means (its their cosine similarity). We have now included such definition right before Definition 5.2, which should address your concern.

---

### Author Response · Authors · 2022-11-13
**Relationship between Hessian bound (Section 4) and RSC+Optimization (Section 5)**

Two reviewers have asked for more details on the connection between the two key contributions: the Hessian spectral norm bound (Section 4) and the RSC+Optimization (Section 5). The details got delegated to the Appendic and here we highlight the central aspect of how they are related.

The Hessian spectral norm bound in Section 4 plays a central role in deriving the RSC condition in Section 5. The proofs of Theorem 5.1 and 5.2 in Appendix B.2 has the details. To see the high level, we start with the Taylor expansion


${\cal L}(\theta') = {\cal L}(\theta_t) + \langle \theta' - \theta_t, \nabla_\theta {\cal L}(\theta_t) \rangle + \frac{1}{2} (\theta'-\theta_t)^\top \frac{\partial^2 {\cal L}(\tilde{\theta}_t)}{\partial \theta^2} (\theta'-\theta_t)~.$

Focusing on the quadratic form in the Taylor expansion and recalling the form of the Hessian,  we can write

$$
(\theta'-\theta_t)^\top \frac{\partial^2 {\cal L}(\tilde{\theta}_t)}{\partial \theta^2} (\theta'-\theta_t)
$$

as

$$
(\theta'-\theta_t)^\top \frac{1}{n} \sum_{i=1}^n \left[ \ell''_i \frac{\partial f(\tilde{\theta}_t;\mathbf{x}_i)}{\partial \theta}   \frac{\partial f(\tilde{\theta}_t;\mathbf{x}_i)}{\partial \theta}^\top  + \ell'_i   \frac{\partial^2 f(\tilde{\theta}_t;\mathbf{x}_i)}{\partial \theta^2}   \right] (\theta'-\theta_t) = I_1 + I_2
$$

where

$$
I_1 = \frac{1}{n} \sum_{i=1}^n  \ell''_i \left\langle \theta'-\theta_t, \frac{\partial f(\tilde{\theta}_t;\mathbf{x}_i)}{\partial \theta} \right\rangle^2
$$

and

$$
I_2 =  \frac{1}{n} \sum_{i=1}^n \ell'_i  (\theta'-\theta_t)^\top \frac{\partial^2 f(\tilde{\theta}_t;\mathbf{x}_i)}{\partial \theta^2}  (\theta'-\theta_t)
$$

with $\ell_i=\ell(y_i,\hat{y}_i)$, $\hat{y}_i = f(\tilde{\theta}_t,\mathbf{x}_i)$ and $\ell'_i=\ell'(y_i,\hat{y}_i)$, $\ell''_i=\ell''(y_i,\hat{y}_i)$ where the derivates are w.r.t. the second argument.

The RSC condition analysis focuses on $\theta' \in Q^t_{\kappa} \cap B_{\rho}(\theta_0)$, the restricted set, and constructs a lower bound of the quadratic form, which exactly corresponds to the smallest eigenvalue of the loss Hessian $\frac{\partial^2 {\cal L}(\tilde{\theta}_t)}{\partial \theta^2}$ on the restricted set.

In the analysis, the $I_1$ term is lower bounded by a scaled version of $|| \bar{\mathbf{g}}_t||_2^2$, where

$\bar{\mathbf{g_t}} = \frac{1}{n} \sum_{i=1}^n \nabla_{\theta} f(\theta_t;\mathbf{x}_i)$.

For the $I_2$ term, the quadratic form can be negative since the Hessian of the predictor $\frac{\partial^2 f(\tilde{\theta}_t;\mathbf{x}_i)}{\partial \theta^2}$ is indefinite, and can have both positive and negative eigenvalues. We use the bound on the Hessian of the predictor from Section 4 is lower bound $I_2$. We also have a similar analysis establishing smoothness where both $I_1, I_2$ are upper bounded, and we use the result from Section 4 for upper bounding $I_2$.

Thus, the spectral norm bound in Section 4 plays a central role in establishing RSC and smoothness in Section 5, which are in turn used for the geometric convergence analysis.

---

### Decision · Program_Chairs · 2023-01-20

**Decision:**

Accept: poster

**Justification For Why Not Higher Score:**

The paper contributes technical improvements (better hessian bounds) and a new RSC approach to proving linear convergence of gradient descent. The large-predictor gradient assumption is corroborated empirically, and could benefit from future analytical work (e.g., proofs that for certain data models this obtains with high probability).

**Justification For Why Not Lower Score:**

The paper significantly improves over previous bounds on the hessian, and also contributes a new approach to proving linear convergence of gradient descent in randomly initialized networks, without needing to resort to the neural tangent kernel. The paper merits acceptance in some form.

**Metareview: Summary, Strengths And Weaknesses:**

The paper studies the convergence of randomly initialized gradient descent for training deep networks with twice differentiable activations. It proves an upper bound on the hessian at random initialization, which holds over the set of networks whose weight matrices lie in a layer-wise spectral norm ball of relatively large radius. These bounds are polynomial in network depth L, for well-chosen values of the initialization variance. This in turn implies geometric convergence of gradient descent, using a restricted strong convexity argument which argues that when the overall loss is not too close to stationary, there is positive curvature in directions with large inner product with the data-averaged gradient of the predictor. This gives a means of analyzing the convergence of gradient descent in randomly initialized networks which is complementary to the standard neural tangent kernel (NTK) approach, and may allow moderate-to-large changes in the weight matrices.

All reviewers positively evaluated the paper’s contributions, noting that (i) the technical results controlling the hessian represent a substantial improvement over related previous work (esp., the paper of Liu et al), and that (ii) restricted strong convexity gives a novel framework for proving linear convergence of gradient descent in randomly initialized networks, which may be well-suited to overparameterized networks, which have many directions of flat or almost-flat training loss. Reviewers, in general, praised the paper’s clarity. Reviewers raised several concerns about presentation and relationship to previous results, which were largely addressed in author responses. Results are conditional, in the sense that they require a lower bound on the data-averaged predictor gradient to hold at all iterations -- this hypothesis is corroborated in experiments, and can be considered an assumption of the analysis. For wide networks, the paper shows that this condition is weaker than the NTK eigenvalue condition. Give the limitations of existing analyses of randomly initialized gradient descent, the paper contributes both technical improvements and a promising direction.

**Note From Pc:**

if the above contains the word "oral" or "spotlight" please see: "oral" presentation means -> notable-top-5% and "spotlight" means -> notable-top-25%. As stated in our emails, we are disassociating presentation type from AC recommendations